# Birch SGD: A Tree Graph Framework for Local and Asynchronous SGD Methods

**Alexander Tyurin**
AXXX, Moscow, Russia
Applied AI Institute, Moscow, Russia

**Danil Sivtsov**
AXXX, Moscow, Russia
Applied AI Institute, Moscow, Russia

## Abstract

We propose a new unifying framework, Birch SGD, for analyzing and designing distributed SGD methods. The central idea is to represent each method as a weighted directed tree, referred to as a *computation tree*. Leveraging this representation, we introduce a general theoretical result that reduces convergence analysis to studying the geometry of these trees. This perspective yields a purely graph-based interpretation of optimization dynamics, offering a new and intuitive foundation for method development. Using Birch SGD, we design eight new methods and analyze them alongside previously known ones, with at least six of the new methods shown to have optimal computational time complexity. Our research leads to two key insights: (i) all methods share the same "iteration rate" of $\mathcal{O}\left((R+1)L\Delta/\varepsilon + \sigma^2 L\Delta/\varepsilon^2\right)$, where $R$ the maximum "tree distance" along the main branch of a tree; and (ii) different methods exhibit different trade-offs—for example, some update iterates more frequently, improving practical performance, while others are more communication-efficient or focus on other aspects. Birch SGD serves as a unifying framework for navigating these trade-offs. We believe these results provide a unified foundation for understanding, analyzing, and designing efficient asynchronous and parallel optimization methods.

## 1 Introduction

Optimization is central to machine learning (ML), data science (DS), and federated learning (FL) (Konečný et al., 2016; Bottou et al., 2018; Kairouz et al., 2021). In these domains, stochastic optimization techniques such as stochastic gradient descent (SGD) (Robbins & Monro, 1951) and its adaptive variants (Adam, AdamW, etc) (Kingma & Ba, 2015; Loshchilov & Hutter, 2019) have become the standard approach for tackling large-scale problems (Schmidt et al., 2021). Due to the rising computational demands of modern functions, the theoretical foundation of distributed algorithms supporting a large number of workers (e.g., CPUs, GPUs, servers) is important (Mayer & Jacobsen, 2020; Kairouz et al., 2021; Douillard et al., 2023).

We consider distributed optimization problems with smooth nonconvex optimization functions:

$$\min_{x\in\mathbb{R}^d} f(x), \tag{1}$$

In nonconvex settings, the goal is to find an $\varepsilon$–stationary point, meaning we want to find a random vector $\bar{x}$ such that $\mathbb{E}\left[\|\nabla f(\bar{x})\|^2\right] \leq \varepsilon$ (Nemirovskij & Yudin, 1983; Murty & Kabadi, 1985). The function $f : \mathbb{R}^d \to \mathbb{R}$ satisfies the following standard assumptions:

**Assumption 1.1.** $f$ is differentiable and $L$–smooth: $\|\nabla f(x) - \nabla f(y)\| \leq L\|x - y\|\ \forall x, y \in \mathbb{R}^d$.

**Assumption 1.2.** There exist $f^* \in \mathbb{R}$ such that $f(x) \geq f^*$ for all $x \in \mathbb{R}^d$.

We focus on problems where workers are limited to computing stochastic gradients. Each worker has access to unbiased stochastic gradients, denoted by $\nabla f(x; \xi)$, whose variance is bounded by $\sigma^2$. In the context of ML, this implies that all workers can access the same data, which is practical when training large language and computer vision models. In such scenarios, privacy is not a critical concern, and devices can sample data from the Internet or shared datasets (Goodfellow et al., 2016).

**Assumption 1.3.** For all $x \in \mathbb{R}^d$, stochastic gradients $\nabla f(x; \xi)$ are unbiased and $\sigma^2$-variance-bounded, i.e., $\mathbb{E}_\xi[\nabla f(x; \xi)] = \nabla f(x)$ and $\mathbb{E}_\xi[\|\nabla f(x; \xi) - \nabla f(x)\|^2] \leq \sigma^2$, where $\sigma^2 \geq 0$.

## 1.1 RELATED WORK

**One worker and optimal oracle complexity.** With a single worker, the most standard optimization method is the Vanilla SGD algorithm, which updates the iterate as $w^{k+1} = w^k - \gamma \nabla f(w^k; \eta^k)$, where $\{\eta^k\}$ are i.i.d., $w^0 \in \mathbb{R}^d$ is a starting point, $\gamma$ is a step size, and $\Delta := f(w^0) - f^*$. Arjevani et al. (2022); Carmon et al. (2020) showed that Vanilla SGD is *optimal* in terms of oracle complexity, which is given by $\Theta\left(L\Delta/\varepsilon + \sigma^2 L\Delta/\varepsilon^2\right)$ for finding an $\varepsilon$-stationary point.

**Multiple workers and optimal time complexity.** Consider now that we have $n$ workers computing stochastic gradients asynchronously and in parallel. In this setup, there are numerous ways to construct a distributed SGD method. The most well-known celebrated and recent approaches include Synchronized SGD (Minibatch SGD), Local SGD (Zinkevich et al., 2010; Stich, 2019), Asynchronous SGD (Recht et al., 2011), Picky SGD (Cohen et al., 2021), Rennala SGD (Tyurin & Richtárik, 2023), and Ringmaster ASGD (Maranjyan et al., 2025). The multi-worker setup is rich and versatile, offering numerous ways to design distributed SGD methods.

One may naturally ask which method offers the best theoretical performance. In distributed settings, the standard oracle complexity becomes less informative, as workers compute stochastic gradients in parallel with varying speeds. A more suitable comparison uses the $h_i$-*fixed computation model* (Mishchenko et al., 2022), where each worker $i$ needs at most $h_i$ seconds to compute a gradient. In this model, Mishchenko et al. (2022); Koloskova et al. (2022) showed that Asynchronous SGD outperforms Synchronized SGD. Its time complexity is further improved by Rennala SGD (Tyurin & Richtárik, 2023) and Ringmaster ASGD[1] (Maranjyan et al., 2025), both optimal under this and the more general *universal computation model* (Tyurin, 2025) (see Section A). However, as we will discuss in more detail later, other factors come into play, such as communication complexity, support for `AllReduce`, peak bandwidth, and model update frequency.

These developments raise several important questions. Rennala SGD and Ringmaster ASGD are known to be optimal, yet differ in design and structure, each with distinct advantages and trade-offs. This leads to our central questions: *Are there other optimal methods? Can we develop a unified framework that encompasses all distributed* SGD *methods and offers theoretical guidance? What fundamental properties make these methods optimal? And, given different system constraints, which method should one choose?*

## 1.2 CONTRIBUTIONS

♠ **New framework: Birch SGD (Section 2).** We propose Birch SGD, a unifying framework that captures a wide range of distributed SGD methods. The key idea is that SGD methods can be represented using weighted directed trees, which we refer to as *computation trees* (see Figure 1). We develop a new theoretical result, Theorem 2.4, that reduces the analysis of SGD methods to analyzing of the structure of these computation trees. The proofs become purely geometric and topological in nature, offering geometric intuition for the design of new methods. Moreover, this geometric viewpoint leads to tighter time complexity guarantees even for Local SGD (FedAvg) approaches (McMahan et al., 2017), as we illustrate in Section H.

♣ **Eight new methods (Table 1 and Section 3).** Using Birch SGD, we identify eight new methods in addition to those already known. For the first time, we prove that at least **six of these newly discovered methods are computationally optimal**, matching the lower bound (Tyurin & Richtárik, 2023). We compare all methods across several dimensions, including computational and communication complexity, `AllReduce` compatibility, peak bandwidth, and model update frequency. Our improvements: i) our newly developed Async-Local SGD and Async-Batch SGD provably improve the communication complexity of Ringmaster ASGD while preserving asynchronicity; ii) we introduce Cycle SGD, which provably reduces peak bandwidth compared to all existing methods; iii) we propose a key modification to the family of local methods and design Local SGD and Dual-Process SGD that, for the first time in the literature, achieve the optimal time complexities within this family and improve upon the classical approach (see Section H); iv) for multi-cluster settings, we introduce Local-Async SGD and Nested Local-Async SGD, incorporating a carefully designed synchronization mechanism that guarantees optimality in computational time complexity; v) we develop a flexible meta-algorithm, Meta Local SGD, which supports arbitrary synchronization strategies, while incorporating a "Hard

---

[1]Asynchronous SGD with a key modification; see Alg.7.

**Algorithm 1** Birch SGD framework

**Input:** starting point $w^0 \in \mathbb{R}^d$, step size $\gamma \geq 0$
Initialize the set of computed points: $V = \{w^0\}$
(and the set of edges $E = \emptyset$)
**for** $k = 0, 1, 2, \ldots$ **do**
    Choose any point $w_{\text{base}} \in V$ from which to compute a new point
    Choose any point $w_{\text{grad}} \in V$ at which to compute a stochastic gradient
    Compute the new point[2]: $w^{k+1} = w_{\text{base}} - \gamma \nabla f(w_{\text{grad}}; \eta), \quad \eta \sim \mathcal{D}_\xi$
    Add $w^{k+1}$ to the set of computed points $V$
    (and add the edge with weight $(w_{\text{base}}, w^{k+1}, \nabla f(w_{\text{grad}}; \eta))$ to the set of edges $E$)
**end for**

*February Azure*,
Igor Grabar. 1904.

Figure 1: A possible computation tree $G$ for SGD method after four steps and beyond.

Sync" mechanism to guarantee convergence rates and to temper overly chaotic synchronization. As a byproduct, we prove that frequent model updates of fully asynchronous methods can lead to faster convergence and improve optimal Rennala SGD.

♦ **Insights and Guidelines (Section 4).** We observe that there is no silver bullet—each method has its own advantages and disadvantages. Some methods update the iterates more frequently, making them more appealing in practice, while others prioritize communication efficiency, support AllReduce, or focus on different aspects. Through our new framework, we uncover insights that provide deeper intuition and a simpler perspective on asynchronous, local, and parallel optimization methods.

## 2 Birch SGD: A General View of SGD Methods

We begin our work by observing that various SGD methods, including Vanilla SGD, Asynchronous SGD, Local SGD, among others, can be constructed in the manner described in Algorithm 1.

Let us explain it. Initially, any SGD method starts at some point $w^0 \in \mathbb{R}^d$, computes a stochastic gradient at $w^0$, and then finds a new point $w^1 = w^0 - \gamma \nabla f(w^0; \cdot)$, which is added to the set $V$ of computed points. In the next step, there are four options for choosing the subsequent point $w^2$: $w^2 = w^i - \gamma \nabla f(w^j; \cdot)$ for $i, j \in \{0, 1\}$. This process continues indefinitely, and the number of possible choices, and hence methods, grows exponentially (see an example in Figure 1).

Note that any instance of Algorithm 1, after any steps, can be represented by a weighted directed tree $G = (V, E)$, called a *computation tree*, where $V$ is the set of computed points and $E$ is the set of edges with weights given by the stochastic gradients used to compute the new points. Our main idea now is to take any computation tree $G$ and analyze its structure to provide convergence guarantees for the corresponding SGD method. Intuitively, the structure of the tree, e.g., number of branches, length of branches, the tree distance between $w_{\text{grad}}$ and $w_{\text{base}}$ in Alg. 1 when we calculate a new point should be related to the convergence speed of the method.

**Example.** Consider Local SGD from Figure 4. There, we illustrate two global steps of the method with 2 workers. In the first round, they compute $M_1 = 2$ and $M_2 = 2$ local steps (first figure in Figure 4). During these steps, worker 1 first calculates $\nabla f(x^0; \eta_1^{0,0})$, finds $z_1^{0,1} = x^0 - \gamma \nabla f(x^0; \eta_1^{0,0})$, then calculates $\nabla f(z_1^{0,1}; \eta_1^{0,1})$ and $z_1^{0,2} = x^0 - \gamma \nabla f(z_1^{0,1}; \eta_1^{0,1})$. Similar steps are performed by worker 2. Then, via a parameter-server or AllReduce, LocalSGD aggregates the stochastic gradients and performs the global step $x^0 - \gamma(\nabla f(x^0; \eta_1^{0,0}) + \nabla f(z_1^{0,1}; \eta_1^{0,1}) + \nabla f(x^0; \eta_2^{0,0}) + \nabla f(z_2^{0,1}; \eta_2^{0,1}))$ to obtain the new global point $x^4$, from which the second global steps will start. The step of

finding $x^4$ is equivalent to the steps $x^1 = x^0 - \gamma\nabla f(x^0; \eta_1^{0,0})$, $x^2 = x^1 - \gamma\nabla f(z_1^{0,1}; \eta_1^{0,1})$, $x^3 = x^2 - \gamma\nabla f(z_1^{0,1}; \eta_1^{0,1})$, and $x^4 = x^3 - \gamma\nabla f(z_2^{0,1}; \eta_2^{0,1})$. This is how we construct the second figure in Figure 4, which is a geometric representation of the first global step. Then, the workers compute $M_1 = 1$ and $M_2 = 3$ local steps, accordingly, and synchronize again (third figure in Figure 4) to find $x^8$, from which the third global steps will start.

## 2.1 MAIN THEORETICAL RESULT ON CONVERGENCE RATES

Before we state our main theorem, we need to introduce sequences and definitions that characterize the structure of computation trees $G$.

**Definition 2.1** (*Main Branch* and *Auxiliary Sequence*)**.** For a given computation tree $G$, we call a sequence $\{x^k\}_{k\geq 0}$ a *main branch* if it forms a path in $G$ starting at the initial node $w^0 \equiv x^0$. That is, for each $k \geq 0$, the node $x^{k+1}$ is a direct successor of $x^k$ in $G$. By the construction of tree $G$, if $\{x^k\}_{k\geq 0}$ is a *main branch*, then for each $k \geq 0$ there exists a unique pair $(z^k, \xi^k)$, where $z^k \in V$ and $\xi^k \sim \mathcal{D}_\xi$, such that $x^{k+1} = x^k - \gamma\nabla f(z^k; \xi^k)$. The sequence $\{(z^k, \xi^k)\}_{k\geq 0}$, which generates the main branch $\{x^k\}_{k\geq 0}$, is called an *auxiliary sequence*.

Although there may be several possible choices and any of them can be chosen in general, the selection of the *main branch* is typically unique and straightforward in all reasonable SGD methods, as it forms the backbone of the tree[3].

Let us consider an example. In Figure 1, we can take a main branch $\{x^k\}_{k\geq 0}$ as follows: $x^0 = w^0, x^1 = w^2, x^2 = w^3, x^3 = w^8, x^4 = w^9, x^5 = w^{10}$. Accordingly, the auxiliary sequence is given by $(z^0, \xi^0) = (w^0, \eta^0), (z^1, \xi^1) = (w^1, \eta^1), (z^2, \xi^2) = (w^2, \eta^2), (z^3, \xi^3) = (w^4, \eta^6), (z^4, \xi^4) = (w^1, \eta^3)$. See Figure 2.

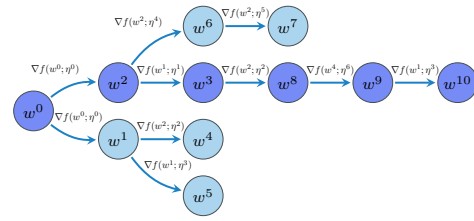

Figure 2: Visualization.

Intuitively, the convergence rate should depend on the distance between $x^k$ and $z^k$. When these points are close (e.g., $x^k = z^k$), the stochastic gradient is computed near the update point, typically yielding descent on average. In contrast, if they are far apart, the gradient at $z^k$ may poorly approximate the local behavior of $f$ at $x^k$, making the update direction irrelevant. Thus, it is crucial to define a suitable distance metric that is both easy to evaluate for any point pair and directly related to the convergence speed of the SGD method. We propose the following:

**Definition 2.2.** For all $y, z \in V$, the tree distance $\text{dist}(y, z)$ between $y$ and $z$ is the maximum number of edges to the common closest ancestor of $y$ and $z$.

As an example, consider Figure 2, where $\text{dist}(w^9, w^4) = \max\{4, 2\} = 4$, because the common ancestor is $w^0$, the number of edges from $w^9$ to $w^0$ is 4, and the number of edges from $w^4$ to $w^0$ is 2. It is left to define the *representation* of a point $y \in V$.

**Definition 2.3.** For all $y \in V$, the representation $\text{repr}(y)$ is the multiset of stochastic gradients applied to $w^0$ to get $y$. In other words, there exist $\{(m^1, \kappa^1), \ldots, (m^p, \kappa^p)\} =: \text{repr}(y)$ for some $p \geq 0$ such that $y = w^0 - \gamma\sum_{j=1}^p \nabla f(m^j, \kappa^j)$.

We define the representation of points to understand how all points are related. An important relation that we need is that $\text{repr}(x) \subseteq \text{repr}(y)$, which essentially means that all stochastic gradients used to compute $x$ are also used to compute $y$. For instance, in Figure 2, $\text{repr}(w^9) = \{(w^0, \eta^0), (w^1, \eta^1), (w^2, \eta^2), (w^4, \eta^6)\}$ and $\text{repr}(w^4) = \{(w^0, \eta^0), (w^2, \eta^2)\}$, which allows to track the path from from the starting point $w^0$ to $w^9$ and $w^4$, and show that $\text{repr}(w^4) \subseteq \text{repr}(w^9)$.

---

[3]A fitting analogy is the Git distributed version control system, which also has a central main branch.

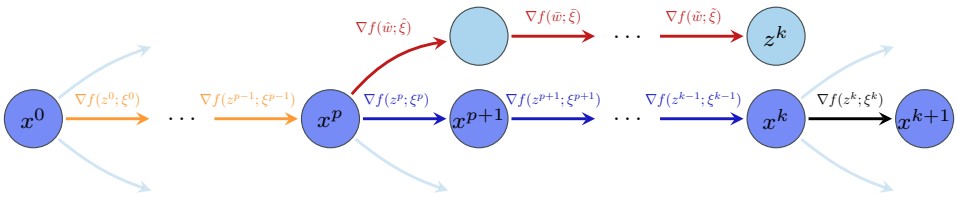

Figure 3: A general representation of the step $x^{k+1} = x^k - \gamma \nabla f(z^k; \xi^k)$ that shows how $x^k$ and $z^k$ are graph-geometrically related.

---

**Theorem 2.4** (Main Theorem). *Let Assumptions 1.1, 1.2, and 1.3 hold. Consider any* SGD *method represented by* computation tree $G = (V, E)$. *Let* $\{x^k\}_{k \geq 0}$ *be a* main branch *of* $G$ *and* $\{(z^k, \xi^k)\}_{k \geq 0}$ *be the corresponding* auxiliary sequence *(see Def. 2.1) that satisfy the following conditions:*
**Condition 1:** *For all* $k \geq 0$, $\xi^k$ *is statistically independent of* $\{(x^{i+1}, z^{i+1}, \xi^i)\}_{i=0}^{k-1}$.
**Condition 2:** *The representation of* $z^k$ *is contained within that of* $x^k$, *i.e.,* $\mathrm{repr}(z^k) \subseteq \mathrm{repr}(x^k)$ *for all* $k \geq 0$. *Equivalently, all stochastic gradients used in the computation of* $z^k$ *are also utilized in calculating* $x^k$.
**Condition 3:** *There exists a constant* $R \in [0, \infty]$ *such that* $\mathrm{dist}(x^k, z^k) \leq R$ *for all* $k \geq 0$. *Then* $\frac{1}{K}\sum_{k=0}^{K-1} \mathbb{E}\left[\|\nabla f(x^k)\|^2\right] \leq \varepsilon$ *for all* $K \geq \frac{4(R+1)L\Delta}{\varepsilon} + \frac{8\sigma^2 L\Delta}{\varepsilon^2}$ *with step size* $\gamma = \min\{\frac{1}{2L}, \frac{1}{2RL}, \frac{\varepsilon}{4\sigma^2 L}\}$, *where* $\Delta = f(x^0) - f^*$.

---

Assumptions 1.1, 1.2, and 1.3 are well-known and standard in the analysis of stochastic optimization methods (Lan, 2020; Arjevani et al., 2022). Let us explain the conditions of the theorem.

**Condition 1.** The first condition condition requires that $\xi^k$ is independent of $\{(x^{i+1}, z^{i+1}, \xi^i)\}_{i=0}^{k-1}$, which is a weak assumption. In Vanilla SGD, where $x^{k+1} = x^k - \gamma \nabla f(x^k; \xi^k)$, it is standard to assume that each $\xi^k$ is an independent sample. Our condition generalizes this to other SGD variants. It guarantees that the stochastic gradient $\nabla f(\cdot; \xi^k)$ is not used in computing $x^k$ or $z^k$. Notably, this remains true even in methods like Local SGD, where gradients may be reused.

**Condition 2.** The second condition is also weak in any *reasonable and effective* SGD method. Figure 3 illustrates that there exists $p \geq 0$ such that

$$z^k = x^0 - \gamma \sum_{i=0}^{p-1} \nabla f(z^i; \xi^i) - \gamma \sum_{(w,\xi) \in S^k} \nabla f(w; \xi), \quad x^k = x^0 - \gamma \sum_{i=0}^{p-1} \nabla f(z^i; \xi^i) - \gamma \sum_{i=p}^{k-1} \nabla f(z^i; \xi^i),$$

where $S^k$ is the set of points and random variables used to compute $z^k$ starting from $x^p$.

Computing each stochastic gradient is time-consuming, so it is desirable to utilize as many computed gradients as possible, including $\{\nabla f(w; \xi)\}_{(w,\xi) \in S^k}$. Once $\nabla f(z^k; \xi^k)$ has been used to compute $x^{k+1}$, the first condition prevents further use of $\{\nabla f(w; \xi)\}_{(w,\xi) \in S^k}$ in subsequent iterations because $z^k$ depends on $\xi$ for all $(w, \xi) \in S^k$. Thus, it is reasonable to assume that if an SGD method employs the stochastic gradient $\nabla f(z^k; \xi^k)$ to compute $x^{k+1}$, then it has already used the gradients $\{\nabla f(w; \xi)\}_{(w,\xi) \in S^k}$ in previous iterations to fully leverage all available information. In other words, all stochastic gradients used in the computation of $z^k$ are also utilized in calculating $x^k$. This is equivalent to the second condition $\mathrm{repr}(z^k) \subseteq \mathrm{repr}(x^k)$.

**Condition 3.** This condition is arguably the most important in Theorem 2.4 because it determines the *iteration rate* of the main branch $\{x^k\}_{k \geq 0}$. In fact, *iteration rate* $\mathcal{O}\left((R+1)L\Delta/\varepsilon + \sigma^2 L\Delta/\varepsilon^2\right)$ depends on $R := \sup_{k \geq 0} \mathrm{dist}(x^k, z^k)$.

**Vanilla SGD** (Section E.1). For instance, consider the simplest method, the classical stochastic gradient descent (Vanilla SGD) method: $w^{k+1} = w^k - \gamma \nabla f(w^k; \eta^k)$, where $w^0$ is a starting point and are $\{\eta^k\}$ are i.i.d. random variables. Taking $x^k = z^k = w^k$ and $\xi^k = \eta^k$ for all $k \geq 0$. Clearly, all conditions of Theorem 2.4 are satisfied: $\xi^k$ is independent of $\{(x^{i+1}, z^{i+1}, \xi^i)\}_{i=0}^{k-1}$, $\mathrm{repr}(x^k) = \mathrm{repr}(z^k)$ for all $k \geq 0$, and $R = 0$. We get the *iteration rate* $\mathcal{O}\left(L\Delta/\varepsilon + \sigma^2 L\Delta/\varepsilon^2\right)$. The corresponding tree is in Figure 15.

Conversely, if an SGD method is overly non-conservative, leading to a large tree distance $R$ between $x^k$ and $z^k$, the *iteration rate* correspondingly increases. The further the maximum tree distance $R$ between $x^k$ and $z^k$, the more iterations are required to achieve the desired accuracy $\varepsilon$.

**Proof novelties.** In Section D.1, we outline the key novelties, challenges, and the intuition guiding our choice of conditions. Although our proof in Section D.2 is compact—which we view as a strength rather than a limitation—it unifies a broad class of methods and provides new insights. Notably, right at the beginning, we introduce a distinct approach to handling the staleness term $\|x^k - z^k\|$, which naturally arises from the update $x^{k+1} = x^k - \gamma \nabla f(z^k; \xi^k)$ in asynchronous and local methods. This treatment fundamentally differs from prior work, as it analyzes staleness through geometric graph reasoning. Moreover, using our framework, we later present our version of Local SGD, which yields tighter guarantees compared to the classical Local SGD (see Sections 3 and H), further validating both our framework and proof technique.

## 3 EXISTING AND NEW ALGORITHMS: SUMMARY AND COMPARISON

In this section, we consider examples of distributed methods. We will show that all of them can be represented by computation trees and analyzed using Theorem 2.4. The detailed analysis of each method is provided in Section E.

**Rennala SGD** (Section E.2). Consider Rennala SGD, which can be written as

$$w^{k+1} = w^k - \gamma \sum_{i=1}^{B} \nabla f(w^k; \eta^{k,i}), \tag{2}$$

where $n$ workers collaboratively calculate the batch of size $B$ (see Alg. 4). This method produces a computation tree constructed as follows: $x^1 = x^0 - \gamma \nabla f(x^0; \xi^0), \ldots, x^B = x^{B-1} - \gamma \nabla f(x^0; \xi^{B-1}), x^{B+1} = x^B - \gamma \nabla f(x^B; \xi^B), \ldots, x^{2B} = x^{2B-1} - \gamma \nabla f(x^B; \xi^{2B-1}), \ldots$, where $B$ is a batch size (see Figure 16) and $\{\xi^k\}$ are i.i.d. from $\mathcal{D}_\xi$. Notice that the computation tree is equivalent to (2) because $x^B = w^1, x^{2B} = w^2$, etc. Here, all conditions of Theorem 2.4 are satisfied for the main branch $\{x^k\}$ with the auxiliary sequence $\{(z^k, \xi^k)\}$ such that $z^0 = \cdots = z^{B-1} = x^0$, $z^B = \cdots = z^{2B-1} = x^B$, etc, and $\xi^0 = \eta^{0,0}, \ldots, \xi^{B-1} = \eta^{0,B-1}, \xi^B = \eta^{1,0}$, etc. However, unlike Vanilla SGD, $R = B - 1$ because $\text{dist}(x^0, z^0) = 0, \text{dist}(x^1, z^1) = 1, \ldots, \text{dist}(x^{B-1}, z^{B-1}) = B - 1, \text{dist}(x^B, z^B) = 0$, etc. Thus, the *iteration rate* is $\mathcal{O}\left(BL\Delta/\varepsilon + \sigma^2 L\Delta/\varepsilon^2\right)$.

**Ringmaster ASGD** (Section E.4). This an Asynchronous SGD method with the update rule

$$w^{k+1} = w^k - \gamma \nabla f(w^{k-\delta^k}; \eta_i^{k-\delta^k}), \tag{3}$$

where $\delta^k$ is a delay such that $\delta^k \leq G - 1$, where $G \geq 1$ is a hyperparameter (see Alg. 7). We take $x^k = w^k$ for all $k \geq 0$. Thus, the corresponding auxiliary sequence is defined by $z^k = x^{k-\delta^k} \equiv w^{k-\delta^k}$ and $\xi^k = \eta_i^{k-\delta^k}$ for all $k \geq 0$. Constructing the computation tree (Figure 17), we can show that the conditions of Theorem 2.4 hold with $R = \max_{k \geq 0} \delta^k \leq G - 1$ and the *iteration rate* is $\mathcal{O}\left(GL\Delta/\varepsilon + \sigma^2 L\Delta/\varepsilon^2\right)$.

Previously, we presented Rennala SGD and Ringmaster ASGD that can be analyzed using Theorem 2.4. This raises the question: Which method is most effective, and how should one choose the appropriate one? In the following sections, we discuss different factors one should consider when selecting a method, and present new algorithms. The discussion here is summarized in Table 1. Before we begin, it is important to note that the iteration complexity in Theorem 2.4 does not reflect the true wall-clock performance. It serves as an intermediate result used to derive the time complexities presented below.

**1. Computational complexity.** One way to compare the methods is to analyze their time complexity under the $h_i$-*fixed computation model* (see Sec. 1.1, A, and F). With a proper choice of the corresponding parameters, i.e., $B = \max\{1, \lceil \sigma^2/\varepsilon \rceil\}$, both Rennala SGD and Ringmaster ASGD are optimal in terms of wall-clock time with the time complexity $\Theta(\min_{m \in [n]}[(1/m \sum_{i=1}^{m} 1/h_i)^{-1} \left(L\Delta/\varepsilon + \sigma^2 L\Delta/m\varepsilon^2\right)])$ provided that communication times are negligible. In the worst-case scenario, on the "very bad function" (Arjevani et al., 2022), all these methods perform equally well. Next, we discuss the strengths and weaknesses of the methods that are not captured by the $h_i$-fixed computation model.

Table 1: **Summary of distributed optimization methods from Sections 3 and E.** In this table, we compare methods across different metrics. A ✓ indicates a favorable property in the corresponding metric. As can be seen, each method has its own advantages and disadvantages. Therefore, for any practical setup, one should choose the most suitable method based on the specific requirements. For all methods, we use the parameters from the theorems of Section F when deriving the metrics.

| Method (Sec. E) | Optimal Computational Complexity (Sec. F) | Communication Com-[g] plexity with Equal Times | Optimal Total Complexity (Sec. I) | AllReduce[a] | Update[d] Frequency | Peak[e] Bandwidth |
|---|---|---|---|---|---|---|
| Rennala SGD (Alg. 4) (Tyurin & Richtárik, 2023) | ✓ | $\tau \frac{L\Delta}{\varepsilon}$ ✓ | ✗ | ✓ | ↑ | $n$ |
| Ringmaster ASGD (Alg. 7) (Maranjyan et al., 2025) | ✓ | $\tau \left( \frac{\sigma^2 L\Delta}{n\varepsilon^2} \vee \frac{L\Delta}{\varepsilon} \right)$ | ✗ | ✗ | ↑↑↑ ✓ | $n$ |
| Local SGD (Alg. 5) **(new)**[f] | ✓ | $\tau \frac{L\Delta}{\varepsilon}$ ✓ | ✗ | ✓ | ↑↑ | $n$ |
| Cycle SGD (Alg. 8) **(new)** | ✗ | $\tau \left( \frac{\sigma^2 L\Delta}{n\varepsilon^2} \vee \frac{L\Delta}{\varepsilon} \right)$ | ✗ | — | ↑↑ | $\frac{n^2 \varepsilon}{\sigma^2} \vee 1$ ✓ |
| Async-Local/Batch SGD (Alg. 9 and Sec. E.7) **(new)** | ✓ | $\tau \frac{L\Delta}{\varepsilon}$ ✓ | — | ✗ | ↑↑ | $n$ |
| (Nested) Local-Async SGD (Alg. 12 and 14) **(new)** | ✓ | —[c] | —[c] | — | ↑↑ | —[c] |
| Dual-Process SGD (Alg. 18) **(new)** | ✓ | $\tau \frac{L\Delta}{\varepsilon}$ ✓ | ✓ | ✗ | ↑↑ | $n$ |
| Meta Local SGD[b] (Alg. 16) **(new)** | — | — | — | — | — | — |

[a] Does a method support `AllReduce`? Asynchronous SGD-like methods do not support it due to their greedy update nature. Cycle SGD synchronizes only a subset of workers; thus, we cannot say definitively. We also cannot say definitively in the case of Local-Async SGD because the local asynchronous steps cannot be implemented with `AllReduce`, while the global steps can be.
[b] This is an abstract method where all metrics (Computational Complexity, Communication Complexity, etc) depend on the chosen strategy.
[c] Similar to `AllReduce`, here we also can not say definitely since Local-Async SGD and Nested Local-Async SGD are specially designed multi-cluster learning methods.
[d] This is a slightly less formal metric that indicates how often an algorithm updates its iterate/model. Rennala SGD asks the workers to compute stochastic gradients at the same point; thus, it updates the iterates less frequently. In contrast, Ringmaster ASGD updates the iterates immediately. All other methods fall somewhere in between. See the discussion in Section 3.
[e] In the Rennala SGD, Ringmaster ASGD, Local SGD, and Async-Local SGD methods, all workers can start communication simultaneously; thus, their peak bandwidth is $\mathcal{O}(n)$ when $n \leq \sigma^2/\varepsilon$ In the Cycle SGD method, the workers communicate in a circular manner, so the peak bandwidth is $\mathcal{O}(n^2\varepsilon/\sigma^2)$ when $n \leq \sigma^2/\varepsilon$, which is smaller.
[f] While we recognize that Local SGD is well-known in the literature, what makes our version novel is the better time complexity compared to the classical version (Sec. H), the stopping criterion $\sum_{i=1}^{n} M_i = B$ in Alg. 5, and the analysis in Sec. E.3, F, and G, which leads to the optimal computational time complexities with a proper choice of $B$.
[g] We report the terms w.r.t. communication time $\tau$ under the $(h, \tau)$-*fixed computation model* from Section G.

**2. Number of model updates.** At the same time, comparing (3) and (2) reveals that Rennala SGD computes $B$ stochastic gradients at the same point, while Ringmaster ASGD both computes and "explores" more by immediately updating the model as in (3). This feature makes Ringmaster ASGD more practically appealing. In fact, this intuition can be formalized using a simple two-dimensional strongly convex quadratic function $f : \mathbb{R}^2 \to \mathbb{R}$ such that $f(x, y) = \mu x^2/2 + L y^2/2$ for all $x, y \in \mathbb{R}$. For this function, we prove that Rennala SGD requires $\widetilde{\Theta}\left(\sigma^2/\varepsilon n \times h \times L/\mu\right)$ seconds to achieve an $\varepsilon$–solution under the $h_i$-fixed computation model with $h_i = h$ for all $i \in [n]$, while Ringmaster ASGD needs $\widetilde{\Theta}\left(h \times L/\mu\right)$, which is $\sigma^2/\varepsilon n$ seconds less (see formal result in Section J). **This is the first result showing that** Ringmaster ASGD/Asynchronous SGD **can be *strongly* better than** Rennala SGD.

**3. Communication complexity.** Communication delays are a major bottleneck in real-world distributed systems. Thus, minimizing communication and synchronization overhead is crucial. Ringmaster ASGD is the least efficient in this regard, requiring frequent communication and lacking `AllReduce` support due to its asynchronous design. In a simple model where sending one stochastic gradient takes $\tau$ seconds and all workers have identical speed $h_i = h$, the time complexity of Ringmaster ASGD is $\Omega\left(\tau\left(L\Delta/\varepsilon + \sigma^2 L\Delta/n\varepsilon^2\right) + h\left(L\Delta/\varepsilon + \sigma^2 L\Delta/n\varepsilon^2\right)\right)$. In contrast, Rennala SGD achieves $\mathcal{O}\left(\tau L\Delta/\varepsilon + h\left(L\Delta/\varepsilon + \sigma^2 L\Delta/n\varepsilon^2\right)\right)$ (see Sec. G), which is better when $\tau$ and $\sigma^2/\varepsilon$ are large.

This is the point where we asked ourselves if it is possible to design a method that has the optimal computational time complexity of Rennala SGD and updates the iterates more frequently than it. It turns out this method is well-known and is called Local SGD:

**Local SGD** (Section E.3). We consider the classical Local SGD strategy, where each worker $i \in [n]$ performs $M_i$ local steps, after which the server aggregates the results (see Alg. 5). Unlike most previous approaches, however, the number of local steps $M_i$ may vary across workers. Moreover, the server waits for a specific condition before aggregating: $\sum_{i=1}^{n} M_i = B$. This strategy is adaptive to fluctuations in the number of local steps performed by individual workers, as the server ensures the total number of steps across all workers reaches the target sum $\sum_{i=1}^{n} M_i = B$, where $B$ is a hyperparameter. An example of the computation tree is shown in Figure 4. In Section E.3, we

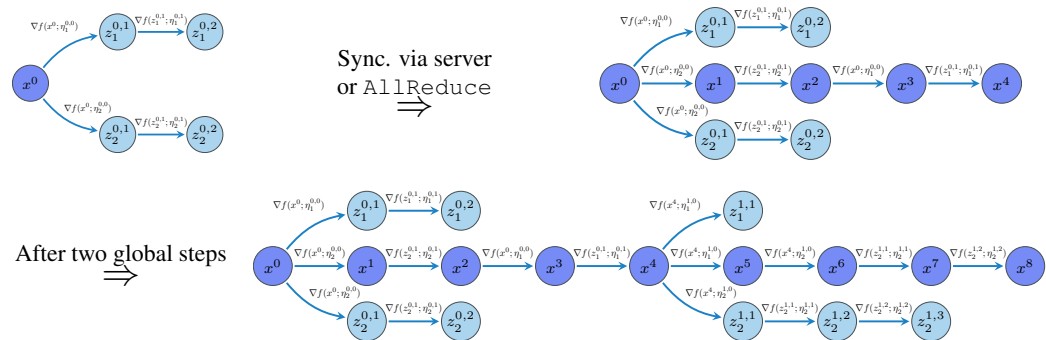

Figure 4: An evolution of a Local SGD computation tree with $B = 4$ and 2 workers, each performing local steps over 2 global steps. In the first round, they compute $M_1 = 2$ and $M_2 = 2$ local steps (first figure), after which they synchronize (second figure). In the second round, they compute $M_1 = 1$ and $M_2 = 3$ local steps and synchronize again (third figure). Note that the maximum distances $\text{dist}(x^3, z_1^{0,1})$ and $\text{dist}(x^7, z_2^{1,2})$, when applying $\nabla f(z_1^{0,1}; \eta_1^{0,1})$ to $x^3$ and $\nabla f(z_2^{1,2}; \eta_2^{1,2})$ to $x^7$, are equal to $B - 1 = \sum_{i=1}^{n} M_i - 1 = 3$. Notice that each stochastic gradient is used 2 times in the tree.

establish the *iteration rate* of Local SGD as $\mathcal{O}\left(BL\Delta/\varepsilon + \sigma^2 L\Delta/\varepsilon^2\right)$. This result follows directly from Theorem 2.4 via a simple geometric argument. In fact, looking at Figure 4 reveals that all the conditions of Theorem 2.4 are satisfied. The only minor difficulty is to show that $R := \sup_{k \geq 0} \text{dist}(x^k, z^k) \leq B - 1$, which is guaranteed by the condition $\sum_{i=1}^{n} M_i = B$.

What is novel in our version of Local SGD is that it achieves better theoretical guarantees within the family of Local SGD approaches (see Section H). Moreover, our stopping condition and the choice of $B$ together ensure its optimality under the $h_i$-fixed computation model (see Theorem F.6).

**Async-Local SGD** (Section E.6). Another idea to leverage the practical benefits of Ringmaster ASGD, while at the same time reducing the communication overhead, is to use Ringmaster ASGD with local steps. The idea is to run $M$ local steps on each worker instead of immediately sending the computed stochastic gradients to the server in an asynchronous fashion (See Figure 5). We formalize this algorithm and prove the iteration rate in Section E.6. Moreover, in Sections F and G, we suggest an optimal choice of parameters that leads to optimal computational complexity and reduced communication complexity, which is better than that of Ringmaster ASGD. We get as similar result with a new method, Async-Batch SGD (Section E.7).

**Dual-Process SGD** (Section E.11). We took a step further and developed a new local method inspired by Local SGD and Async-Local SGD. It is the first local method to achieve the optimal time complexity in the distributed setting, where workers have varying computation and communication times (see Section I). However, unlike Local SGD, it is not AllReduce-friendly.

**4. Peak bandwidth.** Another critical factor is the peak bandwidth. The number of workers the parameter-server or the AllReduce operation can synchronize may be limited when the number of workers $n$ is huge. Notice that the worst-case peak bandwidth of Rennala SGD, Ringmaster ASGD, Local SGD, and Async-Local SGD is $\Theta(n)$.

**Cycle SGD** (Section E.5). To mitigate this issue, we propose a new method called Cycle SGD. Similar to Local SGD, each worker performs local steps. However, once the workers finish computing the initial stochastic gradients $\{\nabla f(z_i^0; \eta_i^0)\}$, only the first group of $s$ workers sends their gradients to the server, where $s$ is a hyperparameter. The server then aggregates these gradients and performs the update $w^1 = w^0 - \gamma \sum_{i=1}^{s} \nabla f(z_i^0; \eta_i^0)$. Meanwhile, the first $s$ workers begin computing their local steps starting from $w^1$, while the remaining workers continue their current local computations. Next, the second group of $s$ workers sends their locally computed vectors, and this process continues in a circular manner. A computation tree presented in Figure 18. The peak bandwidth of Cycle SGD is $\mathcal{O}(s)$ with $s = \min\left\{\max\{\lceil n^2\varepsilon/\sigma^2\rceil, 1\}, n\right\}$, which is smaller than $\Theta(n)$ when $\sigma^2/\varepsilon \geq n$.

**5. Optimization with clusters.** Consider a setup with many clusters of workers, where intra-cluster communication (e.g., InfiniBand) is fast and inter-cluster communication (e.g., Ethernet) is slow.

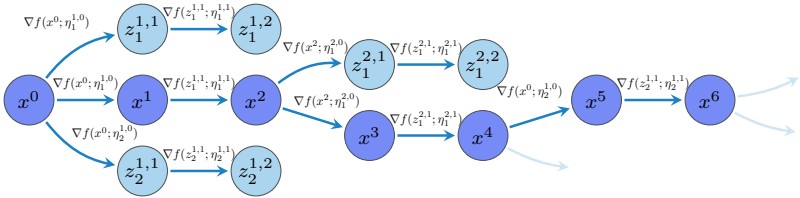

Figure 5: An example of the computation tree for Async-Local SGD with $M = 2$. In this example, the first worker is significantly faster: before the second worker completes its first set of local steps, $x^0 \to z_2^{1,1} \to z_2^{1,2}$, the first worker already completes two rounds of local updates and sends the corresponding stochastic gradients, $(\nabla f(x^0; \eta_1^{1,0}), \nabla f(z_1^{1,1}; \eta_1^{1,1}))$ and $(\nabla f(x^2; \eta_1^{2,0}), \nabla f(z_1^{2,1}; \eta_1^{2,1}))$.

**Local-Async SGD** (Section E.8) We run Asynchronous SGD within each cluster and synchronize clusters after a fixed number of local steps. This setup is feasible due to fast intra-cluster links, while slower inter-cluster links necessitate infrequent synchronization. In Section E.8, we formalize this method, Local-Async SGD, and establish its iteration rate. Section F proves it achieves optimal computational time complexity. A key novelty lies in the synchronization mechanism (see Alg. 12).

**Nested Local-Async SGD** (Section E.9) Our framework extends to a two-level hierarchy: within each cluster, servers with 4–8 GPUs run Asynchronous SGD locally, synchronize at the server level, and then synchronize across clusters. Analyzing such a setup using classical optimization tools would be highly challenging. In contrast, our framework enables a straightforward analysis through geometric graph reasoning.

**6. Flexible synchronization and Meta Local SGD** (Section E.10). We noticed that in all previous methods, the workers are synchronized in a predefined manner or rule. We want to add more flexibility to the synchronization process. Our idea is that the server (or the workers themselves, in a decentralized setup) can select any subset of workers based on any strategy (e.g., randomly or according to current communication speeds), gather their computed stochastic gradients, update the global model, and ask these workers to continue performing local steps from the new point. However, such "anarchic synchronization" can result in a computation tree with a large $R$ if the selected strategy is not chosen carefully. To ensure that $R$ is bounded, in our meta-algorithm (Algorithm 16), we track the current distances $\{d_i\}$ to the head of the main branch and the local steps $\{M_i\}$ performed by each worker. Then, by tracking the value $d_i + \sum_{i=1}^{n} M_i$ for all $i \in [n]$ and comparing it to a parameter $B$, we compulsorily synchronize (Hard Sync) all workers for which $d_i + \sum_{i=1}^{n} M_i = B$. This way, we can ensure that $R$ is bounded by $B$, and the iteration rate of this method is $\mathcal{O}\left(BL\Delta/\varepsilon + \sigma^2 L\Delta/\varepsilon^2\right)$.

## 4 INSIGHTS AND GUIDELINES

All proposed methods share the same iteration rate of $\mathcal{O}\left((R+1)L\Delta/\varepsilon + \sigma^2 L\Delta/\varepsilon^2\right)$, where $R$ is controlled by a method-specific hyperparameter and, at the same time, $R$ is the largest tree distance between $x^k$ and $z^k$. For Rennala SGD, $R = B - 1$, where $B$ denotes the batch size; for Ringmaster ASGD, $R = B-1$, where $B$ is the delay threshold; for Local SGD, $R = B-1$, where $B$ corresponds to the number of local steps; for Cycle SGD, $R = n^2/s$, where $s$ is the group size, etc. In all these methods, $R$ can be controlled, and to achieve the best possible computational and communication guarantees, one should always choose $R = \Theta\left(\sigma^2/\varepsilon\right)$ (see Sections G and F). We believe this is a fundamental principle underlying all parallel optimization methods, and it should be considered a guiding rule when developing new algorithms. This choice is also theoretically justified: by taking $R = \Theta\left(\sigma^2/\varepsilon\right)$, the iteration rate does not change asymptotically: $\mathcal{O}\left((R+1)L\Delta/\varepsilon + \sigma^2 L\Delta/\varepsilon^2\right) = \mathcal{O}\left(L\Delta/\varepsilon + \sigma^2 L\Delta/\varepsilon^2\right)$. Larger values of $R$ allow the methods to be more "parallel-friendly". For instance, a large $R$ enables Ringmaster ASGD to consider stochastic gradients with larger delays, while a large $R$ in Local SGD allows the method to run more local steps. However, taking $R > \sigma^2/\varepsilon$ results in a worse iteration rate, suggesting that the corresponding method operates in an overly "anarchic" asynchronous regime, which may lead to performance degradation. Geometrically, the theory suggests that, to achieve good performance, the tree distance between $x^k$ and $z^k$ in Figure 3 should not exceed $\sigma^2/\varepsilon$.

Notice that there is no single "best" method in Table 1, which we believe is another fundamental law. Each method has its own strengths and weaknesses, and one should develop or choose the most appropriate method for the specific task. This process becomes easier with the help of our new framework, Birch SGD, and insights.

While our primary focus is on training large-scale language and vision models, where the i.i.d. assumption is usually appropriate, we acknowledge that non-i.i.d. scenarios are also important and should be investigated in future work, where it may be necessary to add additional assumptions such as first-order and second-order similarity of the functions (Arjevani & Shamir, 2015; Mishchenko et al., 2022). Moreover, it would be interesting to extend our framework to methods with preconditioning (Kingma & Ba, 2015) and non-Euclidean geometry (Jordan et al.).

We hope our observations will support the future development and analysis of asynchronous optimization methods. Building on these insights, we designed at least eight new methods using our proposed Birch SGD framework and the main result, Theorem 2.4. By reducing the analysis and design of these methods to computation trees, our entire development becomes purely graph-geometrical, offering a new and simpler view on asynchronous optimization methods.

## 5 SUMMARY OF EXPERIMENTAL RESULTS

In Section C, we provide a detailed comparison of methods on logistic regression, image classification with ResNet18 (He et al., 2016), and next-token prediction with GPT2 (Radford et al., 2019). When communication times are negligible (Fig. 6), as expected from Table 1 and the previous discussion, Ringmaster ASGD and Async-Local SGD converge faster on the logistic regression problem. However, when communication times are large (Fig. 7), Ringmaster ASGD becomes less practical due to its frequent updates. Synchronized SGD exhibits the worst performance across all setups. Rennala SGD and Local SGD are more stable, while Async-Local SGD performs well due to its effective balance between frequent updates and local steps.

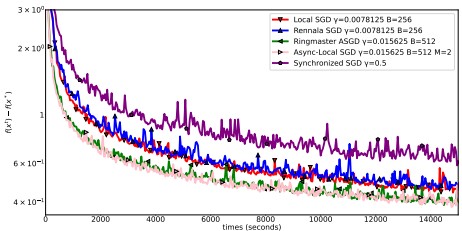

Figure 6: Computation times $h_i = 1$ or 10 randomly, communication times $\tau_i = 0$.

Figure 7: Computation times $h_i = 10$, communication times $\tau_i = 100$.

## ACKNOWLEDGEMENTS

The work was supported by the grant for research centers in the field of AI provided by the Ministry of Economic Development of the Russian Federation in accordance with the agreement 000000C313925P4F0002 and the agreement №139-10-2025-033.

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

## CONTENTS

# A    ADDITIONAL DISCUSSION

## A.1    DISCUSSION OF THE COMPUTATIONAL TIME COMPLEXITIES

In this section, we extend our discussion about the computational time complexities of the methods discussed in the main part of the paper.

To compare parallel and asynchronous methods, Mishchenko et al. (2022) proposed using the $h_i$-*fixed computation model*. The idea is to assume that worker $i$ requires at most $h_i$ seconds to calculate one stochastic gradient for all $i \in [n] := \{1, \ldots, n\}$ (without loss of generality, $h_1 \leq h_2 \leq \cdots \leq h_n$). The authors considered Synchronized SGD, an iterative process defined as $w^{k+1} = w^k - \frac{\gamma}{n} \sum_{i=1}^n \nabla f(w^k; \eta_i^k)$, where each worker calculates one stochastic gradient, synchronize, and a parameter server aggregates them to update the iterate[4]. Using the $h_i$-fixed computation model, it can be easily shown that Synchronized SGD converges after

$$\mathcal{O}\left(\max_{i \in [n]} h_i \times \left(\frac{L\Delta}{\varepsilon} + \frac{\sigma^2 L\Delta}{n\varepsilon^2}\right)\right) \tag{4}$$

seconds, because the method waits for the slowest worker, whose time is $\max_{i \in [n]} h_i = h_n$.

---

**Algorithm 2** Asynchronous SGD

---

**Input:** point $w^0 \in \mathbb{R}^d$, stepsizes $\gamma_k \geq 0$
Workers start computing stochastic gradients at $w^0$
**for** $k = 0, 1, \ldots$ **do**
    Gradient $\nabla f(w^{k-\delta^k}; \eta_i^{k-\delta^k})$ arrives from worker $i$
    Update: $w^{k+1} = w^k - \gamma_k \nabla f(w^{k-\delta^k}; \eta_i^{k-\delta^k})$
    Worker $i$ begins calculating at $w^{k+1}$
**end for**

---

Mishchenko et al. (2022); Koloskova et al. (2022) provided new analyses of Asynchronous SGD (see Algorithm 2) and Cohen et al. (2021) developed Picky SGD to show that this time complexity can be improved to

$$\mathcal{O}\left(\left(\frac{1}{n}\sum_{i=1}^n \frac{1}{h_i}\right)^{-1}\left(\frac{L\Delta}{\varepsilon} + \frac{\sigma^2 L\Delta}{n\varepsilon^2}\right)\right),$$

where the dependence on $\{h_i\}$ is harmonic instead of being based on the maximum. It turns out that this complexity can be further improved[5] to

$$\Theta\left(\min_{m \in [n]}\left[\left(\frac{1}{m}\sum_{i=1}^m \frac{1}{h_i}\right)^{-1}\left(\frac{L\Delta}{\varepsilon} + \frac{\sigma^2 L\Delta}{m\varepsilon^2}\right)\right]\right), \tag{5}$$

which is achieved by the Rennala SGD method (Tyurin & Richtárik, 2023). Moreover, Tyurin & Richtárik (2023) proved a matching lower bound demonstrating that both this complexity and Rennala SGD are optimal. Recently, Maranjyan et al. (2025) developed a new optimal Ringmaster ASGD method, which is essentially Asynchronous SGD with a key modification. Additionally, under the *universal computation model*, Tyurin (2025); Maranjyan et al. (2025) showed that both Rennala SGD and Ringmaster ASGD remain optimal even when computation times are arbitrary, time-varying, and random.

## A.2    MORE RELATED WORK

Our focus is on the homogeneous setting, where all workers have access to the same data distribution or dataset. The heterogeneous data setting is equally important, especially in federated learning (FL) (Konečný et al., 2016) due to privacy constraints. In this context, many other methods have been

---

[4]Alternatively, there is no physical parameter server, and all workers perform an Allreduce.

[5]Note that $\min_{m \in [n]} g(m) \leq g(n)$ for any function $g : \mathbb{N} \to \mathbb{R}$

proposed, including Asynchronous SGD (Mishchenko et al., 2022; Koloskova et al., 2022), Asgrad (Islamov et al., 2024), PIAG (Wu et al., 2022), and Malenia SGD (Tyurin & Richtárik, 2023). Notably, Tyurin & Richtárik (2023); Tyurin (2025) showed that Malenia SGD is optimal under both the fixed and universal computation models, without requiring assumptions of bounded gradients or gradients dissimilarity.

In the homogeneous setting, numerous other works have studied asynchronous SGD methods, including (Lian et al., 2015; Feyzmahdavian et al., 2016; Stich & Karimireddy, 2020; Sra et al., 2016). However, these methods typically require the assumption that the delays in the indices of stochastic gradients are bounded (on average in (Sra et al., 2016)). As a result, their theoretical guarantees in terms of computational time complexity are weaker than those in (Cohen et al., 2021; Koloskova et al., 2022; Mishchenko et al., 2022; Tyurin & Richtárik, 2023; Maranjyan et al., 2025), which do not rely on such assumptions.

### A.3    RELATION TO OTHER FRAMEWORKS

There were several previous approaches to unify SGD methods. Gorbunov et al. (2021) proposed using a parametric assumption to unify the analysis of local methods, Wang et al. (2020) unified FedAvg-like methods, Wang & Joshi (2018) proposed Cooperative SGD to analyze different synchronization mechanisms through mixing matrices, Huang et al. (2022) analyzed a general sample-wise Push–Pull framework in the heterogeneous setting using two-level augmented graphs, and Khaled et al. (2020); Gorbunov et al. (2020) analyzed SGD methods with variance-reduction and compression techniques. These approaches are related to, but not directly comparable with ours, and, to the best of our knowledge, our approach is new and orthogonal.

Another interesting work that analyzes SGD methods is (Even et al., 2024). Their work and ours both use graphs; however, we use graphs in completely different, orthogonal, and unrelated contexts. In their case, nodes represent computers (GPUs, CPUs, servers), and edges represent communication links. In our case, nodes represent points of an algorithm, (directed) edges indicate how one point is calculated from another, and the graphs evolve with every iteration. Similarly, Huang et al. (2022) also use a graph abstraction but with a different meaning for nodes and edges: in their case, nodes correspond to devices and iterates of data samples, edges represent both communication and computation links, and the number of nodes is fixed from the beginning since every node corresponds to one sample or worker. These are different and orthogonal approaches. Another important difference is that they compare methods using iteration complexities, whereas we use time complexities in Table 1, which is a more robust and suitable metric for asynchronous and parallel methods.

## B    NOTATIONS

$\mathbb{N} := \{1, 2, \dots\}$; $\|x\|$ is the output of the standard Euclidean norm for all $x \in \mathbb{R}^d$; $\langle x, y \rangle = \sum_{i=1}^{d} x_i y_i$ is the standard dot product; $g = \mathcal{O}(f)$ : exist $C > 0$ such that $g(z) \leq C \times f(z)$ for all $z \in \mathcal{Z}$; $g = \Omega(f)$ : exist $C > 0$ such that $g(z) \geq C \times f(z)$ for all $z \in \mathcal{Z}$; $g = \Theta(f)$ : $g = \mathcal{O}(f)$ and $g = \Omega(f)$; $g = \widetilde{\Theta}(f)$ : the same as $g = \Omega(f)$ but up to logarithmic factors; $a \vee b := \max\{a, b\}$.

## C    EXPERIMENTS

### C.1    SETUP

The experiments were prepared in Python. The distributed environment was simulated with the Simpy Python library (Matloff, 2008). There are two hardware setups:

● CPU Setup: Intel(R) Xeon(R) Gold 6348 CPU @ 2.60GHz 52 cores (for logistic regression experiments)

● GPU Setup: 2 × Nvidia A100 80 Gb, CPU: Intel(R) Xeon(R) Platinum 8358 CPU @ 2.60GHz 128 cores (for ResNet18 and GPT2 experiments)

The distributed environment is simulated with the help of Simpy. To compare the methods, we consider different computation and communication scenarios by taking different computation times $\{h_i\}$ and communication times $\{\tau_i\}$ of the workers.

For each task, we perform a grid search to identify the best parameters and report the top results across all runs of each algorithm. The individual grid search parameters are drawn from a set of values specified in Section C.7. We plot the convergence rates against the elapsed time.

We evaluate the convergence speeds of all algorithms in four regimes:

• *Classical*: $h_i = 10$ and $\tau_i = 0$ for all $i \in [n]$. All workers have the same computation times, and the communication times are ignored.

• *Slow Communications*: $h_i = 10$ and $\tau_i = 100$ for all $i \in [n]$. The communication takes time.

• *Heterogeneous Computations*: $h_i = \text{random\_choice}(\{1, 10\})$ and $\tau_i = 0$ for all $i \in [n]$. All workers have the different computation times randomly sampled from the set $\{1, 10\}$.

• *Heterogeneous Communications*: $h_i = 10$ and $\tau_i = \text{random\_choice}([1,100])$ for all $i \in [n]$. All workers have the different communication times randomly sampled from the set $\{1, 10\}$.

This setup allows us to observe how different algorithms perform across various regimes and to compare their convergence behaviors under differing computational and communication conditions.

## C.2 EXPERIMENTS WITH LOGISTIC REGRESSION

We begin our experiments with one the simplest ML problems—logistic regression on the MNIST dataset LeCun et al. (2010). In this setting, we evaluate three different numbers of workers: $n \in \{16, 64, 256\}$. We use the standard linear model with the logistic loss.

Starting with $n = 16$ workers, we perform a grid search over the parameters specified in Table 2 across all four regimes. The corresponding results are shown in Figure 8. In the *classical* setup (Figure 8a), all algorithms perform similarly. However, Rennala SGD and Local SGD underperform slightly due to the inability to interrupt an already initiated local step, resulting in occasional update losses. In the *slow communications* setup (Figure 8b), Rennala SGD, Local SGD, and Async-Local SGD perform better, as they aggregate local steps and reduce communication overhead. In contrast, Synchronized SGD and Ringmaster ASGD perform poorly due to excessive communication. In both the *heterogeneous computations* (Figure 8c) and *heterogeneous communications* (Figure 8d) regimes, Async-Local SGD and Ringmaster ASGD achieve the fastest performance. Synchronized SGD, as expected, is the slowest because it is not robust to heterogeneous computations and communications.

For $n = 64$ (grid search parameters in Table 3, results shown in Figure 9) and $n = 256$ (grid search parameters in Table 4, results shown in Figure 10), we observe behavior similar to the $n = 16$ case across all four regimes.

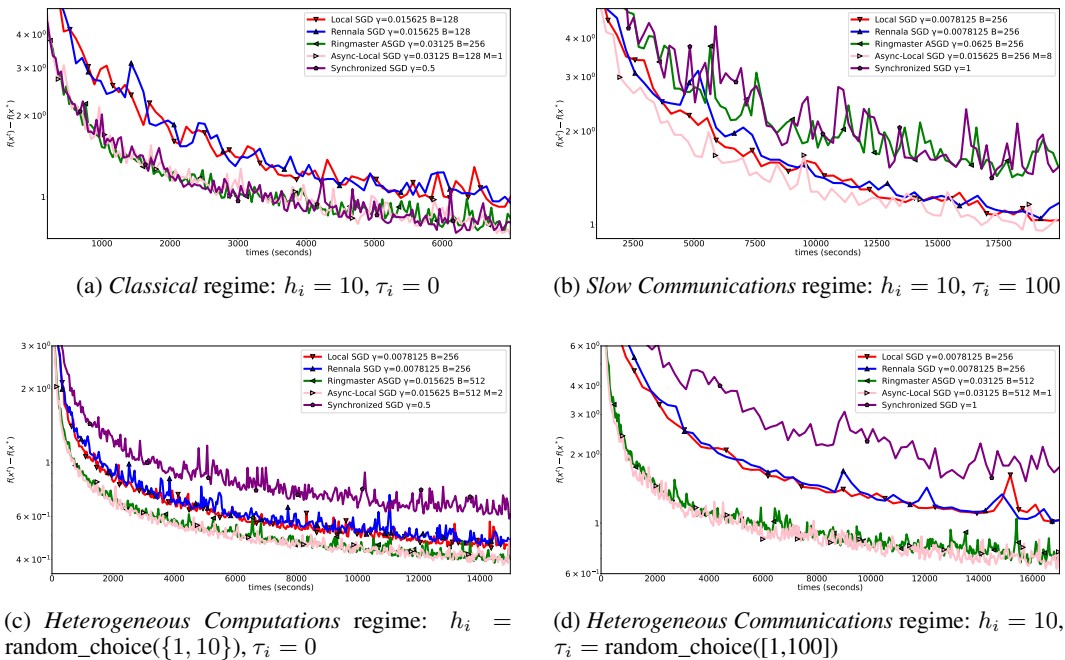

(a) *Classical* regime: $h_i = 10, \tau_i = 0$

(b) *Slow Communications* regime: $h_i = 10, \tau_i = 100$

(c) *Heterogeneous Computations* regime: $h_i = $ random_choice($\{1, 10\}$), $\tau_i = 0$

(d) *Heterogeneous Communications* regime: $h_i = 10$, $\tau_i = $ random_choice([1,100])

Figure 8: Comparison of different optimization algorithms across various distributed computing regimes with $n = 16$. Each plot shows the convergence behavior in terms of loss versus simulated wall-clock time.

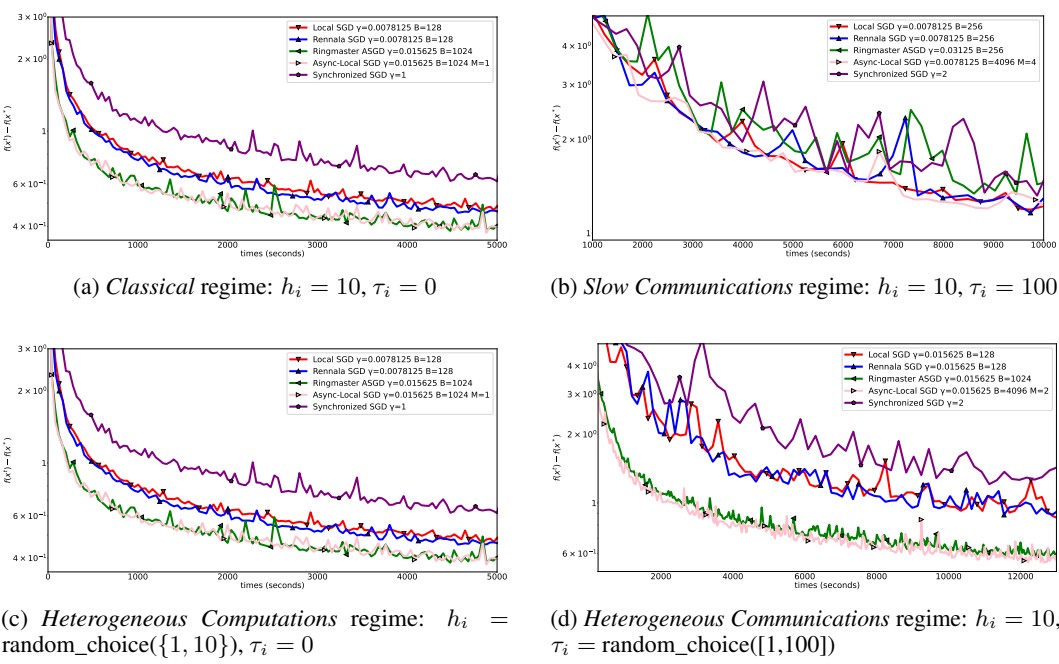

(a) *Classical* regime: $h_i = 10, \tau_i = 0$

(b) *Slow Communications* regime: $h_i = 10, \tau_i = 100$

(c) *Heterogeneous Computations* regime: $h_i = $ random_choice($\{1, 10\}$), $\tau_i = 0$

(d) *Heterogeneous Communications* regime: $h_i = 10$, $\tau_i = $ random_choice([1,100])

Figure 9: Comparison of different optimization algorithms across various distributed computing regimes with $n = 64$. Each plot shows the convergence behavior in terms of loss versus simulated wall-clock time.

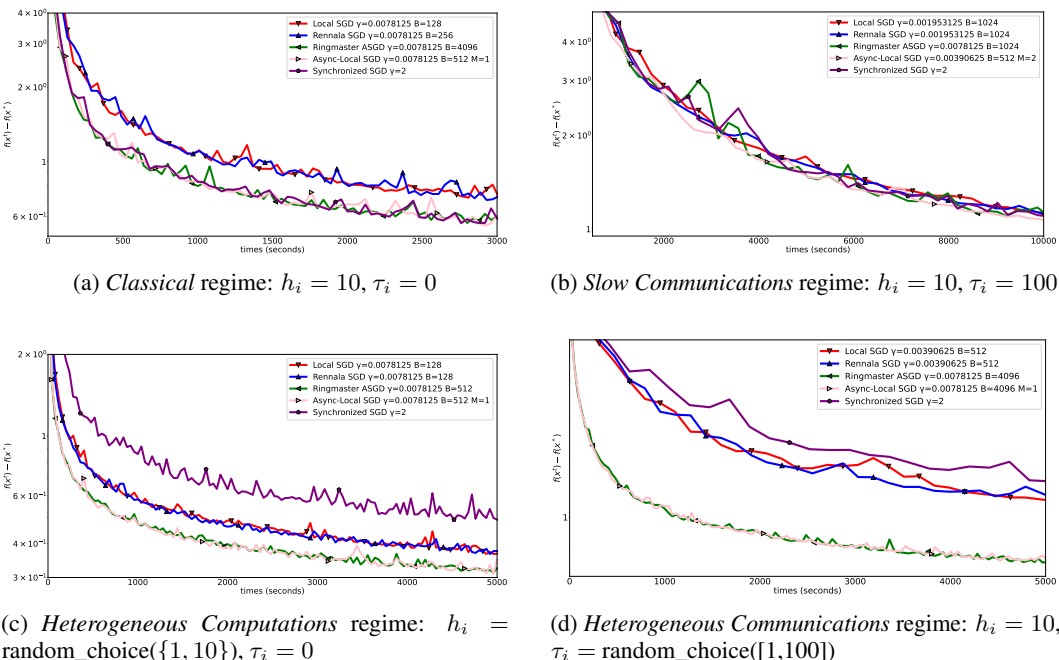

(a) *Classical* regime: $h_i = 10, \tau_i = 0$

(b) *Slow Communications* regime: $h_i = 10, \tau_i = 100$

(c) *Heterogeneous Computations* regime: $h_i =$ random_choice($\{1, 10\}$), $\tau_i = 0$

(d) *Heterogeneous Communications* regime: $h_i = 10$, $\tau_i =$ random_choice([1,100])

Figure 10: Comparison of different optimization algorithms across various distributed computing regimes with $n = 256$. Each plot shows the convergence behavior in terms of loss versus simulated wall-clock time.

## C.3 EXPERIMENTS WITH RESNET18 AND IMAGE CLASSIFICATION

We test the algorithms on the CIFAR10 (Krizhevsky et al., 2009) image recognition task with the ResNet18 (He et al., 2016) deep neural network. For ResNet18, we similarly report the best convergence results from a grid search over the parameters listed in Table 5, using a setup with $n = 8$ workers. Results for all algorithms across the four regimes are presented in Figure 11.

The conclusions largely mirror those from the logistic regression experiments, with a few additional observations. In the *classical* setup (Figure 11a), Ringmaster ASGD outperforms all other methods. We believe that this is due to the frequent updates of the method. In the *slow communications* regime (Figure 11b), the trends are consistent with those observed in the MNIST experiments: Ringmaster ASGD becomes slower, while methods that are less communication-intensive achieve better performance. In the *heterogeneous communications* (Figure 11d) regime, Async-Local SGD has the best performance due to the good balance of frequent model updates and local steps.

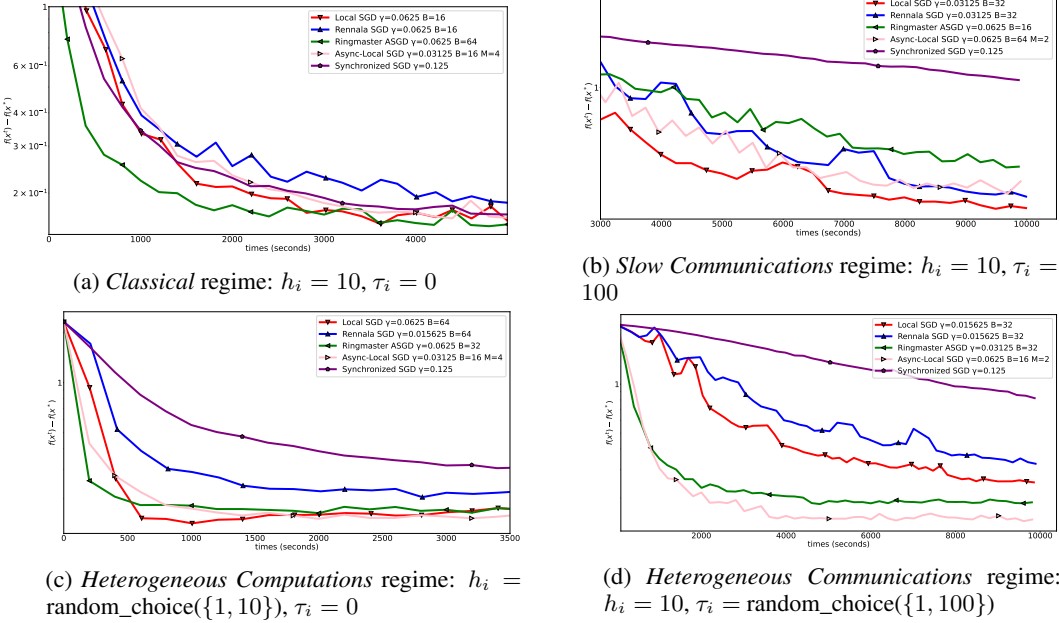

(a) *Classical* regime: $h_i = 10, \tau_i = 0$

(b) *Slow Communications* regime: $h_i = 10, \tau_i = 100$

(c) *Heterogeneous Computations* regime: $h_i = \text{random\_choice}(\{1, 10\}), \tau_i = 0$

(d) *Heterogeneous Communications* regime: $h_i = 10, \tau_i = \text{random\_choice}(\{1, 100\})$

Figure 11: ResNet18 experiments with $n = 8$

## C.4 EXPERIMENTS WITH GPT2 AND TOKEN PREDICTION

We also evaluate the algorithms on the Wikitext-2 (Merity et al., 2016) next token prediction task with GPT2 (Radford et al., 2019). For GPT2, we evaluate all four regimes using a setup with $n = 8$ workers. To achieve faster and more robust convergence, we use the AdamW normalization strategy[6] only in these experiments with GPT2. The resulting convergence curves are shown in Figure 12. Once again, the results are similar to those of the previous experiments. Due to hardware limitations, a narrower grid search range is used; therefore, it is possible that convergence could be further improved with a more extensive search.

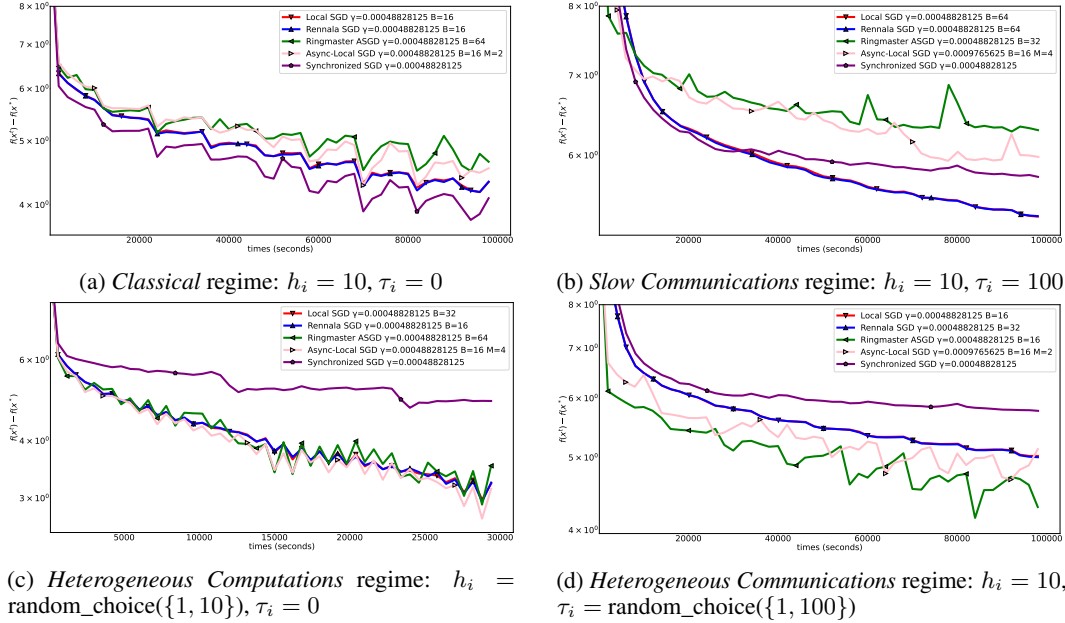

(a) *Classical* regime: $h_i = 10, \tau_i = 0$

(b) *Slow Communications* regime: $h_i = 10, \tau_i = 100$

(c) *Heterogeneous Computations* regime: $h_i = \text{random\_choice}(\{1, 10\}), \tau_i = 0$

(d) *Heterogeneous Communications* regime: $h_i = 10, \tau_i = \text{random\_choice}(\{1, 100\})$

Figure 12: GPT-2 experiments with $n = 8$

## C.5 EXPERIMENTS WITH Cycle SGD AND PEAK BANDWIDTH

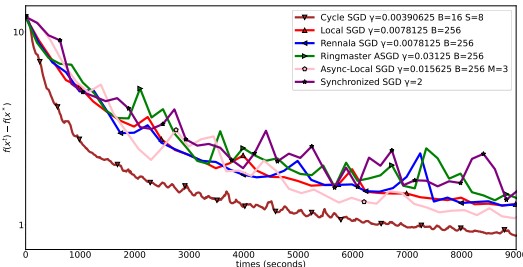

Figure 13: Comparison of methods in the setup, where the communication time depends on the number of synchronized workers. It takes $h_i = 10$ seconds to compute a stochastic gradient, and the communication time is $1.5 \times k$ seconds, where $k$ is the number of synchronized workers.

In Figure 13, we present the comparison of Cycle SGD with other methods to verify the advantageous theoretical property of Cycle SGD. For this particular regime, when the computational times are the same, all methods except Cycle SGD require all the $64$ workers to synchronize at the same time, leading to high peak bandwidth. At the same time, Cycle SGD synchronizes only $8$ workers and

---

[6]Instead of the SGD step $w^{k+1} = w^k - \gamma g^k$, where $g^k$ is a descent direction, we use the AdamW strategy with $g^k$.

reduces the communication time and the total time complexity, which we observe in Figure 13 with logistic regression on the MNIST dataset.

## C.6 SENSITIVITY TO THE CHOICE OF $B$ IN Rennala SGD

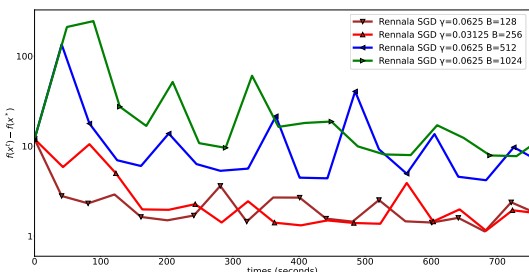

Figure 14: *Slow Communications* regime: $h_i = 10$, $\tau_i = 100$

In this section, we aim to understand the sensitivity of the choice of the value $B$ in Rennala SGD. From the proof of Theorem E.2, we know that $R := \sup_{k \geq 0} \mathrm{dist}(x^k, z^k) = \Theta(B)$. By increasing $B = \Theta(R)$, we can examine whether the method starts to slow down. This slowdown is expected due to the discussion in Section 4. In Figure 14, we present our results on logistic regression problem on the MNIST dataset and $n = 100$ workers, where we see that if we take $B$ too large, the method slows down accordingly.

## C.7 Parameters of the experiments

Table 2: Experimental configuration for logistic regression on MNIST with $n = 16$ workers.

| Parameter | Value |
|---|---|
| Batch size | 1 |
| Optimizer | SGD |
| Number of workers | 16 |
| **Algorithm-specific configurations:** | |
| **Synchronized SGD** | $\gamma$ range: $[2^{-5}, 2^4]$ |
| **Rennala SGD** | $\gamma$ range: $[2^{-15}, 2^{-3}]$ 
 B set: $\{128, 256, 512, 1024\}$ |
| **Local SGD** | $\gamma$ range: $[2^{-15}, 2^{-3}]$ 
 B set: $\{128, 256, 512, 1024\}$ |
| **Ringmaster ASGD** | $\gamma$ range: $[2^{-15}, 2^1]$ 
 B set: $\{128, 256, 512, 1024\}$ |
| **Async-Local SGD** | $\gamma$ range: $[2^{-10}, 2^1]$ 
 B set: $\{64, 128, 256, 512, 1024\}$ 
 M set: $\{1, 2, 4, 8\}$ |

Table 3: Experimental configuration for logistic regression on MNIST with $n = 64$ workers.

| Parameter | Value |
|---|---|
| Batch size | 1 |
| Optimizer | SGD |
| Number of workers | 64 |
| **Algorithm-specific configurations:** | |
| **Synchronized SGD** | $\gamma$ range: $[2^{-5}, 2^4]$ |
| **Rennala SGD** | $\gamma$ range: $[2^{-15}, 2^{-3}]$ 
 B set: $\{1024, 2048, 4096\}$ |
| **Local SGD** | $\gamma$ range: $[2^{-15}, 2^{-3}]$ 
 B set: $\{1024, 2048, 4096\}$ |
| **Ringmaster ASGD** | $\gamma$ range: $[2^{-15}, 2^1]$ 
 B set: $\{512, 1024, 2048, 4096\}$ |
| **Async-Local SGD** | $\gamma$ range: $[2^{-10}, 2^1]$ 
 B set: $\{64, 128, 256, 512, 1024, 4096\}$ 
 M set: $\{1, 2, 4, 8\}$ |

Table 4: Experimental configuration for logistic regression on MNIST with $n = 256$ workers.

| Parameter | Value |
|---|---|
| Batch size | 1 |
| Optimizer | SGD |
| Number of workers | 256 |
| **Algorithm-specific configurations:** | |
| **Synchronized SGD** | $\gamma$ range: $[2^{-5}, 2^4]$ |
| **Rennala SGD** | $\gamma$ range: $[2^{-15}, 2^{-3}]$
B set: $\{1024, 2048, 4096, 8192, 16384\}$ |
| **Local SGD** | $\gamma$ range: $[2^{-15}, 2^{-3}]$
B set: $\{1024, 2048, 4096, 8192, 16384\}$ |
| **Ringmaster ASGD** | $\gamma$ range: $[2^{-15}, 2^1]$
B set: $\{512, 1024, 2048, 4096, 8192\}$ |
| **Async-Local SGD** | $\gamma$ range: $[2^{-10}, 2^1]$
B set: $\{128, 256, 512, 1024, 4096, 8192\}$
M set: $\{1, 2, 4, 8\}$ |

Table 5: Experimental configuration for ResNet18 with $n = 8$ workers.

| Parameter | Value |
|---|---|
| Batch size | 16 |
| Optimizer | SGD |
| Number of workers | 8 |
| **Algorithm-specific configurations:** | |
| **Synchronized SGD** | $\gamma$ range: $[2^{-6}, 2^{-3}]$ |
| **Rennala SGD** | $\gamma$ range: $[2^{-6}, 2^{-3}]$
B set: $\{16, 32, 64\}$ |
| **Local SGD** | $\gamma$ range: $[2^{-6}, 2^{-3}]$
B set: $\{16, 32, 64\}$ |
| **Ringmaster ASGD** | $\gamma$ range: $[2^{-6}, 2^{-3}]$
B set: $\{16, 32, 64\}$ |
| **Async-Local SGD** | $\gamma$ range: $[2^{-6}, 2^{-3}]$
B set: $\{16, 32, 64\}$
M set: $\{2, 4, 8\}$ |

Table 6: Experimental configuration for GPT-2 with $n = 8$ workers.

| Parameter | Value |
|---|---|
| Batch size | 32 |
| Optimizer | AdamW |
| Number of workers | 8 |
| **Algorithm-specific configurations:** | |
| **Synchronized SGD** | $\gamma$ range: $[2^{-11}, 2^{-10}]$ |
| **Rennala SGD** | $\gamma$ range: $[2^{-11}, 2^{-10}]$
B set: $\{16, 32, 64\}$ |
| **Local SGD** | $\gamma$ range: $[2^{-11}, 2^{-10}]$
B set: $\{16, 32, 64\}$ |
| **Ringmaster ASGD** | $\gamma$ range: $[2^{-11}, 2^{-10}]$
B set: $\{16, 32, 64\}$ |
| **Async-Local SGD** | $\gamma$ range: $[2^{-11}, 2^{-10}]$
B set: $\{16, 32, 64\}$
M set: $\{2, 4\}$ |

# D    PROOF OF THEOREM 2.4

## D.1    PROOF TECHNIQUE AND REASONS FOR CHOOSING THE CONDITIONS

Before we state the main theorem and provide the proof, let us explain the intuition, the novelty, and how we identified the conditions of the theorem. The proof of the result is given in Section D.2 and is relatively compact. We believe that the simplicity of our result, together with its ability to unify methods, constitutes an important contribution to the optimization community. While the initial part of the proof follows the same structure as in most related works, starting from (7), our treatment of the staleness term $\|x^k - z^k\|$, which naturally arises from the step $x^{k+1} = x^k - \gamma \nabla f(z^k; \xi^k)$, is novel.

After many attempts to develop a universal theory, let us illustrate how we arrived at our conditions. Looking at Figure 3, which provides all possible relations between $x^k$ and $z^k$, one can easily get

$$\left\| x^k - z^k \right\| = \gamma \left\| \sum_{i=p}^{k-1} \nabla f(z^i; \xi^i) - \sum_{(w,\xi) \in S^k} \nabla f(w; \xi) \right\|.$$

First, we noticed that any reasonable method should utilize $\sum_{(w,\xi) \in S^k} \nabla f(w; \xi)$ in the computation of $z^k$ before applying $\nabla f(z^k; \xi^k)$ (see the previous discussion about Condition 2 in Section 2.1). This implies $\{(w; \xi)\}_{(w,\xi) \in S^k} \subseteq \{(z^i; \xi^i)\}_{i=p}^{k-1}$, leading to the following *identity*:

$$\left\| x^k - z^k \right\| = \gamma \left\| \sum_{j \in \bar{S}^k} \nabla f(z^j; \xi^j) \right\|$$

for some set $\bar{S}^k \subseteq \{p, \ldots, k-1\}$ such that $\bar{S}^k \cup S^k = \{p, \ldots, k-1\}$. The *identity* says that the distance is roughly proportional to the number $|\bar{S}^k|$ of stochastic gradients applied after $x^p$ and before $z^k$, which is *tightly* bounded by the tree distance from $x^k$ to the common ancestor $x^p$, i.e., it is bounded by $|\{p, \ldots, k-1\}|$ since $\bar{S}^k \subseteq \{p, \ldots, k-1\}$.

Under Condition 2, notice that $|\{p, \ldots, k-1\}| = \max\{|\{p, \ldots, k-1\}|, |S^k|\} =: \mathrm{dist}(x^k, z^k)$, where we use $S^k \subseteq \{p, \ldots, k-1\}$ and Definition 2.2. Thus, to get a bound for $\|x^k - z^k\|$, it is natural to introduce Condition 3, which allows us to conclude that $|\bar{S}^k| \leq \mathrm{dist}(x^k, z^k) \leq R$. It remains to use classical mathematical tools to obtain

$$\mathbb{E}\left[\left\| x^k - z^k \right\|^2\right] \leq 2\gamma^2 R \sum_{j=k-R}^{k-1} \mathbb{E}\left[\left\| \nabla f(z^j) \right\|^2\right] + 2\gamma^2 R \sigma^2,$$

where the first term will be canceled by the corresponding term $-\frac{\gamma}{4}\mathbb{E}\left[\left\| \nabla f(z^k) \right\|^2\right]$ from (6).

## D.2    FULL PROOF

**Theorem 2.4** (Main Theorem). *Let Assumptions 1.1, 1.2, and 1.3 hold. Consider any* SGD *method represented by computation tree $G = (V, E)$. Let $\{x^k\}_{k \geq 0}$ be a main branch of $G$ and $\{(z^k, \xi^k)\}_{k \geq 0}$ be the corresponding auxiliary sequence (see Def. 2.1) that satisfy the following conditions:*
**Condition 1:** *For all $k \geq 0$, $\xi^k$ is statistically independent of $\{(x^{i+1}, z^{i+1}, \xi^i)\}_{i=0}^{k-1}$.*
**Condition 2:** *The representation of $z^k$ is contained within that of $x^k$, i.e., $\mathrm{repr}(z^k) \subseteq \mathrm{repr}(x^k)$ for all $k \geq 0$. Equivalently, all stochastic gradients used in the computation of $z^k$ are also utilized in calculating $x^k$.*
**Condition 3:** *There exists a constant $R \in [0, \infty]$ such that $\mathrm{dist}(x^k, z^k) \leq R$ for all $k \geq 0$. Then $\frac{1}{K}\sum_{k=0}^{K-1} \mathbb{E}\left[\|\nabla f(x^k)\|^2\right] \leq \varepsilon$ for all $K \geq \frac{4(R+1)L\Delta}{\varepsilon} + \frac{8\sigma^2 L\Delta}{\varepsilon^2}$ with step size $\gamma = \min\{\frac{1}{2L}, \frac{1}{2RL}, \frac{\varepsilon}{4\sigma^2 L}\}$, where $\Delta = f(x^0) - f^*$.*

*Proof.* As the beginning, the analysis is standard. Using Assumption 1.1, we have

$$f(x^{k+1}) \leq f(x^k) - \gamma \left\langle \nabla f(x^k), \nabla f(z^k; \xi^k) \right\rangle + \frac{L\gamma^2}{2}\left\| \nabla f(z^k; \xi^k) \right\|^2$$

for $x^{k+1} = x^k - \gamma \nabla f(z^k; \xi^k)$. Due to Condition 1 of the theorem and the variance decomposition equality,

$$\mathbb{E}_k\left[f(x^{k+1})\right] \leq f(x^k) - \gamma\left\langle\nabla f(x^k), \nabla f(z^k)\right\rangle + \frac{L\gamma^2}{2}\mathbb{E}_k\left[\left\|\nabla f(z^k; \xi^k)\right\|^2\right]$$

$$= f(x^k) - \gamma\left\langle\nabla f(x^k), \nabla f(z^k)\right\rangle + \frac{L\gamma^2}{2}\left\|\nabla f(z^k)\right\|^2 + \frac{L\gamma^2}{2}\mathbb{E}_k\left[\left\|\nabla f(z^k; \xi^k) - \nabla f(z^k)\right\|^2\right]$$

$$\leq f(x^k) - \gamma\left\langle\nabla f(x^k), \nabla f(z^k)\right\rangle + \frac{L\gamma^2}{2}\left\|\nabla f(z^k)\right\|^2 + \frac{L\gamma^2\sigma^2}{2},$$

where $\mathbb{E}_k\left[\cdot\right]$ is the expectation conditioned on $(x^k, z^k)$. In the last inequality, we use Assumption 1.3. Rewriting the dot product and using $\gamma \leq \frac{1}{2L}$, we obtain

$$\mathbb{E}_k\left[f(x^{k+1})\right]$$

$$\leq f(x^k) - \frac{\gamma}{2}\left(\left\|\nabla f(x^k)\right\|^2 + \left\|\nabla f(z^k)\right\|^2 - \left\|\nabla f(x^k) - \nabla f(z^k)\right\|^2\right) + \frac{L\gamma^2}{2}\left\|\nabla f(z^k)\right\|^2 + \frac{L\gamma^2\sigma^2}{2}$$

$$\leq f(x^k) - \frac{\gamma}{2}\left\|\nabla f(x^k)\right\|^2 - \frac{\gamma}{4}\left\|\nabla f(z^k)\right\|^2 + \frac{\gamma}{2}\left\|\nabla f(x^k) - \nabla f(z^k)\right\|^2 + \frac{L\gamma^2\sigma^2}{2}. \tag{6}$$

In the rest of the proof, we focus on $\left\|\nabla f(x^k) - \nabla f(z^k)\right\|^2$. Using Assumption 1.1, we obtain

$$\left\|\nabla f(x^k) - \nabla f(z^k)\right\|^2 \leq L^2\left\|x^k - z^k\right\|^2. \tag{7}$$

Notice that there exist $p \in \{0, \dots, k\}$ and the closest common ancestor $x^p$ such that

$$x^k = x^p - \gamma\sum_{i=p}^{k-1}\nabla f(z^i; \xi^i) = x^0 - \gamma\sum_{i=0}^{p-1}\nabla f(z^i; \xi^i) - \gamma\sum_{i=p}^{k-1}\nabla f(z^i; \xi^i)$$

and

$$z^k = x^p - \gamma\sum_{(w,\xi)\in S^k}\nabla f(w; \xi) = x^0 - \gamma\sum_{i=0}^{p-1}\nabla f(z^i; \xi^i) - \gamma\sum_{(w,\xi)\in S^k}\nabla f(w; \xi),$$

where $S^k$ is the set of points and random variables used to compute $z^k$ starting from $x^p$ (see Figure 3). Moreover, due to Condition 3, we have $\text{dist}(x^k, z^k) \leq \max\{k - p, |S^k|\} \leq R$, meaning $p \geq k - R$. In total,

$$k \geq p \geq k - R, \tag{8}$$

which we use later. Condition 2 assumes

$$\text{repr}(z^k) := \underbrace{\{(z^i; \xi^i)\}_{i=0}^{p-1}}_{A} \uplus \underbrace{\{(w; \xi)\}_{(w,\xi)\in S^k}}_{C}$$

$$\subseteq \text{repr}(x^k) := \underbrace{\{(z^i; \xi^i)\}_{i=0}^{p-1}}_{A} \uplus \underbrace{\{(z^i; \xi^i)\}_{i=p}^{k-1}}_{B},$$

where $\uplus$ is the multiset union operation. Thus

$$\underbrace{\{(w; \xi)\}_{(w,\xi)\in S^k}}_{C} \subseteq \underbrace{\{(z^i; \xi^i)\}_{i=p}^{k-1}}_{B}$$

and

$$x^k - z^k = -\gamma\left(\sum_{i=p}^{k-1}\nabla f(z^i; \xi^i) - \sum_{(w,\xi)\in S^k}\nabla f(w; \xi)\right) = -\gamma\sum_{j\in\bar{S}^k}\nabla f(z^j; \xi^j), \tag{9}$$

where $\bar{S}^k$ is a set such that $\bar{S}^k \subseteq \{p, \dots, k-1\}$. Substituting (9) to (7),

$$\left\|\nabla f(x^k) - \nabla f(z^k)\right\|^2 \leq L^2\gamma^2\left\|\sum_{j\in\bar{S}^k}\nabla f(z^j; \xi^j)\right\|^2.$$

Next, using Young's inequality $\|x + y\|^2 \leq 2\|x\|^2 + 2\|y\|^2$ for all $x, y \in \mathbb{R}^d$, we get

$$\mathbb{E}\left[\left\|\nabla f(x^k) - \nabla f(z^k)\right\|^2\right] \leq 2L^2\gamma^2\mathbb{E}\left[\left\|\sum_{j \in \bar{S}^k} \nabla f(z^j)\right\|^2\right] + 2L^2\gamma^2\mathbb{E}\left[\left\|\sum_{j \in \bar{S}^k} (\nabla f(z^j; \xi^j) - \nabla f(z^j))\right\|^2\right].$$

Since $\xi^j$ is statistically independent of $\{(x^{i+1}, z^{i+1}, \xi^i)\}_{i=0}^{j-1}$ for all $j \in \bar{S}^k$ (Condition 1) and using Assumption 1.3,

$$\mathbb{E}\left[\left\|\nabla f(x^k) - \nabla f(z^k)\right\|^2\right] \leq 2L^2\gamma^2\mathbb{E}\left[\left\|\sum_{j \in \bar{S}^k} \nabla f(z^j)\right\|^2\right] + 2L^2\gamma^2 \left|\bar{S}^k\right| \sigma^2$$

$$\overset{\text{Jensen's ineq.}}{\leq} 2L^2\gamma^2 \left|\bar{S}^k\right| \sum_{j \in \bar{S}^k} \mathbb{E}\left[\left\|\nabla f(z^j)\right\|^2\right] + 2L^2\gamma^2 \left|\bar{S}^k\right| \sigma^2.$$

Due to $\bar{S}^k \subseteq \{p, \ldots, k-1\}$ and (8):

$$\mathbb{E}\left[\left\|\nabla f(x^k) - \nabla f(z^k)\right\|^2\right] \leq 2L^2\gamma^2 R \sum_{j=k-R}^{k-1} \mathbb{E}\left[\left\|\nabla f(z^j)\right\|^2\right] + 2L^2\gamma^2 R\sigma^2.$$

Substituting this inequality to (6) and taking the full expectation, we obtain

$$\mathbb{E}\left[f(x^{k+1})\right] \leq \mathbb{E}\left[f(x^k)\right] - \frac{\gamma}{2}\mathbb{E}\left[\left\|\nabla f(x^k)\right\|^2\right] - \frac{\gamma}{4}\mathbb{E}\left[\left\|\nabla f(z^k)\right\|^2\right] + \frac{L\gamma^2\sigma^2}{2}$$

$$+ \frac{\gamma}{2}\left(2L^2\gamma^2 R \sum_{j=k-R}^{k-1} \mathbb{E}\left[\left\|\nabla f(z^j)\right\|^2\right] + 2L^2\gamma^2 R\sigma^2\right)$$

$$\leq \mathbb{E}\left[f(x^k)\right] - \frac{\gamma}{2}\mathbb{E}\left[\left\|\nabla f(x^k)\right\|^2\right] - \frac{\gamma}{4}\mathbb{E}\left[\left\|\nabla f(z^k)\right\|^2\right] + L\gamma^2\sigma^2$$

$$+ L^2\gamma^3 R \sum_{j=k-R}^{k-1} \mathbb{E}\left[\left\|\nabla f(z^j)\right\|^2\right] \tag{10}$$

because $\gamma \leq \frac{1}{2RL}$. Note that $\sum_{k=0}^{K-1} \sum_{j=k-R}^{k-1} \mathbb{E}\left[\left\|\nabla f(z^j)\right\|^2\right] \leq R \sum_{k=0}^{K-1} \mathbb{E}\left[\left\|\nabla f(z^k)\right\|^2\right]$. Thus, summing (10) for $k = 0, \ldots, K-1$ and substituting $f^*$,

$$\mathbb{E}\left[f(x^K) - f^*\right] \leq f(x^0) - f^* - \frac{\gamma}{2} \sum_{k=0}^{K-1} \mathbb{E}\left[\left\|\nabla f(x^k)\right\|^2\right] - \frac{\gamma}{4} \sum_{k=0}^{K-1} \mathbb{E}\left[\left\|\nabla f(z^k)\right\|^2\right] + KL\gamma^2\sigma^2$$

$$+ L^2\gamma^3 R^2 \sum_{k=0}^{K-1} \mathbb{E}\left[\left\|\nabla f(z^k)\right\|^2\right]$$

$$\leq f(x^0) - f^* - \frac{\gamma}{2} \sum_{k=0}^{K-1} \mathbb{E}\left[\left\|\nabla f(x^k)\right\|^2\right] + KL\gamma^2\sigma^2$$

because $\gamma \leq \frac{1}{2LR}$. Finally, since $\mathbb{E}\left[f(x^K) - f^*\right] \geq 0$,

$$\frac{1}{K} \sum_{k=0}^{K-1} \mathbb{E}\left[\left\|\nabla f(x^k)\right\|^2\right] \leq \frac{2\Delta}{K\gamma} + 2L\gamma\sigma^2.$$

It is left to use that $\gamma = \min\{\frac{1}{2L}, \frac{1}{2RL}, \frac{\varepsilon}{4\sigma^2 L}\}$ and the bound on $K$ from the theorem statement. $\square$

# E   DETAILED DESCRIPTION OF ALGORITHMS AND ITERATION RATES

In this section, we provide a detailed description together with theoretical analysis of the algorithms from the main part.

## E.1   Vanilla SGD

We start we the celebrated Vanilla SGD algorithm, which formally can be implemented in the following way:

---

**Algorithm 3** Vanilla SGD

---

1: **Input:** starting point $w^0 \in \mathbb{R}^d$, step size $\gamma > 0$
2: **for** $k = 0, 1, 2, \ldots$ **do**
3:     Sample $\eta^k \sim \mathcal{D}_\xi$                                              ($\{\eta^k\}$ are i.i.d.)
4:     Compute stochastic gradient $\nabla f(w^k; \eta^k)$
5:     Update $w^{k+1} = w^k - \gamma \nabla f(w^k; \eta^k)$
6: **end for**

---

The corresponding computation tree can defined by the recursion

$$w^{k+1} = w^k - \gamma \nabla f(w^k; \eta^k) \tag{11}$$

for all $k \geq 0$.

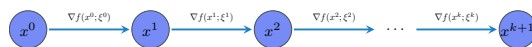

Figure 15: The computation tree of Vanilla SGD

While the iteration rate of Vanilla SGD is well-known (Lan, 2020), we prove its convergence using our new framework for clarity.

**Theorem E.1.** *Let Assumptions 1.1, 1.2, and 1.3 hold. Consider the computation tree (11) of* Vanilla SGD*. Then, $\{x^k\}_{k \geq 0}$ is a main branch with $x^k = w^k$, $\{(z^k, \xi^k)\}_{k \geq 0}$ is the corresponding auxiliary sequence with $(z^k, \xi^k) = (w^k, \eta^k)$ (see Def. 2.1), and $\frac{1}{K}\sum_{k=0}^{K-1} \mathbb{E}\left[\|\nabla f(x^k)\|^2\right] \leq \varepsilon$ for all*

$$K \geq \frac{4L\Delta}{\varepsilon} + \frac{8\sigma^2 L\Delta}{\varepsilon^2}.$$

*with step size $\gamma = \min\{\frac{1}{2L}, \frac{\varepsilon}{4\sigma^2 L}\}$.*

*Proof.* Indeed, $\{x^k\}_{k \geq 0}$ and $\{(z^k, \xi^k)\}_{k \geq 0}$ satisfy Def. 2.1 (see Fig. 15). Moreover, all conditions of Theorem 2.4 are fulfilled: Condition 1 holds because the sequence $\{\eta^k\}$ is i.i.d., we have $\text{repr}(z^k) = \text{repr}(x^k)$ since $x^k = z^k$, and consequently, $R = \sup_{k \geq 0} \text{dist}(x^k, z^k) = 0$. $\qquad \square$

### E.2 Rennala SGD

We now apply Theorem 2.4 to Rennala SGD. The iteration rate of Rennala SGD is also well-known (Tyurin & Richtárik, 2023), but we provide a proof for completness. Rennala SGD can be formally described as follows:

---

**Algorithm 4** Rennala SGD (Tyurin & Richtárik, 2023)

---

1: **Input:** point $w^0 \in \mathbb{R}^d$, stepsize $\gamma > 0$, batch size $B \in \mathbb{N}$
2: Workers start computing stochastic gradients at $w^0$
3: **for** $k = 0, \ldots, K - 1$ **do**
4:     $g_i^k = 0$ for all $i \in [n]$; $b = 0$
5:     **while** $b < B$ **do**
6:         Wait for the moment when stochastic gradient is computed by worker
7:         Gradient $\nabla f(w^{k-\delta}; \eta)$ is computed by worker $i$,     $\eta \sim \mathcal{D}_\xi$
8:         **if** $\delta = 0$ **then**
9:             Update $g_i^k = g_i^k + \nabla f(w^{k-\delta}; \eta)$ locally in worker $i$
10:             $b = b + 1$
11:         **else**
12:             Ignore $\nabla f(w^{k-\delta}; \eta)$
13:         **end if**
14:         Worker $i$ begins calculating gradient at $w^k$
15:     **end while**
16:     Aggregate: $g^k = \sum_{i=1}^n g_i^k$                              (e.g, via `AllReduce`)
17:     Update: $w^{k+1} = w^k - \gamma g^k$
18: **end for**

---

To use Theorem 2.4, we have to construct the computation tree of Rennala SGD. It can be constructed in the following way:

$$x^1 = x^0 - \gamma \nabla f(x^0; \xi^0), \quad \ldots, \quad x^B = x^{B-1} - \gamma \nabla f(x^0; \xi^{B-1}), \tag{12}$$
$$x^{B+1} = x^B - \gamma \nabla f(x^B; \xi^B), \quad \ldots, \quad x^{2B} = x^{2B-1} - \gamma \nabla f(x^B; \xi^{2B-1}), \quad \ldots,$$

where $\{\xi^i\}$ are i.i.d. from $\mathcal{D}_\xi$. See also a visualization in Figure 16. One can easily show that $w^1 = x^0, w^1 = x^B, w^2 = x^{2B}$, etc.

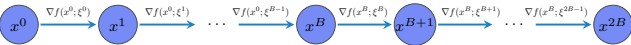

Figure 16: The computation tree of Rennala SGD

**Theorem E.2.** *Let Assumptions 1.1, 1.2, and 1.3 hold. Consider the computation tree* (12) *of* Rennala SGD, *then* $\{x^k\}_{k\geq 0}$ *is a main branch,* $\{(z^k, \xi^k)\}_{k\geq 0}$ *with* $(z^k, \xi^k) = (x^{B\lfloor k/B \rfloor}, \xi^k)$ *is the corresponding auxiliary sequence, and* $\frac{1}{K} \sum_{k=0}^{K-1} \mathbb{E}\left[\|\nabla f(x^k)\|^2\right] \leq \varepsilon$ *for all*

$$K \geq \frac{4BL\Delta}{\varepsilon} + \frac{8\sigma^2 L\Delta}{\varepsilon^2}.$$

*with step size* $\gamma = \min\{\frac{1}{2BL}, \frac{\varepsilon}{4\sigma^2 L}\}$.

*Proof.* Clearly, $\{x^k\}_{k\geq 0}$ is a main branch and $\{(z^k, \xi^k)\}_{k\geq 0}$ is the corresponding sequence by the construction in (12). Moreover, $\xi^k$ is independent of $\{(x^{i+1}, z^{i+1}, \xi^i)\}_{i=0}^{k-1}$ in (12) because $\{\xi^i\}$ are i.i.d. (Condition 1 is satisfied). Next, notice that

$$\mathrm{repr}(z^0) = \mathrm{repr}(x^0) = \emptyset,$$
$$\mathrm{repr}(z^1) = \mathrm{repr}(x^0) = \emptyset \subseteq \mathrm{repr}(x^1),$$

$$\vdots$$

$$\text{repr}(z^{B-1}) = \text{repr}(x^0) = \emptyset \subseteq \text{repr}(x^{B-1})$$

because $z^k = x^0$ for all $k < B$. Next,

$$\text{repr}(z^B) = \text{repr}(x^B),$$
$$\text{repr}(z^{B+1}) = \text{repr}(x^B) \subseteq \text{repr}(x^{B+1}),$$
$$\vdots$$
$$\text{repr}(z^{2B-1}) = \text{repr}(x^B) \subseteq \text{repr}(x^{2B-1}),$$

because $z^k = x^B$ for all $B \leq k < 2B$, where $\text{repr}(x^B) \subseteq \text{repr}(x^{B+1}), \ldots, \text{repr}(x^B) \subseteq \text{repr}(x^{2B-1})$ due to (12). We can continue and show that $\text{repr}(z^k) \subseteq \text{repr}(x^k)$ for all $k \geq 0$ (Condition 2 is satisfied). It is left to notice that

$$\sup_{k \geq 0} \text{dist}(x^k, z^k) \leq B - 1,$$

because

$$\text{dist}(x^0, z^0) = 0,$$
$$\text{dist}(x^1, z^1) = \text{dist}(x^1, x^0) = 1,$$
$$\vdots$$
$$\text{dist}(x^{B-1}, z^{B-1}) = \text{dist}(x^{B-1}, x^0) = B - 1,$$
$$\text{dist}(x^B, z^B) = \text{dist}(x^B, x^B) = 0,$$
$$\text{dist}(x^{B+1}, z^{B+1}) = \text{dist}(x^{B+1}, x^B) = 1,$$
$$\vdots$$

The maximum tree distance between $x^k$ and $z^k$ is $B - 1$. Thus, $R = B - 1$ in Condition 3. $\square$

### E.3 Local SGD

The Local SGD method is described in the following algorithm:

---

**Algorithm 5** Local SGD

---

**Require:** Initial model $w^0$, step size $\gamma$, parameter $B$
1: **for** $k = 0, 1, 2, \ldots$ **do**
2:     Broadcast $w^k$ to all workers
3:     **for** each worker $i \in [n]$ **in parallel do**
4:         Worker $i$ starts `LocalSGDWorker`$(w^k, \gamma)$ from Algorithm 6
5:     **end for**
6:     Wait for the moment when $\sum_{i=1}^{n} M_i = B$         ($\{M_i\}$ from `LocalSGDWorker`$(w^k, \gamma)$)
7:     Ask workers to stop[7] running `LocalSGDWorker`$(w^k, \gamma)$
8:     Aggregate $\gamma \sum_{i=1}^{n} \sum_{j=0}^{M_i - 1} \nabla f(z_i^{k,j}; \eta_i^{k,j})$ from the workers (e.g, via `AllReduce`)
9:     Update $w^{k+1} = w^k - \gamma \sum_{i=1}^{n} \sum_{j=0}^{M_i - 1} \nabla f(z_i^{k,j}; \eta_i^{k,j})$
10: **end for**

---

---

**Algorithm 6** `LocalSGDWorker`$(w, \gamma)$ in worker $i$ at round $k$

---

1: $z_i^{k,0} = w$
2: $M_i \leftarrow 0$
3: **while** True **do**
4:     $z_i^{k,M_i+1} = z_i^{k,M_i} - \gamma \nabla f(z_i^{k,M_i}; \eta_i^{k,M_i}), \quad \eta_i^{k,M_i} \sim \mathcal{D}_\xi$
5:     $M_i = M_i + 1$
6: **end while**

---

One key change compared to the previous work is that individual local steps $M_i$ are not predefined. Moreover, the server tracks the sum $\sum_{i=1}^{n} M_i$ and waits for the moment $\sum_{i=1}^{n} M_i = B$ before collecting the locally calculated gradients. With a proper choice of $B$, we will prove the optimal computational time complexity of the method in Section F.

The corresponding computation tree of Local SGD can be constructed in the following way. Define $N_k := k \times B$ and take $k = 0$. Then

$$
\begin{aligned}
z_i^{k,1} &= z_i^{k,0} - \gamma \nabla f(x^{N_k}; \eta_i^{k,0}), \\
z_i^{k,2} &= z_i^{k,1} - \gamma \nabla f(z_i^{k,1}; \eta_i^{k,1}), \\
&\vdots \\
z_i^{k,M_i} &= z_i^{k,M_i-1} - \gamma \nabla f(z_i^{k,M_i-1}; \eta_i^{k,M_i-1}),
\end{aligned}
\tag{13}
$$

---

[7]Alternatively, allow the workers to finish computing their stochastic gradients without waiting for them (since `AllReduce` can be run in parallel), but discard these gradients in subsequent iterations, as they are no longer relevant. This approach may introduce a delay before the workers begin their next local steps.

    Another option is to allow the workers to finish computing their stochastic gradients without waiting for them, and send these gradients in the next iteration. If some gradients are still not computed by then due to delays, simply discard them.

for all $i \in [n]$, and

$$x^{N_k+1} = x^{N_k} - \gamma \nabla f(z_1^{k,0}; \eta_1^{k,0}),$$

$$\vdots$$

$$x^{N_k+M_1} = x^{N_k+M_1-1} - \gamma \nabla f(z_1^{k,M_1-1}; \eta_1^{k,M_1-1}),$$

$$x^{N_k+M_1+1} = x^{N_k+M_1} - \gamma \nabla f(z_2^{k,0}; \eta_2^{k,0}),$$

$$\vdots \tag{14}$$

$$x^{N_k+M_1+M_2} = x^{N_k+M_1+M_2-1} - \gamma \nabla f(z_2^{k,M_2-1}; \eta_2^{k,M_2-1}),$$

$$\vdots$$

$$x^{N_{k+1}} = x^{N_k+\sum_{i=1}^{n} M_i-1} - \gamma \nabla f(z_n^{k,M_n-1}; \eta_n^{k,M_n-1}).$$

Repeat the previous steps with $k = k + 1$ starting at $x^{N_{k+1}} = x^{N_k+B}$. See illustration in Figure 4. One can easily show that $w^1 = x^B, w^2 = x^{2B}, \dots, w^k = x^{kB}, \dots$, where $w^k$ is the sequence from Algorithm 5.

**Theorem E.3.** *Let Assumptions 1.1, 1.2, and 1.3 hold. Consider the computation tree ((13) and (14)) of* Local SGD*, then $\{x^k\}_{k \geq 0}$ is a main branch and $\frac{1}{K} \sum_{k=0}^{K-1} \mathbb{E}\left[\|\nabla f(x^k)\|^2\right] \leq \varepsilon$ for all*

$$K \geq \frac{4BL\Delta}{\varepsilon} + \frac{8\sigma^2 L\Delta}{\varepsilon^2}.$$

*with step size $\gamma = \min\{\frac{1}{2BL}, \frac{\varepsilon}{4\sigma^2 L}\}$.*

Although the proof may seem technical due to the heavy notation in (14), it is actually straightforward when you refer to Figure 4. This figure clearly shows that all conditions of Theorem 2.4 are satisfied with $R = B - 1$ because $\sum_{i=1}^{n} M_i = B$ in every global iteration. The condition $\sum_{i=1}^{n} M_i = B$ helps us to insure that the maximum tree distance $\sup_{k \geq 0} \text{dist}(x^k, z^k) \leq B - 1$.

*Proof.* Clearly, $\{x^k\}_{k \geq 0}$ is a main branch by Definition 2.1. The corresponding auxiliary sequence can be inferred from (14): $(z^0, \xi^0) = (z_1^{0,0}, \eta_1^{0,0}), \dots, (z^{M_1}, \xi^{M_1}) = (z_1^{0,M_1}, \eta_1^{0,M_1})$, and etc. Condition 1 is satisfied because $\{\eta_i^{k,j}\}$ are i.i.d., and by the construction (14). Condition 2 of Theorem 2.4 holds because the same stochastic gradients used for computing $z^k$ are also used for $x^k$, as shown in Figure 4. This can be formally verified using (14) and (13). It is left to notice that

$$\sup_{k \geq 0} \text{dist}(x^k, z^k) \leq B - 1$$

because the maximum number of edges to the common closest ancestor can not exit $B - 1$. ☐

### E.4 Ringmaster ASGD

---

**Algorithm 7** Ringmaster ASGD (Maranjyan et al., 2025)

---

1: **Input:** point $w^0 \in \mathbb{R}^d$, stepsize $\gamma > 0$, delay threshold $G \in \mathbb{N}$
2: Set $k = 0$
3: Workers start computing stochastic gradients at $w^0$
4: **while** True **do**
5:      Gradient $\nabla f(w^{k-\delta^k}; \eta_i^{k-\delta^k})$ arrives from worker $i$
6:      **if** $\delta^k < G$ **then**
7:          Update: $w^{k+1} = w^k - \gamma \nabla f(w^{k-\delta^k}; \eta_i^{k-\delta^k})$
8:          Worker $i$ begins calculating at $w^{k+1}$          ($\{\eta_i^k\}$ are i.i.d.)
9:          Update the iteration number $k = k + 1$
10:     **else**
11:          Ignore the outdated gradient $\nabla f(w^{k-\delta^k}; \eta_i^{k-\delta^k})$
12:          Worker $i$ begins calculating at $w^k$
13:     **end if**
14: **end while**

---

In this method, a main branch can be defined as

$$x^k = w^k \tag{15}$$

and the auxiliary sequence is defined as $(z^k, \xi^k) = (x^{k-\delta^k}, \eta_i^{k-\delta^k})$ for all $k \geq 0$.

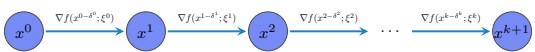

Figure 17: The computation tree of Ringmaster ASGD

**Theorem E.4.** *Let Assumptions 1.1, 1.2, and 1.3 hold. Consider the computation tree of* Ringmaster ASGD, *then* $\{x^k\}_{k \geq 0}$, *defined in* (15), *is a main branch and* $\frac{1}{K} \sum_{k=0}^{K-1} \mathbb{E}\left[\|\nabla f(x^k)\|^2\right] \leq \varepsilon$ *for all*

$$K \geq \frac{4GL\Delta}{\varepsilon} + \frac{8\sigma^2 L\Delta}{\varepsilon^2}.$$

*with step size* $\gamma = \min\{\frac{1}{2GL}, \frac{\varepsilon}{4\sigma^2 L}\}$.

*Proof.* Condition 1 is satisfied because $\{\eta_i^{k-\delta^k}\}$ are i.i.d., $x^k = w^k$ and $z^k = x^{k-\delta^k}$ do not depend on $\xi^k = \eta_i^{k-\delta^k}$. Condition 2 is satisfied because $\text{repr}(z^k) = \text{repr}(w^{k-\delta^k}) \subseteq \text{repr}(w^k) = \text{repr}(x^k)$. Condition 3 is satisfied with $R = G - 1$ because

$$\text{dist}(x^k, z^k) = \text{dist}(x^k, x^{k-\delta^k}) = \delta^k \leq G - 1,$$

where the second equality due to the number of edges between $x^k$ and $x^{k-\delta^k}$ and the last inequality due to the fact that $\delta^k$ is bounded by $B$ in Algorithm 7. $\square$

### E.5   Cycle SGD

We now present a new method, called Cycle SGD:

---
**Algorithm 8** Cycle SGD
---

**Require:** Initial model $w^0$, step size $\gamma$, group size $s$
 1: Partition workers into groups of size $s$:

$$G_1 = \{1, \ldots, s\}, G_2 = \{s+1, \ldots, 2s\}, \ldots, G_{\lceil n/s \rceil} = \{(\lceil n/s \rceil - 1)s + 1, \ldots, n\}$$

   in a circular manner.
 2: Broadcast $w^0$ to all workers and assign the local variables $z_i^0 = w^0$ and $M_i = 0$ for all $i \in [n]$
 3: **while** True **do**
 4:    **for** group index $g = 1$ to $\lceil n/s \rceil$ **do**
 5:       **for** each worker $i \in [n]$ **in parallel do**
 6:          $z_i^{M_i+1} = z_i^{M_i} - \gamma \nabla f(z_i^{M_i}; \eta_i^{M_i}), \quad \eta_i^{M_i} \sim \mathcal{D}_\xi$
 7:          $M_i = M_i + 1$
 8:       **end for**
 9:       Aggregate $\gamma \sum_{i \in G_g} \sum_{j=1}^{M_i} \nabla f(z_i^j; \eta_i^j)$ from the workers of group $G_g$ only
10:       Server aggregates and updates the model:

$$w^{r+1} = w^r - \gamma \sum_{i \in G_g} \sum_{j=0}^{M_i-1} \nabla f(z_i^j; \eta_i^j)$$

11:       Broadcast $w^{r+1}$ to all workers of group $g$ and assign the local variables $z_i^0 = w^{r+1}$ and $M_i = 0$ for all $i \in G_g$
12:       $r = r + 1$
13:    **end for**
14: **end while**

---

This method operates similarly to Local SGD, with workers performing local steps. However, a key difference is that only $s$ workers synchronize at each step, rather than all $n$ workers. This strategy can be advantageous in scenarios where reducing peak bandwidth is desirable. A visualization of the corresponding computation tree is in Figure 18. For this algorithm, the first $\sum_{i=1}^{n} M_i$ nodes of the main branch can be defined as

$$x^1 = x^0 - \gamma \nabla f(z_1^0; \eta_1^0),$$

$$\vdots$$

$$x^{M_1} = x^{M_1-1} - \gamma \nabla f(z_1^{M_1-1}; \eta_1^{M_1-1}),$$

$$\vdots$$

$$x^{\sum_{i=1}^{s-1} M_i + 1} = x^{\sum_{i=1}^{s-1} M_i} - \gamma \nabla f(z_s^0; \eta_s^0), \tag{16}$$

$$\vdots$$

$$x^{\sum_{i=1}^{s} M_i} = x^{\sum_{i=1}^{s} M_i - 1} - \gamma \nabla f(z_s^{M_s-1}; \eta_s^{M_s-1}),$$

$$\vdots$$

Notice that $x^{\sum_{i=1}^{s} M_i} \equiv w^1$, where we capture and unroll all stochastic gradients from the first group. The next nodes of the main branch can be defined in a similar way going through all groups circularly.

**Theorem E.5.** *Let Assumptions 1.1, 1.2, and 1.3 hold. Consider the computation tree of* Cycle SGD *(Alg. 8), then $\{x^k\}_{k \geq 0}$, defined in (16), is a main branch and $\frac{1}{K} \sum_{k=0}^{K-1} \mathbb{E}\left[\|\nabla f(x^k)\|^2\right] \leq \varepsilon$ for all*

$$K \geq \frac{8n^2 L \Delta}{s\varepsilon} + \frac{8\sigma^2 L \Delta}{\varepsilon^2}.$$

*with step size $\gamma = \min\{\frac{s}{4n^2 L}, \frac{\varepsilon}{4\sigma^2 L}\}$.*

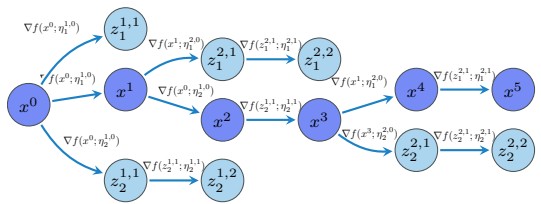

Figure 18: An example of Cycle SGD computation tree.

*Proof.* Once again, the proof is geometric. As an example, consider Figure 18 together with Algorithm 8. One can easily show that Conditions 1 and 2 are satisfied similarly to the proof of Theorem 5. However, the maximum tree distance is different since we synchronize the workers in a circular manner.

First, the number of local steps $M_i \leq \left\lceil \frac{n}{s} \right\rceil \leq \frac{2n}{s}$ because each worker computes one stochastic gradient in the inner loop and synchronizes every $\left\lceil \frac{n}{s} \right\rceil$ loops.

Next, the maximum tree distance between a point $x^k$ on the main branch and the corresponding point of the auxiliary sequence $z^k$ is at most $\frac{2n^2}{s}$. Let us explain this step. Consider any $x^k$ and $z^k$, and their closest common ancestor $w^k$ (in Figure 18, for instance, take $x^5$, $z_2^{2,2}$, and $x^3$ accordingly).

The number of edges from $z^k$ to $w^k$ never exceeds $\frac{2n}{s}$ due to the bound on the number of local steps. The number of edges from $x^k$ to $w^k$ never exceeds $\frac{2n^2}{s}$ because, while one worker performs local steps, other workers can grow the main branch by at most $\left\lceil \frac{n}{s} \right\rceil \times (n-1) \leq \frac{2n(n-1)}{s}$ points before the worker that computed $z^k$ is synchronized[8].

Thus, we can take $R = \frac{2n^2}{s}$ in Condition 3 of Theorem 2.4. □

---

[8]For instance, see Figure 18, where, before the algorithm applies $\nabla f(z_2^{2,1}; \xi_2^{2,1})$ from the second worker, the main branch grows by two edges, from $x^3$ to $x^5$, due to gradients computed by the first worker.

### E.6 Async-Local SGD

The following algorithm is a mixture of Asynchronous SGD and Local SGD, which we formalize in the following way.

---

**Algorithm 9** Async-Local SGD

---

1: **Input:** point $x^0 \in \mathbb{R}^d$, stepsize $\gamma > 0$, delay threshold $B \in \mathbb{N}$, number of local steps $M$
2: Set $k = 0$
3: Workers start running local steps at $w^0$ with Alg. 10 for $M$ steps
4: **while** True **do**
5:     Sum $\gamma \sum_{p=0}^{M-1} \nabla f(z_{i_k}^p; \eta_{i_k}^p)$ arrives from some worker $i_k$
6:     Find the tree distance $\delta^k = \text{dist}(w^k, z_{i_k}^0)$
    (delay $\delta^k$ of $w^{k-\delta^k}$, at which point worker $i_k$ started local steps)
7:     **if** $\delta^k < B$ **then**
8:         Update: $w^{k+1} = w^k - \gamma \sum_{p=0}^{M-1} \nabla f(z_{i_k}^p; \eta_{i_k}^p)$
9:         Worker $i$ starts running local steps at $w^{k+1}$ with Alg. 10 for $M$ steps
10:        Update the iteration number $k = k + 1$
11:     **else**
12:         Ignore the outdated sum $\gamma \sum_{p=0}^{M-1} \nabla f(z_{i_k}^p; \eta_{i_k}^p)$
13:         Worker $i$ starts running local steps at $w^k$ with Alg. 10 for $M$ steps
14:     **end if**
15: **end while**

---

**Algorithm 10** `LocalSGDWorker`$(w, \gamma, M)$ in worker $i$

---

1: $z_i^0 = w$
2: **for** $p = 0, \ldots M - 1$ **do**
3:     $z_i^{p+1} = z_i^p - \gamma \nabla f(z_i^p; \eta_i^p), \quad \eta_i^p \sim \mathcal{D}_\xi$
4: **end for**
5: Send to the server $\gamma \sum_{p=0}^{M-1} \nabla f(z_i^p; \eta_i^p)$

---

If $M = 1$, then this method reduces to Ringmaster ASGD (Alg. 7). Taking $M > 1$, we can improve the time complexity of Ringmaster ASGD by decreasing the number of times when workers synchronize with the server. For this method, it is natural to take a main branch as

$$x^1 = x^0 - \gamma \nabla f(z_{i_1}^0; \eta_{i_1}^0),$$

$$\vdots$$

$$x^M = x^{M-1} - \gamma \nabla f(z_{i_1}^{M-1}; \eta_{i_1}^{M-1}),$$

$$\vdots$$

$$x^{M(k-1)+1} = x^{M(k-1)} - \gamma \nabla f(z_{i_k}^0; \eta_{i_k}^0), \tag{17}$$

$$\vdots,$$

$$x^{Mk} = x^{Mk-1} - \gamma \nabla f(z_{i_k}^{M-1}; \eta_{i_k}^{M-1}),$$

$$\vdots$$

and so on. Notice that $x^0 \equiv w^0, x^M \equiv w^1$, etc.

**Theorem E.6.** *Let Assumptions 1.1, 1.2, and 1.3 hold. Consider the computation tree of* Async-Local SGD *(Alg. 9), then $\{x^k\}_{k \geq 0}$, defined in (17), is a main branch and $\frac{1}{K} \sum_{k=0}^{K-1} \mathbb{E}\left[\|\nabla f(x^k)\|^2\right] \leq \varepsilon$ for all*

$$K \geq \frac{4(B + M - 1)L\Delta}{\varepsilon} + \frac{8\sigma^2 L\Delta}{\varepsilon^2}.$$

*with step size* $\gamma = \min\{\frac{1}{4(B+M-1)L}, \frac{\varepsilon}{4\sigma^2 L}\}$.

*Proof.* Similar to the previous proofs, Condition 1 is satisfied for the main branch $\{x^k\}_{k\geq 0}$ because all random variables $\{\eta_j^i\}$ in (17) are i.i.d., and $x^0$ and $z_{i_1}^0$ do not depend on $\eta_{i_1}^0$. Points $x^{M-1}$ and $z_{i_1}^{M-1}$ do not depend on $\eta_{i_1}^{M-1}$, and so on. Conditions 2 is satisfied because all stochastic gradients used to compute $z_{i_k}^p$ are also used to compute the corresponding point on the main branch for all $p \in \{0, \ldots, M-1\}$ and $k \geq 0$ (see Figure 5). Condition 3 is satisfied with $R = B - 1 + M - 1 = B + M - 2$ due to the inequality $\delta^k = \text{dist}(w^k, z_{i_k}^0) < B$ in Algorithm 9 and the fact every worker calculates $M$ stochastic gradients, which ensures that the tree distance between $z_{i_k}^0$ and the corresponding point from the main brain branch is at most $B - 1$, the tree distance between $z_{i_k}^1$ and the corresponding point from the main brain branch is at most $B - 2$, $\ldots$, the tree distance between $z_{i_k}^{M-1}$ and the corresponding point from the main brain branch is at most $B + M - 2$. $\qquad \square$

### E.7  Async-Batch SGD

This method does the same steps as Async-Local SGD with the only difference that the workers calculate mini-batches instead of local steps:

---

**Algorithm 11** `BatchSGDWorker`$(w, \gamma, M)$ in worker $i$

1: $z_i^0 = w$
2: **for** $p = 0, \ldots M - 1$ **do**
3:     Calculate $\nabla f(z_i^p; \eta_i^p), \quad \eta_i^p \sim \mathcal{D}_\xi$
4:     $z_i^{p+1} = z_i^p$
5: **end for**
6: Send to the server $\gamma \sum_{p=0}^{M-1} \nabla f(z_i^p; \eta_i^p)$

---

One can easily show that these methods share the same theoretical guarantees (Sections E.6, F, and G) as Async-Local SGD.

### E.8 Local-Async SGD

One way to interpret the following algorithm is that the workers are partitioned into groups, with each group running Asynchronous SGD. Then, at certain points, all workers synchronize, and start running Asynchronous SGD at a new point. One of the important novelties here is the condition $\sum_{g=1}^{s} m_g = B$, which, with a proper $B$, leads to the optimal computational time complexity (Section F).

---

**Algorithm 12** Local-Async SGD

---

**Require:** Initial model $w^0$, step size $\gamma$, parameter $B$, group partitions $G_1, \ldots, G_s$
1: **for** $k = 0, 1, 2, \ldots$ **do**
2:     Broadcast $w^k$ to all groups
3:     **for** each worker $g \in [s]$ **in parallel do**
4:         Group $g$ starts AsynchronousSGDGroup($w^k, \gamma$) from Algorithm 13
5:     **end for**
6:     Wait for the moment when $\sum_{g=1}^{s} m_g = B$    ($\{m_g\}$ from AsynchronousSGDGroup($w^k, \gamma$))
7:     Ask the groups to stop[9] running AsynchronousSGDGroup($w^k, \gamma$)
8:     Aggregate $\gamma \sum_{g=1}^{s} \sum_{j=0}^{m_g-1} \nabla f(v_g^{j-\delta^j}; \eta_g^j)$ from the groups           ($\{\eta_g^j\}$ are i.i.d.)
9:     Update $w^{k+1} = w^k - \gamma \sum_{g=1}^{s} \sum_{j=0}^{m_g-1} \nabla f(v_g^{j-\delta^j}; \eta_g^j)$
10: **end for**

---

**Algorithm 13** AsynchronousSGDGroup($w, \gamma$) in group $g$

---

**Input:** point $v_g^0 \in \mathbb{R}^d$, stepsize $\gamma > 0$
Set $m_g = 0$
Workers from group $g$ start computing stochastic gradients at $v_g^0$
**while** True **do**
    Gradient $\nabla f(v_g^{m_g - \delta^{m_g}}; \eta_g^{m_g})$ arrives from worker $i$ with delay $\delta^{m_g}$
    Update: $v_g^{m_g+1} = v_g^{m_g} - \gamma \nabla f(v_g^{m_g - \delta^{m_g}}; \eta_g^{m_g})$
    Worker $i$ begins calculating stochastic gradient at $v_g^{m_g+1}$
    Update the iteration number $m_g = m_g + 1$
**end while**

---

For this method, it is natural to take a main branch of the computation tree as

$$x^1 = x^0 - \gamma \nabla f(v_1^{0-\delta^0}; \eta_1^0),$$

$$\vdots$$

$$x^{m_1} = x^{m_1-1} - \gamma \nabla f(v_1^{m_1-1-\delta^{m_1-1}}; \eta_1^{m_1-1}),$$

$$\vdots$$

$$x^{\sum_{g=1}^{s-1} m_i + 1} = x^{\sum_{g=1}^{s-1} m_i} - \gamma \nabla f(v_s^{0-\delta^0}; \eta_s^0), \tag{18}$$

$$\vdots,$$

$$x^{\sum_{g=1}^{s} m_i} = x^{\sum_{g=1}^{s} m_i - 1} - \gamma \nabla f(v_s^{m_s-1-\delta^{m_s-1}}; \eta_s^{m_s-1})$$

$$\vdots,$$

---

[9]Alternatively, allow the workers to finish computing their stochastic gradients without waiting for them (since `AllReduce` can be run in parallel), but discard these gradients in subsequent iterations, as they are no longer relevant. This approach may introduce a delay before the workers begin their next local steps.

Another option is to allow the workers to finish computing their stochastic gradients without waiting for them, and send these gradients in the next iteration. If some gradients are still not computed by then due to delays, simply discard them.

where one can see that $x^{\sum_{g=1}^{s} m_i} \equiv x^B \equiv w^1$, and $\{v_g^j\}$ is defined in Algorithm 13.

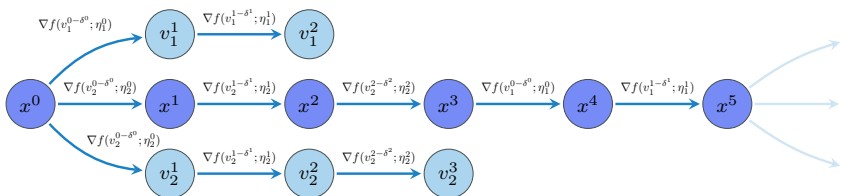

Figure 19: An example of a Local-Async SGD computation tree with two groups and $B = 5$. One group performs $m_1 = 3$ steps of Asynchronous SGD, while the other performs $m_2 = 2$ steps. Note that the maximum tree distance is $\text{dist}(x^4, v_1^{1-\delta^1})$ when applying $\nabla f(v_1^{1-\delta^1}; \eta_1^1)$ to $x^4$, and it equals $B - 1 = m_1 + m_2 - 1 = 4$. Then, the groups synchronize and continue from $x^5$.

**Theorem E.7.** *Let Assumptions 1.1, 1.2, and 1.3 hold. Consider the computation tree of* Local-Async SGD *(Alg. 12), then* $\{x^k\}_{k \geq 0}$, *defined in* (18)*, is a main branch and* $\frac{1}{K} \sum_{k=0}^{K-1} \mathbb{E}\left[\|\nabla f(x^k)\|^2\right] \leq \varepsilon$ *for all*

$$K \geq \frac{4BL\Delta}{\varepsilon} + \frac{8\sigma^2 L\Delta}{\varepsilon^2}.$$

*with step size* $\gamma = \min\{\frac{1}{4BL}, \frac{\varepsilon}{4\sigma^2 L}\}$.

*Proof.* The proof closely follows that of Theorem E.3, with the only difference being that the *auxiliary branches* in Algorithm 13 are constructed using asynchronous steps rather than local steps (compare Figure 4 and Figure 19). As in Theorem E.3, the condition $\sum_{g=1}^{s} m_g = B$ ensures that $\sup_{k \geq 0} \text{dist}(x^k, z^k) \leq B - 1$. $\qquad\square$

### E.9 Nested Local-Async SGD

In this section, we formalize a hierarchical version of Algorithm 12. Our framework, Theorem 2.4, is flexible enough to support such a two-level structure, where each cluster consists of servers equipped with (4–8) GPUs. The GPUs run Asynchronous SGD, the servers synchronize within their clusters, and finally, the clusters synchronize with each other.

In the following algorithm, all workers are partitioned into $\{G_{ij}\}$ groups, where $i$ is the cluster index and $j$ is the server index within the cluster. The set $G_{ij}$ contains the indices of the workers (GPUs).

---

**Algorithm 14** Nested Local-Async SGD

---

**Require:** Initial model $w^0$, step size $\gamma$, parameters $B_i$, global parameter $B$, group partitions $\{G_{ij}\}$
1: **for** $k = 0, 1, 2, \ldots$ **do**
2:     Broadcast $w^k$ to all clusters
3:     **for** each cluster $i$ **in parallel do**
4:         Set $w_i^0 = w^k$
5:         **for** $p_i = 0, 1, 2, \ldots$ **do**
6:             Broadcast $w_i^{p_i}$ to all local groups
7:             **for** each server $j$ **in parallel do**
8:                 Group $G_{ij}$ starts AsynchronousSGDGroup($w_i^{p_i}, \gamma$) from Algorithm 13
9:             **end for**
10:            Cluster $i$ waits for the moment when $\sum_j m_{ij} = B_i$
11:            Ask the groups in cluster $i$ to stop running AsynchronousSGDGroup($w_i^{p_i}, \gamma$)
12:            Update $w_i^{p_i+1} = w_i^{p_i} - \gamma \sum_j \sum_{\ell=0}^{m_{ijp_i}-1} \nabla f(v_{ijp_i}^{\ell-\delta^\ell}; \eta_{ijp_i}^\ell)$
13:         **end for**
14:     **end for**
15:     Wait for the moment the total number of local steps in the clusters starting from the last broadcast is $B$
16:     Ask all groups in all servers to stop running AsynchronousSGDGroup($w^k, \gamma$)
17:     Update $w^{k+1} = w^k - \sum_i (w_i^{p_i} - w_i^0) = w^k - \gamma \sum_i \sum_{k=0}^{p_i-1} \sum_j \sum_{\ell=0}^{m_{ijk}-1} \nabla f(v_{ijk}^{\ell-\delta^\ell}; \eta_{ijk}^\ell)$
18: **end for**

---

---

**Algorithm 15** AsynchronousSGDGroup($w, \gamma$) in group $G_{ij}$

---

  **Input:** point $v_{ijp_i}^0 \in \mathbb{R}^d$, stepsize $\gamma > 0$
  Set $m_{ij} = 0$
  Workers from group $G_{ij}$ start computing stochastic gradients at $v_{ijp_i}^0$
  **while** True **do**
    Gradient $\nabla f(v_{ijp_i}^{m_{ijp_i}-\delta^{m_{ijp_i}}}; \eta_{ijp_i}^{m_{ijp_i}})$ arrives from worker $i$ with delay $\delta^{m_{ijp_i}}$
    Update: $v_{ijp_i}^{m_{ijp_i}+1} = v_{ijp_i}^{m_{ijp_i}} - \gamma \nabla f(v_{ijp_i}^{m_{ijp_i}-\delta^{m_{ijp_i}}}; \eta_{ijp_i}^{m_{ijp_i}})$
    Worker $i$ begins calculating stochastic gradient at $v_{ijp_i}^{m_{ijp_i}+1}$
    Update the iteration number $m_{ijp_i} = m_{ijp_i} + 1$
  **end while**

---

We believe that analyzing this algorithm directly using classical optimization tools would be challenging due to heavy notations. However, using our framework and geometrical graph reasoning, we can easily prove the iteration rate of this algorithm. As in all previous cases, a main branch $x^k$ can be defined by taking each component of the sum $\sum_i \sum_{k=0}^{p_i-1} \sum_j \sum_{\ell=0}^{m_{ijk}-1} \nabla f(v_{ijk}^{\ell-\delta^\ell}; \eta_{ijk}^\ell)$ and applying each stochastic gradient to $x^0$, $x^1 = x^0 - \gamma \nabla f(v_{110}^0; \eta_{110}^0)$, and so on.

**Theorem E.8.** *Let Assumptions 1.1, 1.2, and 1.3 hold. Consider the computation tree of* Nested Local-Async SGD *(Alg. 14), then $\frac{1}{K} \sum_{k=0}^{K-1} \mathbb{E}\left[\|\nabla f(x^k)\|^2\right] \leq \varepsilon$ for all*

$$K \geq \frac{4BL\Delta}{\varepsilon} + \frac{8\sigma^2 L\Delta}{\varepsilon^2}.$$

*with step size $\gamma = \min\{\frac{1}{4BL}, \frac{\varepsilon}{4\sigma^2 L}\}$ for the main branch $\{x^k\}$ (slightly informally) defined above.*

*Proof.* Similarly to the previous proofs, Conditions 1 and 2 are satisfied by the construction of the algorithm. Using geometric graph reasoning, Condition 3 is satisfied with $R \leq B$ due to the requirement that "the total number of local steps in the clusters starting from the last broadcast is $B$." This ensures that the distance between the points of the main branch and the corresponding points of the auxiliary sequence defined by $v_{\cdot}^{\cdot}$ does not exceed $B$. $\qquad\square$

*Remark* E.9. One can see that the converge rate does not depend on $\{B_i\}$. Theoretically, it is sufficient to take $B_i = \infty$. However, practically, it may be better to take $B_i < \infty$ to ensure that the GPUs synchronize more often and share information with others, but it can lead to communication overhead and less efficient utilization of the GPUs.

### E.10 Meta Local SGD

Compared to previous algorithms, Meta Local SGD is an abstract or meta-method, as it includes one abstract step: "Wait if needed and take any set of workers $S$." This step is not explicitly defined, allowing users to apply any strategy they prefer. It may be random, where the algorithm chooses a uniformly random subset; it may follow a condition as in Local SGD, where the algorithm waits until $\sum_{i=1}^{n} M_i = B$; or it may be based on the current communication speeds of the workers, where the algorithm selects the workers with the fastest communication speeds at the current optimization moment.

However, as we explain in the main part, this can lead to a computation tree with a large $R$. That is why we check the condition $\max_{j \in [n]} d_j + \sum_{i=1}^{n} M_i < B$ in the algorithm, where $\{d_i\}$ are the current distances to the head of the main branch and $\{M_i\}$ are the local steps performed by each worker. If this condition is satisfied, we can take any set of workers $S$. Otherwise, we find a set of workers $S = \{j \in [n] \mid d_j + \sum_{i=1}^{n} M_i = B\}$ and ask them to send their calculated stochastic gradients. The latter case is required to synchronize workers with "very old stochastic gradients". Intuitively, if we do not synchronize them, their stochastic gradients may become too outdated and harmful to the optimization process.

---

**Algorithm 16** Meta Local SGD

1: **Input:** point $w^0 \in \mathbb{R}^d$, stepsize $\gamma > 0$, parameter $B \in \mathbb{N}$
2: Set an auxiliary distance variable $d_i = 0$ for all $i \in [n]$
3: Workers start running local steps at $w^0$ with Alg. 17
4: **for** $k = 0, 1, \ldots$ **do**
5:     **if** $\max_{j \in [n]} d_j + \sum_{i=1}^{n} M_i < B$ **then**
6:         Wait if needed and take any set of workers $S$                             (Soft Sync)
        (Here, we do not specify the selection method, it could be random or based on the current communication speeds. One can choose any strategy.)
7:     **else**
8:         Find a set of workers $S = \{j \in [n] \mid d_j + \sum_{i=1}^{n} M_i = B\}$               (Hard Sync)
9:     **end if**
10:    Ask workers from $S$ to send the calculated stochastic gradients and stop the loops in Alg. 17
11:    Receive $\gamma \sum_{i \in S} \sum_{p=0}^{M_i-1} \nabla f(z_i^p; \eta_i^p)$
12:    Update: $w^{k+1} = w^k - \gamma \sum_{p=0}^{M_i-1} \nabla f(z_i^p; \eta_i^p)$
13:    Worker from $S$ start running local steps at $w^{k+1}$ with Alg. 17
14:    Set $d_i = 0$ for all $i \in S$
15:    Update $d_i = d_i + \sum_{j \in S} M_j$ for all $i \notin S$
16: **end for**

---

**Algorithm 17** `LocalSGDWorker`$(w, \gamma)$ in worker $i$

1: $z_i^0 = w$
2: $M_i = 0$
3: **while** True **do**
4:    Calculate $\nabla f(z_i^{M_i}; \eta_i^{M_i})$,    $\eta^{M_i} \sim \mathcal{D}_\xi$
5:    **if** $\max_{j \in [n]} d_j + \sum_{i=1}^{n} M_i < B$ **then**
6:       $z_i^{M_i+1} = z_i^{M_i} - \gamma \nabla f(z_i^{M_i}; \eta_i^{M_i})$
7:       $M_i = M_i + 1$
8:    **end if**
9: **end while**

---

A main branch in the computation tree can be defined as follows. Assume that $S^k = \{i_1, \ldots, i_{p_k}\}$ is the set of workers participating in iteration $k$. Then, the computation tree with the main branch $\{x^k\}$

can be constructed as

$$x^1 = x^0 - \gamma \nabla f(z_{i_1}^0; \eta_{i_1}^0),$$
$$x^2 = x^1 - \gamma \nabla f(z_{i_1}^1; \eta_{i_1}^1),$$
$$\vdots$$
$$x^{M_{i_1}} = x^{M_{i_1}-1} - \gamma \nabla f(z_{i_1}^{M_{i_1}-1}; \eta_{i_1}^{M_{i_1}-1}),$$
$$\vdots \tag{19}$$
$$x^{\sum_{j=1}^{p_k-1} M_{ij}+1} = x^{\sum_{j=1}^{p_k-1} M_{ij}} - \gamma \nabla f(z_{i_{p_k}}^0; \eta_{i_{p_k}}^0),$$
$$\vdots$$
$$x^{\sum_{j=1}^{p_k} M_{ij}} = x^{\sum_{j=1}^{p_k} M_{ij}-1} - \gamma \nabla f(z_{i_{p_k}}^{M_{i_{p_k}}-1}; \eta_{i_{p_k}}^{M_{i_{p_k}}-1}),$$
$$\vdots$$

Notice that the end of each iteration block can be written as

$$w^1 \equiv x^{\sum_{j=1}^{p_0} M_{ij}}, \quad w^2 \equiv x^{\sum_{j=1}^{p_0} M_{ij} + \sum_{j=1}^{p_1} M_{ij}}, \quad \text{and so on.}$$

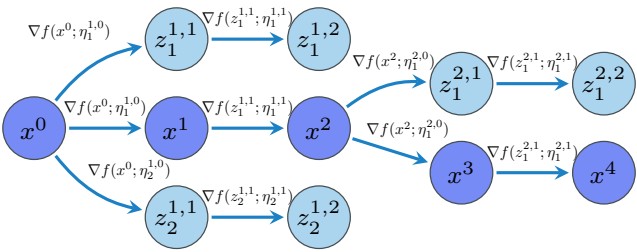

Figure 20: An example of the computation tree for Meta Local SGD with two workers. In this example, the first worker completes its first set of local steps, $x^0 \to z_2^{1,1} \to z_2^{1,2}$, and sends the stochastic gradients, which are used to calculate $x^1$ and $x^2$. A similar sequence of steps is repeated by the first worker to produce $x^2 \to z_2^{2,1} \to z_2^{2,2}$, followed by $x^3$ and $x^4$. At the same time, the second worker has only completed $x^0 \to z_1^{1,1} \to z_1^{1,2}$ and has not yet synchronized or sent the corresponding stochastic gradients. At this moment in time, the number of local steps is $M_2 = 2$ and $d_2 = 4$, because $d_2$ is the number of edges between the current main branch head $x^4$ and the point $x^0$, where the local branch of the second worker started. At the same time, $M_1 = 0$ and $d_1 = 0$, because the first worker has just started the third set of local steps at $x^4$ and has not yet calculated local stochastic gradients.

**Theorem E.10.** *Let Assumptions 1.1, 1.2, and 1.3 hold. Consider the computation tree of* Meta Local SGD *(Alg. 16), then* $\{x^k\}_{k \geq 0}$*, defined in (19), is a main branch and* $\frac{1}{K} \sum_{k=0}^{K-1} \mathbb{E}\left[\|\nabla f(x^k)\|^2\right] \leq \varepsilon$ *for all*

$$K \geq \frac{4BL\Delta}{\varepsilon} + \frac{8\sigma^2 L\Delta}{\varepsilon^2}.$$

*with step size* $\gamma = \min\{\frac{1}{4BL}, \frac{\varepsilon}{4\sigma^2 L}\}$.

*Proof.* Similarly to the previous proofs, it is clear that Conditions 1 and 2 from Theorem 2.4 are satisfied for the main branch (19).

It remains to show that $\text{dist}(x^k, z^k) \leq B$ for all $k \geq 0$. In the algorithm, we track two key sets of variables: $\{d_i\}$ and $\{M_i\}$. The variable $M_i$ denotes the current number of local steps performed by worker $i$, while $d_i$ represents the number of edges between the current end of the main branch and the point where worker $i$ began its local updates. When worker $i \notin S$, the distance $d_i$ increases as follows: $d_i = d_i + \sum_{j \in S} M_j$, since the workers in $S$ extend the main branch with their accumulated local updates.

The algorithm is constructed so that the quantity $\max_{j \in [n]} d_j + \sum_{i=1}^{n} M_i$ remains bounded by $B$ throughout the entire optimization process, ensuring that Condition 3 is satisfied with $R = B$. To clarify, assume that $i \in S$ in Algorithm 16. In the worst-case scenario, all other workers $j \in S$, with $j \neq i$, apply their local updates, increasing the tree distance from worker $i$'s branch to the main branch by at most $\sum_{j \in S, j \neq i} M_j$. Thus, the updated tree distance becomes at most $d_i + \sum_{j \in [n], j \neq i} M_j$. Since worker $i$ has also performed $M_i$ local steps, the tree distance is bounded by $d_i + \sum_{j \in [n], j \neq i} M_j + M_i \leq B$. $\qquad\square$

### E.11 Dual-Process SGD

We now present a new method, Dual-Process SGD, which is very similar to Local SGD. In fact, when communication is free, the two methods are equivalent. However, Local SGD requires all workers to send the sum of stochastic gradients only at the end of each round. In contrast, in Dual-Process SGD, workers do not wait until the end of the round; instead, they begin communicating sequentially as soon as possible.

Initially, each worker waits for the first stochastic gradients with index $0$ and immediately sends them once available. Then, while these are being transmitted, the workers continue their local computations. After the server receives the gradients with index $0$, the workers begin sending the next batch of stochastic gradients, starting from index $1$ up to the latest index they have computed at that moment. This process continues until the server has received a total of $B$ stochastic gradients, accumulated through the communicated sums. This logic is implemented in Algorithm 19.

---

**Algorithm 18** Dual-Process SGD

**Require:** Initial model $w^0$, step size $\gamma$, parameter $B$
1: **for** $k = 0, 1, 2, \ldots$ **do**
2:     Broadcast $w^k$ to all workers
3:     **for** each worker $i \in [n]$ **in parallel do**
4:         Worker $i$ starts `DualProcessLocalSGDWorker`$(w^k, \gamma)$ from Algorithm 19
5:     **end for**
6:     Start receiving the sum from the workers
7:     Wait for the moment when the total # of received gradients $\sum_{i=1}^n M_i = B$
8:     Ask workers to stop running `DualProcessLocalSGDWorker`$(w^k, \gamma)$
9:     Update $w^{k+1} = w^k - \gamma \sum_{i=1}^n \sum_{j=0}^{M_i-1} \nabla f(z_i^{k,j}; \eta_i^{k,j})$
10: **end for**

---

**Algorithm 19** `DualProcessLocalSGDWorker`$(w, \gamma)$ in worker $i$ at round $k$

1: $z_i^{k,0} = w$
2: $\widetilde{M}_i = \bar{M}_i = M_i = 0$
3: **Launch in parallel the following two processes:**
4: **Process 1:**
5: **while** True **do**
6:     Calculate $\nabla f(z_i^{k,\widetilde{M}_i}; \eta_i^{k,\widetilde{M}_i})$,   $\eta_i^{k,\widetilde{M}_i} \sim \mathcal{D}_\xi$
7:     $z_i^{k,\widetilde{M}_i+1} = z_i^{k,\widetilde{M}_i} - \gamma \nabla f(z_i^{k,\widetilde{M}_i}; \eta_i^{k,\widetilde{M}_i})$
8:     $\widetilde{M}_i = \widetilde{M}_i + 1$
9: **end while**
10:
11: **Process 2:**
12: **while** True **do**
13:     Wait until at least one new stochastic gradient is computed in Process 1.
14:     Set temporary variable $\bar{M}_i = \widetilde{M}_i$
15:     Send $\sum_{j=M_i}^{\bar{M}_i-1} \nabla f(z_i^{k,j}; \eta_i^{k,j})$
16:     Wait until the transmission is complete
17:     Update $M_i = \bar{M}_i$
18: **end while**

---

The computation tree of Dual-Process SGD defined in (13) and (14) is similar to Local SGD.

**Theorem E.11.** *Let Assumptions 1.1, 1.2, and 1.3 hold. Consider the computation tree ((13) and (14)) of Dual-Process SGD, then $\{x^k\}_{k \geq 0}$ is a main branch and $\frac{1}{K} \sum_{k=0}^{K-1} \mathbb{E}\left[\|\nabla f(x^k)\|^2\right] \leq \varepsilon$ for*

*all*

$$K \geq \frac{4BL\Delta}{\varepsilon} + \frac{8\sigma^2 L\Delta}{\varepsilon^2}.$$

*with step size* $\gamma = \min\{\frac{1}{2BL}, \frac{\varepsilon}{4\sigma^2 L}\}$.

*Proof.* The proof is exactly the same as in Theorem E.3 since the computation tree of Dual-Process SGD is similar to Local SGD. $\qquad\square$

# F    COMPUTATIONAL TIME COMPLEXITIES OF ALGORITHMS UNDER $h_i$-FIXED COMPUTATION MODEL

To compare methods, we consider the $h_i$-*fixed computation model* (Mishchenko et al., 2022). In this model, it is assumed that

$$\text{worker } i \text{ takes no more than } h_i \text{ seconds to compute a single stochastic gradient} \tag{20}$$

and

$$0 < h_1 \leq h_2 \leq \cdots \leq h_n, \tag{21}$$

without loss of generality.

Note that it is possible to consider the *universal computation model* (Tyurin, 2025) and capture virtually all possible computation behaviors of the workers. While the $h_i$-*fixed computation model* may seem more restrictive, it turns out that all optimal methods (Maranjyan et al., 2025) in the *universal computation model* are also optimal in the $h_i$-*fixed computation model*. Thus, for simplicity, we stick to the $h_i$-*fixed computation model*.

## F.1    Rennala SGD

**Theorem F.1** (Rennala SGD). *Consider Theorem E.2 and its conditions. Under the $h_i$-fixed computation model* (20)*, the computational time complexity of* Rennala SGD *(Alg. 4) is*

$$\mathcal{O}\left(\min_{m \in [n]} \left[ \left( \frac{1}{m} \sum_{i=1}^{m} \frac{1}{h_i} \right)^{-1} \left( \frac{L\Delta}{\varepsilon} + \frac{\sigma^2 L\Delta}{m\varepsilon^2} \right) \right] \right) \tag{22}$$

*with $B = \max\left\{ \left\lceil \frac{\sigma^2}{\varepsilon} \right\rceil, 1 \right\}$.*

We start with the following lemma.

**Lemma F.2.** *Let us define*

$$T_{\mathrm{R}}(B) := 2 \min_{m \in [n]} \left[ \left( \sum_{i=1}^{m} \frac{1}{h_i} \right)^{-1} (B + m) \right] \tag{23}$$

*Under the $h_i$-fixed computation model* (20)*, the time required to calculate $x^1, \ldots, x^B$ of the main branch is at most $T_{\mathrm{R}}(B)$ seconds, the time required to calculate $x^{B+1}, \ldots, x^{2B}$ is at most $T_{\mathrm{R}}(B)$ seconds, and so on.*

*Proof.* The idea of Rennala SGD (Alg. 4) is pretty simple. Notice that all workers calculate stochastic gradients at the same point in parallel until the server collects a batch of size $B$ (condition $\delta = 0$ ensures that). Since they work in parallel, under the fixed computational model, after $t$ seconds the workers will calculate

$$\sum_{i=1}^{n} \max \left\{ \left\lfloor \frac{t}{h_i} \right\rfloor - 1, 0 \right\} \tag{24}$$

stochastic gradients because $\left\lfloor \frac{t}{h_i} \right\rfloor$ is the number of stochastic gradients computed by worker $i$ in $t$ seconds. We subtract 1 because at most one stochastic gradient be can be ignored due to the condition $\delta = 0$ in Alg. 4.

Notice that

$$T_{\mathrm{R}}(B) = 2 \left( \sum_{i=1}^{m^*} \frac{1}{h_i} \right)^{-1} (B + m^*)$$

for some $m^* \in [n]$. Substituting it to (24), we get

$$\sum_{i=1}^{n} \max \left\{ \left\lfloor \frac{T_{\mathrm{R}}(B)}{h_i} \right\rfloor - 1, 0 \right\} \geq \sum_{i=1}^{m^*} \max \left\{ \left\lfloor \frac{T_{\mathrm{R}}(B)}{h_i} \right\rfloor - 1, 0 \right\} \geq \sum_{i=1}^{m^*} \left\lfloor \frac{T_{\mathrm{R}}(B)}{h_i} \right\rfloor - m^*$$

$$\geq \sum_{i=1}^{m^*} \frac{T_{\mathrm{R}}(B)}{h_i} - 2m^* = 2(B + m^*) - 2m^* \geq B.$$

Thus, after $T_{\mathrm{R}}(B)$ seconds, the server collects $B$ stochastic gradients, which is equivalent to calculating $x^1, \ldots, x^B$ of the main branch. The same argument can be applied to the next $B$ point of the main branch, and so on. □

*Proof of Theorem F.1.* Due to Theorem E.2, we know that $\frac{1}{K} \sum_{k=0}^{K-1} \mathbb{E}\left[\|\nabla f(x^k)\|^2\right] \leq \varepsilon$ for

$$K = \left\lceil \frac{4BL\Delta}{\varepsilon} + \frac{8\sigma^2 L\Delta}{\varepsilon^2} \right\rceil.$$

From Lemma F.2, we know that the time required to calculate $x^1, \ldots, x^B$ of the main branch is at most $T_{\mathrm{R}}(B)$ seconds, the time required to calculate $x^{B+1}, \ldots, x^{2B}$ is at most $T_{\mathrm{R}}(B)$ seconds, and so on. Thus, the total time to find an $\varepsilon$–stationary point is

$$\mathcal{O}\left(T_{\mathrm{R}}(B) \times \frac{K}{B}\right) = \mathcal{O}\left(T_{\mathrm{R}}(B) \times \left(\frac{L\Delta}{\varepsilon} + \frac{\sigma^2 L\Delta}{B\varepsilon^2}\right)\right).$$

Using the choice of $B$,

$$\mathcal{O}\left(T_{\mathrm{R}}(B) \times \frac{K}{B}\right) = \mathcal{O}\left(\min_{m \in [n]} \left[\left(\sum_{i=1}^{m} \frac{1}{h_i}\right)^{-1} (B + m)\right] \times \frac{L\Delta}{\varepsilon}\right)$$

$$= \mathcal{O}\left(\min_{m \in [n]} \left[\left(\frac{1}{m}\sum_{i=1}^{m} \frac{1}{h_i}\right)^{-1} \left(\frac{L\Delta}{\varepsilon} + \frac{\sigma^2 L\Delta}{m\varepsilon^2}\right)\right]\right).$$

□

## F.2 Ringmaster ASGD

**Theorem F.3** (Ringmaster ASGD). *Consider Theorem E.4 and its conditions. Under the $h_i$-fixed computation model (20), the computational time complexity of* Ringmaster ASGD *is*

$$\mathcal{O}\left(\min_{m \in [n]} \left[\left(\frac{1}{m}\sum_{i=1}^{m} \frac{1}{h_i}\right)^{-1} \left(\frac{L\Delta}{\varepsilon} + \frac{\sigma^2 L\Delta}{m\varepsilon^2}\right)\right]\right)$$

*with $B = \max\left\{\left\lceil \frac{\sigma^2}{\varepsilon} \right\rceil, 1\right\}$.*

We use Lemma 4.1 from (Maranjyan et al., 2025).

**Lemma F.4.** *((Maranjyan et al., 2025)) Let the workers' computation times satisfy the $h_i$-fixed computation model ((20) and (21)). Let $B$ be the delay threshold of Alg. 7. The time required to complete any $B$ consecutive iterate updates of Alg. 7 is at most*

$$T_A(B) := 2 \min_{m \in [n]} \left[\left(\frac{1}{m}\sum_{i=1}^{m} \frac{1}{\tau_i}\right)^{-1} \left(1 + \frac{R}{m}\right)\right]. \tag{25}$$

**Corollary F.5.** *In view of Lemma F.4, Under the $h_i$-fixed computation model (20), the time required to calculate $x^1, \ldots, x^B$ of the main branch is at most $T_A(B)$ seconds, the time required to calculate $x^{B+1}, \ldots, x^{2B}$ is at most $T_A(B)$ seconds, and so on.*

*Proof of Theorem F.3.* The proof of Theorem F.3 is similar to the proof of Theorem F.1. From Corollary F.5, we know that the time required to calculate $x^1, \ldots, x^B$ of the main branch is at most $T_A(B)$ seconds, the time required to calculate $x^{B+1}, \ldots, x^{2B}$ is at most $T_A(B)$ seconds, and so on. Thus, the total time to find an $\varepsilon$–stationary point is

$$\mathcal{O}\left(T_A(B) \times \frac{K}{B}\right) = \mathcal{O}\left(\min_{m \in [n]} \left[\left(\frac{1}{m}\sum_{i=1}^{m} \frac{1}{h_i}\right)^{-1} \left(\frac{L\Delta}{\varepsilon} + \frac{\sigma^2 L\Delta}{m\varepsilon^2}\right)\right]\right).$$

□

### F.3 Local SGD

**Theorem F.6** (Local SGD)**.** *Consider Theorem E.3 and its conditions. Under the $h_i$-fixed computation model* (20)*, the computational time complexity of* Local SGD *(Alg. 5) is*

$$\mathcal{O}\left(\min_{m\in[n]}\left[\left(\frac{1}{m}\sum_{i=1}^{m}\frac{1}{h_i}\right)^{-1}\left(\frac{L\Delta}{\varepsilon}+\frac{\sigma^2 L\Delta}{m\varepsilon^2}\right)\right]\right)$$

*with $B=\max\left\{\left\lceil\frac{\sigma^2}{\varepsilon}\right\rceil,1\right\}$.*

**Lemma F.7.** *Let us define*

$$T_{\mathrm{L}}(B):=2\min_{m\in[n]}\left[\left(\sum_{i=1}^{m}\frac{1}{h_i}\right)^{-1}(B+m)\right] \tag{26}$$

*Under the $h_i$-fixed computation model* (20)*, the time required to calculate $x^1,\ldots,x^B$ of the main branch is at most $T_{\mathrm{L}}(B)$ seconds, the time required to calculate $x^{B+1},\ldots,x^{2B}$ is at most $T_{\mathrm{L}}(B)$ seconds, and so on.*

*Proof.* The idea is the same as in Lemma F.2. All workers calculate stochastic gradients in parallel, with the only difference being that the points at which they compute the stochastic gradients differ due to the local steps. If the server can stop the workers, then after $t$ seconds it is possible to collect

$$\sum_{i=1}^{n}\left\lfloor\frac{t}{h_i}\right\rfloor \tag{27}$$

stochastic gradients. If it is infeasible to stop the calculations (see footnote 7), then after $t$ seconds it is possible to collect

$$\sum_{i=1}^{n}\max\left\{\left\lfloor\frac{t}{h_i}\right\rfloor-1,0\right\}, \tag{28}$$

where we subtract 1 because at most one stochastic gradient can be ignored if it is nonrelevant. Similarly to Lemma F.2, substituting $T_{\mathrm{L}}(B)$ into (27) and (28), one can show that $T_{\mathrm{L}}(B)$ is sufficient to collect $B=\sum_{i=1}^{n}M_i$ stochastic gradients, or, in other words, to calculate $x^1,\ldots,x^B$ of the main branch. The same argument can be applied to the next $B$ points of the main branch, and so on. □

*Proof of Theorem F.6.* The proof essentially the same as the proof of Theorem F.1. □

### F.4 Local-Async SGD

**Theorem F.8** (Local-Async SGD)**.** *Consider Theorem E.7 and its conditions. Under the $h_i$-fixed computation model* (20)*, the computational time complexity of* Local-Async SGD *(Alg. 12) is*

$$\mathcal{O}\left(\min_{m\in[n]}\left[\left(\frac{1}{m}\sum_{i=1}^{m}\frac{1}{h_i}\right)^{-1}\left(\frac{L\Delta}{\varepsilon}+\frac{\sigma^2 L\Delta}{m\varepsilon^2}\right)\right]\right)$$

*with $B=\max\left\{\left\lceil\frac{\sigma^2}{\varepsilon}\right\rceil,1\right\}$.*

**Lemma F.9.** *Let us define*

$$T_{\mathrm{LA}}(B):=2\min_{m\in[n]}\left[\left(\sum_{i=1}^{m}\frac{1}{h_i}\right)^{-1}(B+m)\right] \tag{29}$$

*Under the $h_i$-fixed computation model* (20)*, the time required to calculate $x^1,\ldots,x^B$ of the main branch is at most $T_{\mathrm{LA}}(B)$ seconds, the time required to calculate $x^{B+1},\ldots,x^{2B}$ is at most $T_{\mathrm{LA}}(B)$ seconds, and so on.*

*Proof.* The idea is the same as in Lemmas F.2 and F.7. All workers calculate stochastic gradients in parallel, with the only difference being that the points at which they compute the stochastic gradients differ due to the asynchronous steps in the groups. Similarly, one can show that $T_{\mathrm{LA}}(B)$ is sufficient time to calculate $B = \sum_{g=1}^{s} m_g$ stochastic gradients in Algorithm 12, or, equivalently, to calculate $x^1, \ldots, x^B$ of the main branch. The same argument can be applied to the next $B$ point of the main branch. □

*Proof of Theorem F.8.* The proof essentially the same as the proof of Theorem F.1. □

### F.5 Nested Local-Async SGD

**Theorem F.10** (Nested Local-Async SGD). *Consider Theorem E.8 and its conditions. Under the $h_i$-fixed computation model (20), the computational time complexity of* Nested Local-Async SGD *(Alg. 14) is*

$$\mathcal{O}\left(\min_{m \in [n]} \left[\left(\frac{1}{m}\sum_{i=1}^{m}\frac{1}{h_i}\right)^{-1}\left(\frac{L\Delta}{\varepsilon} + \frac{\sigma^2 L\Delta}{m\varepsilon^2}\right)\right]\right)$$

*with $B = \max\left\{\left\lceil\frac{\sigma^2}{\varepsilon}\right\rceil, 1\right\}$ and $B_i = \infty$[10] for all $i \in [n]$.*

*Proof.* The proof essentially the same as the proof of Theorem F.1. □

### F.6 Async-Local SGD

**Theorem F.11** (Async-Local SGD). *Consider Theorem E.6 and its conditions. Under the $h_i$-fixed computation model (20), the computational time complexity of* Async-Local SGD *(Alg. 9) is*

$$\mathcal{O}\left(\min_{m \in [n]} \left[\left(\frac{1}{m}\sum_{i=1}^{m}\frac{1}{h_i}\right)^{-1}\left(\frac{L\Delta}{\varepsilon} + \frac{\sigma^2 L\Delta}{m\varepsilon^2}\right)\right]\right)$$

*with $B = \max\left\{\left\lceil\frac{\sigma^2}{\varepsilon}\right\rceil, 1\right\}$ and $M = \max\left\{\left\lceil\frac{\sigma^2}{n\varepsilon}\right\rceil, 1\right\}$.*

**Lemma F.12.** *Let us define*

$$T_{\mathrm{AL}}(B, M) := 2\min_{m \in [n]}\left[\left(\sum_{i=1}^{m}\frac{1}{h_i}\right)^{-1}(B + Mm)\right] \tag{30}$$

*Under the $h_i$-fixed computation model (20), the time required to calculate $x^1, \ldots, x^B$ of the main branch is at most $T_{\mathrm{AL}}(B, M)$ seconds, the time required to calculate $x^{B+1}, \ldots, x^{2B}$ is at most $T_{\mathrm{AL}}(B, M)$ seconds, and so on.*

*Proof.* Let us fix $B$ and $M \geq 1$. Note that

$$T_{\mathrm{AL}}(B, M) := 2\min_{m \in [n]}\left[\left(\sum_{i=1}^{m}\frac{1}{h_i}\right)^{-1}(B + Mm)\right] = 2\left(\sum_{i=1}^{m^*}\frac{1}{h_i}\right)^{-1}(B + Mm^*) \tag{31}$$

for some $m^* \in [n]$, which depends on $B$ and $M$.

For any $k \geq 1$, consider the sequence $x^k, \ldots, x^{k+B}$ on the main branch. Using a proof by contradiction, assume that it requires more than $T_{\mathrm{AL}}(B, M)$ seconds to calculate $x^{k+1}, \ldots, x^{k+B}$. Thus, the algorithm can progress up to $x^{k+B-1}$ after $T_{\mathrm{AL}}(B, M)$ seconds.

In Algorithm 9, each worker computes $M$ stochastic gradients and sends their sum to the server. The server then performs the update $w^{k+1} = w^k - \gamma \sum_{p=0}^{M-1} \nabla f(z_{i_k}^p; \eta_{i_k}^p)$, which is equivalent to

---

[10]It is possible to take $B_i < \infty$, but the computational time complexity may decrease due to less utilization of workers. For simplicity, in this theorem, we take $B_i = \infty$. See also Remark E.9.

extending the main branch by $M$ points. Therefore, after $t$ seconds, the main branch will have progressed by at least

$$\sum_{i=1}^{n} \max\left\{ M \left\lfloor \frac{t}{Mh_i} \right\rfloor - M, 0 \right\}, \tag{32}$$

points (which is less than $B$ by assumption). This is because worker $i$ requires at most $Mh_i$ seconds to compute $M$ stochastic gradients before sending them to the server. Note that during any $B-1$ consecutive updates on the main branch, the server may ignore $M$ gradients from each worker at most once, because $\delta^k$ can be $\geq B$ at most once during $B-1$ consecutive updates. This explains the subtraction of $M$ in the formula.

Substituting $T_{\mathrm{AL}}(B, M)$ to (32),

$$\sum_{i=1}^{n} \max\left\{ M \left\lfloor \frac{T_{\mathrm{AL}}(B, M)}{Mh_i} \right\rfloor - M, 0 \right\} \geq \sum_{i=1}^{m^*} \max\left\{ M \left\lfloor \frac{T_{\mathrm{AL}}(B, M)}{Mh_i} \right\rfloor - M, 0 \right\}$$

$$\geq \sum_{i=1}^{m^*} M \left\lfloor \frac{T_{\mathrm{AL}}(B, M)}{Mh_i} \right\rfloor - Mm^* \geq \sum_{i=1}^{m^*} \frac{T_{\mathrm{AL}}(B, M)}{h_i} - 2Mm^*$$

because $\lfloor x \rfloor \geq x - 1$ for all $x \in \mathbb{R}$. Using (31),

$$\sum_{i=1}^{n} \max\left\{ M \left\lfloor \frac{T_{\mathrm{AL}}(B, M)}{Mh_i} \right\rfloor - M, 0 \right\} \geq 2(B + Mm^*) - 2Mm^* \geq B.$$

Thus, after $T_{\mathrm{AL}}(B, M)$ seconds, the server collects $B$ stochastic gradients. It is equivalent to calculating $x^{k+1}, \dots, x^{k+B}$, which contradicts the assumption. $\qquad\square$

*Proof of Theorem F.11.* Due to Theorem E.6, we know that $\frac{1}{K} \sum_{k=0}^{K-1} \mathbb{E}\left[\|\nabla f(x^k)\|^2\right] \leq \varepsilon$ for

$$K = \left\lceil \frac{4(B + M - 1)L\Delta}{\varepsilon} + \frac{8\sigma^2 L\Delta}{\varepsilon^2} \right\rceil.$$

From Lemma F.2, we know that the time required to calculate $x^1, \dots, x^B$ of the main branch is at most $T_{\mathrm{AL}}(B, M)$ seconds, the time required to calculate $x^{B+1}, \dots, x^{2B}$ is at most $T_{\mathrm{AL}}(B, M)$ seconds, and so on. Thus, the total time to find an $\varepsilon$–stationary point is

$$\mathcal{O}\left( T_{\mathrm{AL}}(B, M) \times \frac{K}{B} \right) = \mathcal{O}\left( T_{\mathrm{AL}}(B, M) \times \left( \frac{L\Delta(B + M)}{B\varepsilon} + \frac{\sigma^2 L\Delta}{B\varepsilon^2} \right) \right).$$

Using the choice of $B$ and $M$, we obtain $M \leq B$ and

$$\mathcal{O}\left( T_{\mathrm{R}}(B) \times \frac{K}{B} \right) = \mathcal{O}\left( T_{\mathrm{AL}}(B, M) \times \left( \frac{L\Delta}{\varepsilon} + \frac{\sigma^2 L\Delta}{B\varepsilon^2} \right) \right)$$

$$= \mathcal{O}\left( \min_{m \in [n]} \left[ \left( \sum_{i=1}^{m} \frac{1}{h_i} \right)^{-1} (B + Mm) \right] \times \left( \frac{L\Delta}{\varepsilon} + \frac{\sigma^2 L\Delta}{B\varepsilon^2} \right) \right)$$

$$= \mathcal{O}\left( \min_{m \in [n]} \left[ \left( \sum_{i=1}^{m} \frac{1}{h_i} \right)^{-1} (B + Mm) \right] \times \frac{L\Delta}{\varepsilon} \right)$$

because $B \geq \frac{\sigma^2}{\varepsilon}$. Since $M \leq \frac{\sigma^2}{n\varepsilon} + 1$ and $B \leq \frac{\sigma^2}{\varepsilon} + 1$,

$$\mathcal{O}\left( T_{\mathrm{R}}(B) \times \frac{K}{B} \right) = \mathcal{O}\left( \min_{m \in [n]} \left[ \left( \sum_{i=1}^{m} \frac{1}{h_i} \right)^{-1} \left( 1 + \frac{\sigma^2}{\varepsilon} + m + \frac{m\sigma^2}{n\varepsilon} \right) \right] \times \frac{L\Delta}{\varepsilon} \right)$$

$$= \mathcal{O}\left( \min_{m \in [n]} \left[ \left( \sum_{i=1}^{m} \frac{1}{h_i} \right)^{-1} \left( \frac{\sigma^2}{\varepsilon} + m \right) \right] \times \frac{L\Delta}{\varepsilon} \right)$$

$$= \mathcal{O}\left(\min_{m \in [n]}\left[\left(\frac{1}{m}\sum_{i=1}^{m}\frac{1}{h_i}\right)^{-1}\left(\frac{L\Delta}{\varepsilon} + \frac{\sigma^2 L\Delta}{m\varepsilon^2}\right)\right]\right),$$

where we use that $m \leq n$ for all $m \in [n]$. $\qquad \square$

### F.7 Cycle SGD

**Theorem F.13** (Cycle SGD). *Consider Theorem E.5 and its conditions. Under the $h_i$-fixed computation model* (20)*, the computational time complexity of* Cycle SGD *(Alg. 8) is*

$$\mathcal{O}\left(\max_{i \in [n]} h_i\left(\frac{L\Delta}{\varepsilon} + \frac{\sigma^2 L\Delta}{m\varepsilon^2}\right)\right)$$

*with* $s = \min\left\{\max\left\{\left\lceil\frac{n^2\varepsilon}{\sigma^2}\right\rceil, 1\right\}, n\right\}.$

*Proof.* According to Theorem E.5, $\frac{1}{K}\sum_{k=0}^{K-1}\mathbb{E}\left[\|\nabla f(x^k)\|^2\right] \leq \varepsilon$ for

$$K = \left\lceil\frac{8n^2 L\Delta}{s\varepsilon} + \frac{8\sigma^2 L\Delta}{\varepsilon^2}\right\rceil.$$

In the beginning, the algorithm has "warm-up", where, in the first iteration of the inner loop, the server collects $s$ stochastic gradients from $s$ workers, which is equivalent to calculating $x^1, \ldots, x^s$ of the main branch. Then, the server collects $2s$ stochastic gradients from the next group of $s$ workers because they calculated $s$ stochastic in the previous iteration. Starting from the $\left\lceil\frac{n}{s}\right\rceil^{\text{th}}$ iteration, each group of $s$ workers will return $s \times \left\lceil\frac{n}{s}\right\rceil$ stochastic gradients in every subsequent iteration. Every iterations takes at most $\max_{i \in [n]} h_i$ seconds, because they work in parallel and calculate one stochastic gradient.

Thus, the total time to calculate $x^1, \ldots, x^K$ and find an $\varepsilon$–stationary point is

$$\mathcal{O}\left(\underbrace{\max_{i \in [n]} h_i \times \left\lceil\frac{n}{s}\right\rceil}_{\text{"warm-up" phase}} + \max_{i \in [n]} h_i \times \frac{K}{\left(s \times \left\lceil\frac{n}{s}\right\rceil\right)}\right)$$

$$= \mathcal{O}\left(\max_{i \in [n]} h_i \times \frac{n}{s} + \max_{i \in [n]} h_i \times \left(\frac{nL\Delta}{s\varepsilon} + \frac{\sigma^2 L\Delta}{n\varepsilon^2}\right)\right)$$

$$= \mathcal{O}\left(\max_{i \in [n]} h_i \times \left(\frac{nL\Delta}{s\varepsilon} + \frac{\sigma^2 L\Delta}{n\varepsilon^2}\right)\right).$$

because $\frac{L\Delta}{\varepsilon} \geq \frac{1}{2}$ without loss of generality (if $\frac{L\Delta}{\varepsilon} < \frac{1}{2}$, then $x^0$ is an $\varepsilon$–stationary point). Finally,

$$\mathcal{O}\left(\max_{i \in [n]} h_i \times \left(\frac{nL\Delta}{s\varepsilon} + \frac{\sigma^2 L\Delta}{n\varepsilon^2}\right)\right) = \mathcal{O}\left(\max_{i \in [n]} h_i \times \left(\frac{L\Delta}{\varepsilon} + \frac{\sigma^2 L\Delta}{n\varepsilon^2}\right)\right)$$

due to the choice of $s$.

$\qquad \square$

### F.8 Dual-Process SGD

**Theorem F.14** (Dual-Process SGD). *Consider Theorem E.11 and its conditions. Under the $h_i$-fixed computation model* (20)*, the computational time complexity of* Dual-Process SGD *(Alg. 18) is*

$$\mathcal{O}\left(\min_{m \in [n]}\left[\left(\frac{1}{m}\sum_{i=1}^{m}\frac{1}{h_i}\right)^{-1}\left(\frac{L\Delta}{\varepsilon} + \frac{\sigma^2 L\Delta}{m\varepsilon^2}\right)\right]\right)$$

*with* $B = \max\left\{\left\lceil\frac{\sigma^2}{\varepsilon}\right\rceil, 1\right\}.$

*Proof.* The proof is essentially the same as the proof of Theorem F.6 since Dual-Process SGD is equivalent to Local SGD if the communication times are ignored. □

# G   TOTAL TIME COMPLEXITIES OF ALGORITHMS UNDER $(h, \tau)$-FIXED COMPUTATION MODEL

To compare the communication complexities and the total time complexities of the methods, we now assume that it takes $\tau$ seconds to send a vector from a worker to a parameter server and $\tau$ seconds to send a vector from the server to the workers in the centralized setting. Alternatively, it takes $\tau$ seconds to send a vector to all other workers in the decentralized setting. Moreover, we assume that all workers have the same computational performance: worker $i$ takes $h$ seconds to compute a single stochastic gradient for all $i \in [n]$. We refer to this as the $(h, \tau)$-*fixed computation model*.

Note that it is possible to assume that each worker has its own communication time bound $\tau_i$ and computation time bound $h_i$ and consider $(h_i, \tau_i)$-*fixed computation model* (Tyurin et al., 2024). However, for simplicity, we assume $\tau_i = \tau$ and $h_i = h$ for all $i \in [n]$. See Section I for a more general case $(h_i, \tau_i)$-*fixed computation model*.

## G.1   Rennala SGD

**Theorem G.1.** *Consider Theorem E.2 and its conditions. Under $(h, \tau)$-fixed computation model, the total time complexity of* Rennala SGD *(Alg. 4) is*

$$\mathcal{O}\left(\tau \times \frac{L\Delta}{\varepsilon} + h \times \left(\frac{L\Delta}{\varepsilon} + \frac{\sigma^2 L\Delta}{n\varepsilon^2}\right)\right)$$

*with $B = \max\left\{\left\lceil \frac{\sigma^2}{\varepsilon} \right\rceil, 1\right\}$.*

*Proof.* Note that the communication between of vectors happens every $B$ calculated stochastic gradients, which is equivalent to every $B$ updates of the main branch. Thus the total number of communications is

$$\mathcal{O}\left(\frac{K}{B}\right),$$

where $K = \Theta\left(\frac{BL\Delta}{\varepsilon} + \frac{\sigma^2 L\Delta}{\varepsilon^2}\right)$ due to Theorem E.2. The total communication complexity is

$$\mathcal{O}\left(\tau \times \frac{K}{B}\right) = \mathcal{O}\left(\frac{\tau}{B} \times \left(\frac{BL\Delta}{\varepsilon} + \frac{\sigma^2 L\Delta}{\varepsilon^2}\right)\right) = \mathcal{O}\left(\tau \times \frac{L\Delta}{\varepsilon}\right),$$

where we use the choice of $B$. It left to take into account the computation factor, which is the same as in Theorem F.1.

$$(22) = \mathcal{O}\left(h \times \left(\frac{L\Delta}{\varepsilon} + \frac{\sigma^2 L\Delta}{n\varepsilon^2}\right)\right)$$

under the $(h, \tau)$-fixed computation model. □

## G.2   Local SGD

**Theorem G.2.** *Consider Theorem E.3 and its conditions. Under $(h, \tau)$-fixed computation model, the total time complexity of* Local SGD *(Alg. 5) is*

$$\mathcal{O}\left(\tau \times \frac{L\Delta}{\varepsilon} + h \times \left(\frac{L\Delta}{\varepsilon} + \frac{\sigma^2 L\Delta}{n\varepsilon^2}\right)\right)$$

*with $B = \max\left\{\left\lceil \frac{\sigma^2}{\varepsilon} \right\rceil, 1\right\}$.*

*Proof.* The proof essentially the same as the proof of Theorem G.1. □

### G.3 Cycle SGD

**Theorem G.3.** *Consider Theorem E.5 and its conditions. Under $(h, \tau)$-fixed computation model, the total time complexity of* Cycle SGD *(Alg. 8) is*

$$\mathcal{O}\left(\tau \times \left(\frac{L\Delta}{\varepsilon} + \frac{\sigma^2 L\Delta}{n\varepsilon^2}\right) + h \times \left(\frac{L\Delta}{\varepsilon} + \frac{\sigma^2 L\Delta}{n\varepsilon^2}\right)\right)$$

*with $s = \min\left\{\max\left\{\left\lceil\frac{n^2\varepsilon}{\sigma^2}\right\rceil, 1\right\}, n\right\}$.*

*Proof.* Similarly to the proof of Theorem F.13, one can show that the total time complexity is

$$\mathcal{O}\left(\underbrace{(\tau + h) \times \left\lceil\frac{n}{s}\right\rceil}_{\text{``warm-up'' phase}} + (\tau + h) \times \frac{K}{\left(s \times \left\lceil\frac{n}{s}\right\rceil\right)}\right)$$

because every worker from group $s$ sends one vector $\sum_{j=1}^{M_i} \nabla f(z_i^j; \eta_i^j)$ to the server in the inner loop. Substituting the choice of $s$, one can get the final result. $\qquad\square$

### G.4 Async-Local SGD

**Theorem G.4.** *Consider Theorem E.6 and its conditions. Under $(h, \tau)$-fixed computation model, the total time complexity of* Async-Local SGD *(Alg. 9) is*

$$\mathcal{O}\left(\tau \times \frac{L\Delta}{\varepsilon} + h \times \left(\frac{L\Delta}{\varepsilon} + \frac{\sigma^2 L\Delta}{n\varepsilon^2}\right)\right)$$

*with $B = \max\left\{\left\lceil\frac{\sigma^2}{\varepsilon}\right\rceil, 1\right\}$ and $M = \max\left\{\left\lceil\frac{\sigma^2}{n\varepsilon}\right\rceil, 1\right\}$.*

*Proof.* Under $(h, \tau)$-fixed computation model, all workers send the sums of $M$ stochastic gradients at the same time. According to Theorem E.6, the server should collect

$$\mathcal{O}\left(\frac{(B + M - 1)L\Delta}{\varepsilon} + \frac{\sigma^2 L\Delta}{\varepsilon^2}\right) = \mathcal{O}\left(\frac{L\Delta}{\varepsilon} + \frac{\sigma^2 L\Delta}{\varepsilon^2}\right)$$

stochastic gradients, where the last equality due to the choice of $B$ and due to $B \geq M$. Since the workers work in parallel and have the equal performance, only $\Theta\left(\min\left\{\frac{B}{M}, n\right\}\right) = \Theta\left(\min\left\{\max\left\{1, \frac{\sigma^2}{M\varepsilon}\right\}, n\right\}\right)$ workers will participate in optimization. Thus, every worker, which participates in optimization, has to send

$$\mathcal{O}\left(\frac{L\Delta}{\min\left\{\frac{B}{M}, n\right\}\varepsilon} + \frac{\sigma^2 L\Delta}{\min\left\{\frac{B}{M}, n\right\}\varepsilon^2}\right) = \mathcal{O}\left(\frac{L\Delta}{\varepsilon} + \frac{\sigma^2 L\Delta}{n\varepsilon^2} + \frac{ML\Delta}{\varepsilon}\right)$$

stochastic gradients. Such a worker calculates $M$ stochastic gradients and only then sends the sum; thus, the maximum number of communications by one worker is

$$\mathcal{O}\left(\frac{L\Delta}{M\varepsilon} + \frac{\sigma^2 L\Delta}{Mn\varepsilon^2} + \frac{L\Delta}{\varepsilon}\right).$$

For every communication, the worker needs to send $M$ stochastic gradients, which takes $h$ seconds, and sends a sum, which takes $\tau$ seconds. Thus, the total time complexity is

$$\mathcal{O}\left((\tau + Mh)\left(\frac{L\Delta}{M\varepsilon} + \frac{\sigma^2 L\Delta}{Mn\varepsilon^2} + \frac{L\Delta}{\varepsilon}\right)\right). \tag{33}$$

Substituting the choice of $M$, we get the final result. $\qquad\square$

### G.5   Ringmaster ASGD

**Theorem G.5.** *Consider Theorem E.4 and its conditions. Under $(h, \tau)$-fixed computation model, the total time complexity of* Ringmaster ASGD *is*

$$\mathcal{O}\left(\tau \times \left(\frac{L\Delta}{\varepsilon} + \frac{\sigma^2 L\Delta}{n\varepsilon^2}\right) + h \times \left(\frac{L\Delta}{\varepsilon} + \frac{\sigma^2 L\Delta}{n\varepsilon^2}\right)\right) \tag{34}$$

*with $B = \max\left\{\left\lceil \frac{\sigma^2}{\varepsilon} \right\rceil, 1\right\}$.*

*Proof.* The proof repeats the proof of Theorem G.4. The only difference is that the workers send $M = 1$ stochastic gradients. Substituting $M = 1$ to (33), we get the final result. $\qquad\square$

*Remark* G.6. While (34) is only an upper bound, using the same steps as in the proof of Theorem G.4, one can easily show that the total time complexity of Ringmaster ASGD is lower bounded by

$$\Omega\left(\tau \times \left(\frac{L\Delta}{\varepsilon} + \frac{\sigma^2 L\Delta}{n\varepsilon^2}\right) + h \times \left(\frac{L\Delta}{\varepsilon} + \frac{\sigma^2 L\Delta}{n\varepsilon^2}\right)\right), \tag{35}$$

assuming that the iteration rate $\Theta\left(\frac{BL\Delta}{\varepsilon} + \frac{\sigma^2 L\Delta}{\varepsilon^2}\right)$ from Theorem E.4 is tight. As far as we know, this is the current state-of-the-art iteration rate of an Asynchronous SGD-like method (Maranjyan et al., 2025; Mishchenko et al., 2022; Koloskova et al., 2022; Cohen et al., 2021).

## H   COMPARISON BETWEEN OUR Local SGD AND THE CANONICAL Local SGD

In this section, we show that our version of Local SGD (Algorithm 5) achieves a better time complexity than the classical Local SGD. Although we focus in this section only on Local SGD, we expect similar improvements to extend to other new methods from Table 1. The purpose of this section is to highlight the tightness of the Birch SGD framework, using Local SGD as a case study.

In Section G.2, we prove that our version of Local SGD yields the total time complexity

$$\Theta\left(\tau\frac{L\Delta}{\varepsilon} + h\left(\frac{L\Delta}{\varepsilon} + \frac{\sigma^2 L\Delta}{n\varepsilon^2}\right)\right). \tag{36}$$

We now illustrate that this result is provably better than the best theoretical result for the canonical version of Local SGD (Algorithm 20) known to us.

---

**Algorithm 20** Local SGD (FedAvg) (McMahan et al., 2017)

**Require:** initial model $x^0$, step size $\gamma$, # of local steps $K$
1: **for** $k = 0, 1, 2, \dots$ **do**
2:     Broadcast $x^k$ to all workers
3:     **for** each worker $i \in \{1, \dots, n\}$ **in parallel do**
4:         $z_i^{k,0} = x^k$
5:         **for** $j = 0, \dots, K-1$ **do**
6:             $z_i^{k,j+1} = z_i^{k,j} - \gamma\nabla f(z_i^{k,j}; \eta_i^{k,j})$
7:         **end for**
8:     **end for**
9:     $x^{k+1} = \frac{1}{n}\sum_{i=1}^{n} z_i^{k,K}$
10: **end for**

---

To the best of our knowledge, the state-of-the-art analysis of Algorithm 20 in the nonconvex setting is provided by Koloskova et al. (2020); Luo et al. (2025). Under Assumptions 1.1, 1.2, and 1.3, with a proper $\gamma$, they establish the state-of-the-art iteration complexity

$$\Theta\left(\frac{L\Delta}{\varepsilon} + \frac{L\sigma^2\Delta}{nK\varepsilon^2} + \frac{L\sigma\Delta}{K^{1/2}\varepsilon^{3/2}}\right)$$

for finding an $\varepsilon$–stationary point for all $K \geq 1$. Next, under $(h, \tau)$-*fixed computation model*, this iteration complexity yields the time complexity

$$\bar{T} := \tau \left( \frac{L\Delta}{\varepsilon} + \frac{L\sigma^2\Delta}{nK\varepsilon^2} + \frac{L\sigma\Delta}{K^{\frac{1}{2}}\varepsilon^{\frac{3}{2}}} \right) + hK \left( \frac{L\Delta}{\varepsilon} + \frac{L\sigma^2\Delta}{nK\varepsilon^2} + \frac{L\sigma\Delta}{K^{\frac{1}{2}}\varepsilon^{\frac{3}{2}}} \right)$$

(up to constant factors) because in each iteration the workers communicate, which takes $\tau$ seconds, and each worker (in parallel) computes $K$ stochastic gradients, which takes $h \times K$ seconds. Ignoring non-negative terms,

$$\bar{T} \geq \tau \left( \frac{L\Delta}{\varepsilon} + \frac{L\sigma\Delta}{K^{\frac{1}{2}}\varepsilon^{\frac{3}{2}}} \right) + h \left( \frac{KL\Delta}{\varepsilon} + \frac{L\sigma^2\Delta}{n\varepsilon^2} + \frac{K^{\frac{1}{2}}L\sigma\Delta}{\varepsilon^{\frac{3}{2}}} \right)$$

and $\bar{T}$ is lower bounded by

$$\Theta \left( \sqrt{\tau h \frac{L^2\sigma^2\Delta^2}{\varepsilon^3}} + \tau \frac{L\Delta}{\varepsilon} + h \left( \frac{L\Delta}{\varepsilon} + \frac{\sigma^2 L\Delta}{n\varepsilon^2} \right) \right) \tag{37}$$

for all $K \geq 1$ due to the AM-GM inequality. Notice that (36) $\leq$ (37). However, (37) can be arbitrarily larger due to the first term. Indeed, for sufficiently large $n$, we have

$$(36) = \Theta \left( \tau \frac{L\Delta}{\varepsilon} + h \left( \frac{L\Delta}{\varepsilon} \right) \right),$$

while

$$(37) = \Theta \left( \sqrt{\tau h \frac{L^2\sigma^2\Delta^2}{\varepsilon^3}} + \tau \frac{L\Delta}{\varepsilon} + h \left( \frac{L\Delta}{\varepsilon} \right) \right).$$

Note that the latter expression has a $1/\varepsilon^{3/2}$ dependency, whereas our result has a $1/\varepsilon$ dependency. Thus, our result is provably tighter.

Note that we obtain the time complexity (36) for several other new methods, including Async-Local SGD, Async-Batch SGD, and Dual-Process SGD.

# I   Total Time Complexities of Algorithms under $(h_i, \tau_i)$-Fixed Computation Model

We now assume that each worker has its own communication time bound $\tau_i$ and computation time bound $h_i$ and consider *the $(h_i, \tau_i)$-fixed computation model* (Tyurin et al., 2024). It takes $\tau_i$ seconds to send a vector from worker $i$ to a parameter server and $\tau_i$ seconds to send a vector from the server to worker $i$ in the centralized setting. Alternatively, it takes $\tau_i$ seconds to send a vector to all other workers in the decentralized setting.

This setting reduces to $h_i$-fixed computation model when $\tau_i = 0$ for all $i \in [n]$, and reduces to $(h, \tau)$-fixed computation model when $h_i = h$ and $\tau_i = \tau$ for all $i \in [n]$. Without loss of generality, we assume that $\max\{h_1, \tau_1\} \leq \cdots \leq \max\{h_n, \tau_n\}$. Otherwise, the workers can be sorted according to these inequalities.

Notice that Rennala SGD, Local SGD, and Cycle SGD wait for the slowest worker by the designs. If $\max_{i \in [n]} \tau_i \to \infty$, then their total complexity tends to $\infty$. Thus, they are suboptimal under the $(h_i, \tau_i)$-fixed computation model. Ringmaster ASGD is suboptimal even under the $(h, \tau)$-fixed computation model. Async-Local SGD and Async-Batch SGD are optimal under the $(h, \tau)$-fixed computation model, but we conjecture that they are suboptimal under the $(h_i, \tau_i)$-fixed computation model.

We now prove that Dual-Process SGD is optimal under the $(h_i, \tau_i)$-fixed computation model within the family of methods that communicate either with a server (centralized setting) or with each other (decentralized setting).

### I.1 Dual-Process SGD

**Theorem I.1** (Dual-Process SGD). *Consider Theorem E.11 and its conditions. Under the $(h_i, \tau_i)$-fixed computation model, the total time complexity of* Dual-Process SGD *(Alg. 18) is*

$$\mathcal{O}\left(\min_{m\in[n]}\left[\max\left\{\max\{h_m, \tau_m\}, \left(\sum_{i=1}^{m}\frac{1}{h_i}\right)^{-1}\frac{\sigma^2}{\varepsilon}\right\}\right]\frac{L\Delta}{\varepsilon}\right)$$

*with $B = \max\left\{\left\lceil\frac{\sigma^2}{\varepsilon}\right\rceil, 1\right\}$.*

This complexity is optimal for distributed methods without compression communicating with a server (centralized setting) or with each other (decentralized setting) (Tyurin et al., 2024; Tyurin & Richtárik, 2024). Notice that it is robust to slow communications. Indeed, if $\tau_n \to \infty$, then this complexity will ignore worker $n$ due to the $\min_{m\in[n]}$ operation.

**Lemma I.2.** *Let us define*

$$T(B) := 4\min_{m\in[n]}\left[\max\left\{\max\{h_m, \tau_m\}, \left(\sum_{i=1}^{m}\frac{1}{h_i}\right)^{-1}B\right\}\right]. \tag{38}$$

*Under the $(h_i, \tau_i)$-fixed computation model, the time required to calculate $x^1, \ldots, x^B$ of the main branch is at most $3T(B)$ seconds, the time required to calculate $x^{B+1}, \ldots, x^{2B}$ is at most $3T(B)$ seconds, and so on.*

*Proof.* Notice that

$$T(B) = 4\max\left\{\max\{h_{m^*}, \tau_{m^*}\}, \left(\sum_{i=1}^{m^*}\frac{1}{h_i}\right)^{-1}B\right\}.$$

for some $m^* \in [n]$. The idea is similar as in Lemmas F.2 and F.7. All workers calculate stochastic gradients in parallel. For all $t \geq 0$, after $t$ seconds the first $m^*$ workers can calculate at at least

$$\sum_{i=1}^{m^*}\max\left\{\left\lfloor\frac{t}{h_i}\right\rfloor - 1, 0\right\}, \tag{39}$$

stochastic gradients, where we subtract 1 because at most one stochastic gradient can be ignored. Substituting $T(B)$ into (39), we have

$$\sum_{i=1}^{m^*}\max\left\{\left\lfloor\frac{T(B)}{h_i}\right\rfloor - 1, 0\right\} \geq \sum_{i=1}^{m^*}\left\lfloor\frac{T(B)}{h_i}\right\rfloor - m^* \geq \sum_{i=1}^{m^*}\frac{T(B)}{h_i} - 2m^*.$$

Recall that $\max\{h_1, \tau_1\} \leq \cdots \leq \max\{h_{m^*}, \tau_{m^*}\}$. Thus,

$$T(B) \geq 2\max\{h_{m^*}, \tau_{m^*}\} + 2\left(\sum_{i=1}^{m^*}\frac{1}{h_i}\right)^{-1}B \geq 2h_i + 2\left(\sum_{i=1}^{m^*}\frac{1}{h_i}\right)^{-1}B$$

for all $i \leq m^*$, and

$$\sum_{i=1}^{m^*}\max\left\{\left\lfloor\frac{T(B)}{h_i}\right\rfloor - 1, 0\right\} \geq \sum_{i=1}^{m^*}\left(2 + \frac{2}{h_i}\left(\sum_{i=1}^{m^*}\frac{1}{h_i}\right)^{-1}B\right) - 2m^* \geq B.$$

Thus, by the time $T(B)$, the first $m^*$ workers can calculate $B$ stochastic gradients.

Next, we need to estimate the communication time. It takes at most $\max_{i\in[m^*]}\tau_i \leq \max\{h_{m^*}, \tau_{m^*}\} \leq T(B)$ seconds to receive a vector from the server (in the decentralized setting, we do not account this time). Similarly, it takes at most $\max_{i\in[m^*]}\tau_i \leq T(B)$ seconds to send a vector to the server (in the decentralized setting, to send a vector to other workers). Thus, one round in Alg 18 takes at most $3 \times T(B)$ seconds, which is equivalent to calculating $x^1, \ldots, x^B$ of the main branch. The same argument can be applied to the next $B$ point of the main branch, and so on. $\quad\square$

*Proof of Theorem I.1.* The proof is similar to the proof of Lemma F.2. Due to Theorem E.2, we know that $\frac{1}{K} \sum_{k=0}^{K-1} \mathbb{E}\left[\|\nabla f(x^k)\|^2\right] \leq \varepsilon$ for

$$K = \left\lceil \frac{4BL\Delta}{\varepsilon} + \frac{8\sigma^2 L\Delta}{\varepsilon^2} \right\rceil.$$

Using Lemma I.2, the total time to find an $\varepsilon$–stationary point is

$$\mathcal{O}\left(T(B) \times \frac{K}{B}\right) = \mathcal{O}\left(T(B) \times \left(\frac{L\Delta}{\varepsilon} + \frac{\sigma^2 L\Delta}{B\varepsilon^2}\right)\right).$$

Using the choice of $B$,

$$\mathcal{O}\left(T(B) \times \frac{K}{B}\right) = \mathcal{O}\left(\min_{m\in[n]}\left[\max\left\{\max\{h_m, \tau_m\}, \left(\sum_{i=1}^m \frac{1}{h_i}\right)^{-1} \frac{\sigma^2}{\varepsilon}\right\}\right] \times \frac{L\Delta}{\varepsilon}\right).$$

$\square$

## J  PERFORMANCE OF Rennala SGD AND Ringmaster ASGD ON A QUADRATIC FUNCTION

In this section, we formally prove that the convergence of Ringmaster ASGD can be provably faster than Rennala SGD due to the frequent model updates.

**Theorem J.1.** *Consider* Rennala SGD *(Alg. 4) and* Ringmaster ASGD *(Alg. 7) with the optimal parameters $B$ from Sec. F. Then, there exists a $\mu$-strongly convex function and corresponding stochastic gradients that satisfy Assumptions 1.1, 1.2, and 1.3 with $\sigma^2/\varepsilon \geq n$, such that* Rennala SGD, *with **any step size** $\gamma$, requires*

$$\widetilde{\Theta}\left(\frac{\sigma^2}{n\varepsilon} \times h \times \frac{L}{\mu}\right)$$

*seconds to find $\varepsilon$-stationary point under the $h_i$-fixed computation model (20) with $h_i = h$ for all $i \in [n]$. At the same time, there **exists a step size** for* Ringmaster ASGD *such that it requires at most*

$$\widetilde{\mathcal{O}}\left(h \times \frac{L}{\mu}\right)$$

*seconds to find $\varepsilon$-stationary point.*

*Proof.* In this construction, we take $f : \mathbb{R}^2 \to \mathbb{R}$ such that

$$f(w \equiv (x, y)) = \frac{\mu x^2}{2} + \frac{L y^2}{2} \tag{40}$$

for all $x, y \in \mathbb{R}$. Moreover, we assume that the stochastic gradients $\nabla f(w; \xi)$ are equal to the true gradient $\nabla f(w)$; thus, there is no randomness. Note that *a priori*, both methods do not have this information and therefore must choose $B = \Theta\left(\frac{\sigma^2}{\varepsilon}\right)$, even though the effective variance is zero.

By the design of Rennala SGD, its algorithm is equivalent to the following steps:

$$w^{t+1} = w^t - \gamma B \nabla f(w^t) \tag{41}$$

because the workers calculate $B$ gradients in every global round. Each round takes

$$\Theta\left(h \times \frac{B}{n}\right) = \Theta\left(h\frac{\sigma^2}{n\varepsilon}\right)$$

seconds, because the workers have the computation speed $h$ and $B = \Theta\left(\frac{\sigma^2}{\varepsilon}\right)$ in Theorem F.1.

It is well known that the sequence (41) requires

$$\widetilde{\Theta}\left(\frac{L}{\mu}\right)$$

iterations (up to logarithmic factors) to find an $\varepsilon$-solution or $\varepsilon$-stationary point with the function (40), even when the step size $\gamma$ can be tuned. Thus, the computational time complexity of Rennala SGD is

$$\widetilde{\Theta}\left(\frac{\sigma^2}{n\varepsilon} \times h \times \frac{L}{\mu}\right)$$

seconds.

Consider now the steps of Ringmaster ASGD w.r.t. the first argument $x$. In this algorithm, we take $\gamma = \frac{1}{2Ln}$. In the case when the computation time is equal for all workers, the first $n$ steps are

$$x^1 = x^0 - \gamma\mu x^0 = (1 - \gamma\mu)x^0,$$
$$x^2 = x^1 - \gamma\mu x^0 = (1 - 2\gamma\mu)x^0,$$
$$\vdots$$
$$x^n = x^{n-1} - \gamma\mu x^0 = (1 - n\gamma\mu)x^0,$$

because the workers start calculating at the same point and return the gradients at the same time. Notice that $0 \le x^n \le \cdots \le x^2 \le x^1$. Then, the first worker starts calculating at $x^1$, the seconds worker starts calculating at $x^2$, and so on. Therefore, the next steps are

$$
\begin{aligned}
x^{n+1} &= x^n - \gamma\mu x^1 \\
&= (1 - n\gamma\mu)x^0 - \gamma\mu(1 - \gamma\mu)x^0 = (1 - (n+1)\gamma\mu + \gamma^2\mu^2)x^0, \\
x^{n+2} &= x^{n+1} - \gamma\mu x^2 \\
&= (1 - (n+1)\gamma\mu + \gamma^2\mu^2)x^0 - \gamma\mu(1 - 2\gamma\mu)x^0 = (1 - (n+2)\gamma\mu + 3\gamma^2\mu^2)x^0, \\
&\vdots \\
x^{2n} &= x^{2n-1} - \gamma\mu x^n = \left(1 - 2n\gamma\mu + \frac{n(n+1)}{2}\gamma^2\mu^2\right)x^0 \le (1 - n\gamma\mu)^2 x^0.
\end{aligned}
$$

For $\gamma = 1/2Ln$, we have $0 \le x^{2n} \le \cdots \le x^{n+1} \le x^n \le \cdots \le x^2 \le x^1$. Using mathematical induction, assume that $0 \le x^{kn} \le \cdots \le x^1$ for some $k \ge 1$ and $x^{pn} \le (1 - n\gamma\mu)^p x^0$ for all $p \le k$, which is true for $k = 2$ (base case). We now prove it for $k + 1$. Ringmaster ASGD calculates $x^{kn+1}$ as follows:

$$
x^{kn+1} = x^{kn} - \gamma\mu x^{(k-1)n+1},
$$

which ensures that $x^{kn+1} \le (1 - \gamma\mu)x^{kn} \le x^{kn}$ and $x^{kn+1} \ge x^{(k-1)n+1} - \gamma\mu x^{(k-1)n+1} \ge 0$ for $\gamma = 1/2Ln$. We can continue:

$$
x^{kn+2} = x^{kn+1} - \gamma\mu x^{(k-1)n+2},
$$

which ensures that

$$
x^{kn+2} \le x^{kn+1} - \gamma\mu x^{kn} \le (1 - \gamma\mu)x^{kn} - \gamma\mu x^{kn} = (1 - 2\gamma\mu)x^{kn}
$$

and $x^{kn+2} \ge x^{(k-1)n+2} - \gamma\mu x^{(k-1)n+2}$. Continuing, we have

$$
x^{(k+1)n} = x^{(k+1)n-1} - \gamma\mu x^{kn}.
$$

One can show that

$$
x^{(k+1)n} \le (1 - (n-1)\gamma\mu)x^{kn} - \gamma\mu x^{kn} \le (1 - n\gamma\mu)x^{kn},
$$

and $x^{(k+1)n} \ge x^{kn} - \gamma\mu x^{kn} \ge 0$. We have proved the next case, $k + 1$, of the mathematical induction.

Thus, the sequence $\{x^{pn}\}_{p\ge2}$ monotonically decreases with the rate

$$
x^{pn} \le (1 - n\gamma\mu)^p x^0.
$$

Using the same reasoning, we one can show the similar result holds for the second argument $y$ of the function but with $L$ instead of $\mu$.

Recall that it takes $h$ seconds to calculate $x^n$ because $n$ workers work in parallel, it takes $h$ seconds to calculate $x^{2n}$, and so on. Thus, the computational time complexity of Ringmaster ASGD is

$$
\widetilde{\mathcal{O}}\left(h \times \frac{L}{\mu}\right)
$$

with step size $\gamma = \frac{1}{2Ln}$. $\qquad\square$

