# OpenReview forum: "Birch SGD: A Tree Graph Framework for Local and Asynchronous SGD Methods"
_ICLR.cc/2026/Conference — ICLR 2026 Poster_

### Official Review · Reviewer_s9uQ · 2025-11-01

**Soundness:** 3
**Presentation:** 3
**Contribution:** 3
**Rating:** 6
**Confidence:** 4

**Summary:**

The paper proposes a unified framework to represent various distributed SGD algorithms in the form of a computation tree. Each branch of the tree captures different versions of the model (potentially at different workers) and how they are updated with fresh or stale gradients. Some of the algorithms that are subsumed in the framework include local SGD, asynchronous SGD, and several other variants of these methods. The paper provides a unified analysis that matches previous results for these algorithms and sometimes gives improved analysis and optimal communication/computation complexity bounds.

**Strengths:**

- The elegance of the unified framework and analysis is a clear strength of this paper. Even if the bounds are not improved or new algorithms are not proposed, a unified representation is valuable for researchers working in this field to put algorithms in perspective.
- The addition of experimental results to complement the theory is well appreciated. Since this is a theory-focused paper, I do not have high expectations about the size and scale of experimental results. Nevertheless, I appreciated that the authors included these comparisons.

**Weaknesses:**

- The paper introduces eight new local SGD methods. While presented as a strength, this seems excessive and might confuse system designers choosing an algorithm. The insights section doesn't clearly guide which method to use under different communication and computation constraints. The comparisons in Section 3 are somewhat informative but could be clearer. Can you identify specific regimes—such as (1) communication-delay-bounded, (2) compute-bounded, and (3) bandwidth-bounded—and recommend the best algorithm for each?

- While the writing is generally clear, it needs improvement in organization and conciseness. I suggest spending more time explaining the computation graph structure, specifically how the number of workers $M$ relates to the tree branches. Currently, the paper gives minimal explanation of this aspect (see my question below) and jumps too quickly to theoretical results and new algorithm proposals.

- The writing comes across as overselling and condescending. I recommend toning down some claims to avoid alienating readers. For example, "virtually all local SGD algorithms" sounds too strong—please clarify which algorithms are not covered. The paper frequently uses adjectives and superlatives that add little information.

- The results apply only to homogeneous data distributions across workers and would not hold in heterogeneous data settings (like federated learning). I don't consider this a major weakness since the paper provides a valuable unified framework and analysis for the homogeneous setting. However, I would like the authors to clearly acknowledge this limitation and provide some ideas on whether and how the framework could be extended to heterogeneous data settings.

- The literature review is mostly thorough but omits other papers that provide unified analyses of different SGD algorithms, such as https://arxiv.org/pdf/1808.07576, https://arxiv.org/abs/2007.07481, https://arxiv.org/abs/2011.02828, https://arxiv.org/abs/2207.03730. I suggest broadening the literature survey to cover more key papers in the field rather than limiting references to papers from a few selected authors.

**Questions:**

- In local SGD and its variants, gradients are aggregated using AllReduce to update all models. Where does this step appear in the computation graph? At first glance, each edge seems to represent a single stochastic gradient (per-sample or mini-batch) rather than an average of gradients from multiple workers. However, upon closer reading and examining Figure 4, it appears that edges can represent averages of multiple gradients computed at different model versions. Is the gradient variance (now reduced due to taking the average) captured by $\eta$? How does the aggregation step of local SGD occur in the computation graph?

- Related to the above question, where does the number of workers appear in the convergence bound? Which term(s) explicitly depend on the number of workers?

- The framework subsumes asynchronous SGD and its variants. A crucial aspect affecting these algorithms and their analysis is gradient staleness—the mismatch between the model index at which the gradient is computed and the model updated using this gradient. I initially thought $R$ would capture a bound on this staleness, but I don't understand how it does this based on the distance metric dist(x,z) defined in the paper. Please clarify.

- How does this paper relate to https://arxiv.org/abs/2207.03730? That paper also models local SGD variants using a computation graphs with edges representing stochastic gradient computations?

---

> ### Author Response · Authors · 2025-11-18
> **Official Comment by Authors (Part 1/2)**
>
> Thank you for the review! Let us respond to the comments.
>
> > The paper introduces eight new local SGD methods. While presented as a strength, this seems excessive and might confuse system designers choosing an algorithm. The insights section doesn't clearly guide which method to use under different communication and computation constraints. The comparisons in Section 3 are somewhat informative but could be clearer. Can you identify specific regimes—such as (1) communication-delay-bounded, (2) compute-bounded, and (3) bandwidth-bounded—and recommend the best algorithm for each?
>
> This is precisely why we prepared Table 1. In Table 1, for each method, we indicate whether it has the optimal computational complexity, whether it achieves the optimal complexity when computation and communication times are equal, whether it supports AllReduce, and so on. The green tick symbol indicates a favorable property in the corresponding metric and whether the corresponding method is advantageous for that metric. For instance, if one wants AllReduce support, one can choose either Rennala SGD or Local SGD. If one needs the optimal total complexity with both heterogeneous computational and communication times, then one should choose Dual-Process SGD (which unfortunately does not support AllReduce). In summary, select your requirements, find the methods in Table 1 that satisfy them (those with green ticks), and choose any of these methods.
>
> >  In local SGD and its variants, gradients are aggregated using AllReduce to update all models. Where does this step appear in the computation graph? At first glance, each edge seems to represent a single stochastic gradient (per-sample or mini-batch) rather than an average of gradients from multiple workers. However, upon closer reading and examining Figure 4, it appears that edges can represent averages of multiple gradients computed at different model versions. Is the gradient variance (now reduced due to taking the average) captured by ? How does the aggregation step of local SGD occur in the computation graph?
>
> Please consider the updated PDF, where we clarified the steps to construct the graph of Local SGD. Please see the example (blue text) in Section 2 and the improved Figure 4 for Local SGD. We have tried to explain every step, and hopefully everything is now clear.
>
> >   While the writing is generally clear, it needs improvement in organization and conciseness. I suggest spending more time explaining the computation graph structure, specifically how the number of workers relates to the tree branches. Currently, the paper gives minimal explanation of this aspect (see my question below) and jumps too quickly to theoretical results and new algorithm proposals.
>
> >   Related to the above question, where does the number of workers appear in the convergence bound? Which term(s) explicitly depend on the number of workers?
>
> In the main Theorem 2.4, there is no dependence on the number of workers, and there should not be. Notice that any computational graph can be constructed with any number of workers, including just one worker. For instance, the graph in Figure 4 can be constructed by one worker if this worker takes the first set of local steps with $M_1 = 2,$ then the second set of local steps with $M_2 = 2,$ and then aggregates the calculated stochastic gradients. In this way, one worker generates the same computational graph as when it is constructed by two workers. Is there any practical sense in running Local SGD on only one worker? Probably not. But it is possible.
>
> The dependence appears when we start proving the time complexities of constructing such graphs (see Table 1). For instance, in Theorem F.6, we prove the time required to execute Local SGD, or equivalently, the time required to construct the computational graph. This time complexity, of course, depends on $n.$
>
> >  The writing comes across as overselling and condescending. I recommend toning down some claims to avoid alienating readers. For example, "virtually all local SGD algorithms" sounds too strong—please clarify which algorithms are not covered. The paper frequently uses adjectives and superlatives that add little information.
>
> Agree. We have described the limitations of our approach in Section 4 and slightly toned down the claims. We acknowledge that it would be interesting to extend our framework to methods with preconditioning (e.g., Adam) and non-Euclidean geometry (e.g. Muon).

---

> > ### Author Response · Authors · 2025-11-18
> > **Official Comment by Authors (Part 2/2)**
> >
> > >  The results apply only to homogeneous data distributions across workers and would not hold in heterogeneous data settings (like federated learning). I don't consider this a major weakness since the paper provides a valuable unified framework and analysis for the homogeneous setting. However, I would like the authors to clearly acknowledge this limitation and provide some ideas on whether and how the framework could be extended to heterogeneous data settings.
> >
> > Also agree. We extended the discussion in Section 4, where we say non-i.i.d. scenarios are also important, and it might be possible that we have to introduce extra assumptions for the smooth development of a new framework for the i.i.d. setting.
> >
> > >  The literature review is mostly thorough but omits other papers that provide unified analyses of different SGD algorithms, such as https://arxiv.org/pdf/1808.07576, https://arxiv.org/abs/2007.07481, https://arxiv.org/abs/2011.02828, https://arxiv.org/abs/2207.03730. I suggest broadening the literature survey to cover more key papers in the field rather than limiting references to papers from a few selected authors.
> >
> > Thank you for the references. We've extended our literature overview in Section A.3 (see the updated PDF).
> >
> > >  The framework subsumes asynchronous SGD and its variants. A crucial aspect affecting these algorithms and their analysis is gradient staleness—the mismatch between the model index at which the gradient is computed and the model updated using this gradient. I initially thought  would capture a bound on this staleness, but I don't understand how it does this based on the distance metric dist(x,z) defined in the paper. Please clarify.
> >
> > The metric $dist(x,z)$ generalizes your intuition. Please consider "Ringmaster ASGD" at the beginning of Section 3. For this method, in Theorem E.4, we prove that $\textnormal{dist}(x^k, z^k) = \textnormal{dist}(x^{k}, x^{k-\delta^k}) = \delta^k$, where $\delta^k$ is the staleness, exactly what you expected! Notice that $\textnormal{dist}(x, z)$ generalizes gradient staleness for other methods like Local SGD, Rennala SGD, and so forth. In other words, $\delta^k$ is a difference between indices (as you understand). At the same time, the metric $dist(x,z)$ finds the common ancestor $w$ of $x$ and $z$, computes the differences between the indices of $w$ and $x$ and of $w$ and $z$, respectively, and takes the maximum of them. This is a natural generalization of the standard staleness.
> >
> > >  How does this paper relate to https://arxiv.org/abs/2207.03730? That paper also models local SGD variants using a computation graphs with edges representing stochastic gradient computations?
> >
> > The paper https://arxiv.org/abs/2207.03730 also considers graphs, but they use them in an entirely different way. In their case, nodes correspond to devices and iterates of data samples, while in our case, nodes represent only the iterates of algorithms. In their case, edges represent both communication and computation links, while in our case, edges indicate how one point is calculated from another. These are two different, orthogonal approaches. For instance, in their case, the number of nodes is fixed from the beginning since every node corresponds to one sample or worker. In our case, the graph evolves with every iteration (see Figures 1 and 4 in the updated PDF), and the iterates do not correspond to samples. Moreover, they consider the finite-sum setting, while we focus on the stochastic/online setting. We have clarified this in Section A.3.
> >
> > ---
> >
> > **Thank you once again for your review, which is very informative and positively impactful for our paper. We’ve improved the quality of the paper thanks to you, and we hope that you will increase the score. If you have more questions, please let us know.**

---

> > > ### Comment · Reviewer_s9uQ · 2025-11-24
> > > **Thanks for the clarification**
> > >
> > > I appreciate the authors' effort to address my comments.

---

### Official Review · Reviewer_ZST5 · 2025-11-01

**Soundness:** 4
**Presentation:** 4
**Contribution:** 3
**Rating:** 8
**Confidence:** 4

**Summary:**

The paper proposes a unified theoretical framework for analyzing stochastic gradient based optimization method. The tree visualization is interesting. By using Lipschitz smoothness to bound how wrong the stale gradients can be the authors propose a unified analysis for convergence guarantees.

**Strengths:**

The unified framework provides a helpful tool to analyze many different implementations of SGD. The paper is clearly written with many helpful visualizations. The convergence guarantee theory is based on topology alone and seems quite powerful.

**Weaknesses:**

How tight is Theorem 2.4? It would be interesting to include some numerical experiment where R grows over time and show that it fails. Or include some correlated noise. I suspect that would be less harmful.

It seems quite conservative to use the upper bound on R rather than an average distance. I suspect most of the proof will go through using average distance but changing to a weaker convergence guarantee (in expectation maybe).

Some of the new methods (cycle SGD) was not tested in numerical studies. When they do get added, please also include the total time, as some of them need to keep track of/communicate additional meta data/information and might take time in distributed settings. Though this part could be tricky to simulate.

**Questions:**

What about non-iid data? Could we add a measure of heterogeneity in the analysis?

---

> ### Author Response · Authors · 2025-11-18
>
> Thank you for your positive review! Let us respond to the questions:
>
> > How tight is Theorem 2.4? It would be interesting to include some numerical experiment where R grows over time and show that it fails. Or include some correlated noise. I suspect that would be less harmful.
>
> The tightness of Theorem 2.4 is justified by the fact that it leads to optimal time complexities (see Table 1). In all theorems where we prove time complexities, Theorem 2.4 plays the main crucial role. If Theorem 2.4 were not tight, then Theorem 2.4 and the subsequent time complexities could be improved, which is impossible due to the optimality of the time complexities.
>
> In Section C and Section C.6, we have prepared additional numerical experiments to see what happens when we start increasing $R \simeq B$. We observe exactly what we expect in Section 4: once we increase $R \simeq B$, the time complexity increases as well.
>
> > It seems quite conservative to use the upper bound on R rather than an average distance. I suspect most of the proof will go through using average distance but changing to a weaker convergence guarantee (in expectation maybe).
>
> This is a good future research question. It might be possible that instead of taking $R = \sup_{k \geq 0} \textnormal{dist}(x^k, z^k)$, we could replace the $\sup$ with an average. However, as noted in the previous response, the choice of $R$ is tight in the sense that it is sufficient to obtain optimal time complexities.
>
> > Some of the new methods (cycle SGD) was not tested in numerical studies. When they do get added, please also include the total time, as some of them need to keep track of/communicate additional meta data/information and might take time in distributed settings. Though this part could be tricky to simulate.
>
> In Section C.5, we have prepared an additional experiment to verify the robustness of Cycle SGD to peak bandwidth compared to other methods. This experiment concurs with our theoretical predictions from the main part of the paper.
>
> > What about non-iid data? Could we add a measure of heterogeneity in the analysis?
>
> Our primary focus is on training large-scale language and vision models, where the i.i.d. assumption is usually appropriate. In these settings, all devices typically sample from a common dataset (e.g., such as ImageNet or the Internet), which supports the i.i.d. formulation. Extending our analysis to non-i.i.d. scenarios is an important and challenging direction for future work. We have described this in Section 4 and also added possible assumptions that would allow us to extend the framework to the non-i.i.d. setting.
>
> ---
>
> Thank you once again! If you have more questions, please let us know.

---

### Official Review · Reviewer_sATR · 2025-11-04

**Soundness:** 2
**Presentation:** 2
**Contribution:** 3
**Rating:** 4
**Confidence:** 2

**Summary:**

This paper presents a unified framework to frame, analyze, and design distributed SGD algorithm. Basically, the framework models any distributed SGD algorithm as a weighted directed tree. The convergence properties of the algorithm only rely on the geometric properties of the directed tree. The authors provide a unified analysis and discussed many concrete examples, including both old and new distributed SGD variants.

**Strengths:**

- The idea of framing distributed SGD algorithms as a directed tree is very novel and interesting.
- The unified convergence analysis is also solid and clean.

**Weaknesses:**

The writing is not super clear. From algorithm 1, it is hard to tell why local SGD can be treated as an instance of birch SGD. I did not see any part of the algorithm to handle the communication (ie averaging gradients or parameters).

**Questions:**

1. From algorithm 1, it is hard to tell why local SGD can be treated as an instance of birch SGD. I did not see any part of the algorithm to handle the communication (ie averaging gradients or parameters).
2. The convergence rate is established in terms of the main branch? If so, then K is not the gradient computation times as in previous literature. As shown in figure 10, the main branch only has 5 iterates but there are total 10 gradient computations.
3. In table 1, why does local SGD listed as new? It is not a new algorithm.

---

> ### Author Response · Authors · 2025-11-18
>
> Thank you for the review! We now respond to the comments and questions.
>
> > The writing is not super clear. From algorithm 1, it is hard to tell why local SGD can be treated as an instance of birch SGD. I did not see any part of the algorithm to handle the communication (ie averaging gradients or parameters).
>
> Please take a look at the updated PDF, where we clarified how the computational graph of Local SGD is constructed in Section 2 (blue text) and in Figure 4. We’ve added a detailed example in the section and improved Figure 4 to illustrate each step of the process. If anything is still unclear, feel free to let us know.
>
> >  The convergence rate is established in terms of the main branch? If so, then K is not the gradient computation times as in previous literature. As shown in figure 10, the main branch only has 5 iterates but there are total 10 gradient computations.
>
> Yes, the rate in Theorem 2.4 is expressed in terms of the length of the main branch. However, this is not a weakness of our paper or results. The reviewer might think that we do not take into account all the computations, but we do once we start deriving the *time complexities* of the algorithms. In Line 323, we explain that the iteration complexity $K$ in Theorem 2.4 does not reflect the true wall-clock performance; it serves as an intermediate result used to derive the time complexities. After Theorem 2.4, we derive the time complexity required to construct the trees, and this is where we start accounting for the computed gradients in all branches.
>
> For instance, consider Rennala SGD or Local SGD. We show that the iteration rate, that is, the length of the main branch that we must take, is
> $O\left(\frac{B L \Delta}{\varepsilon} + \frac{\sigma^2 L \Delta}{\varepsilon^2}\right)$,
> where $B$ is a parameter. In the case of Rennala SGD, $B$ is the batch size; in the case of Local SGD, $B$ is the total number of local steps. Given this result, we can derive the time complexities in Theorem F.1 and Theorem F.6, which capture all gradient computations: what remains is to determine the time needed to collect $B$ samples or perform $B$ local steps, which is done in those theorems.
>
> In other words, Theorem~2.4 is an intermediate but important result that determines the required length of the main branch. Then, it remains only to derive the time needed to "build" the tree with a main branch of this size $O\left(\frac{B L \Delta}{\varepsilon} + \frac{\sigma^2 L \Delta}{\varepsilon^2}\right).$
>
> >  In table 1, why does local SGD listed as new? It is not a new algorithm.
>
> We acknowledge this. This is why we added the footnote (f), where we say that we recognize that Local SGD is well-known in the literature. However, what makes our version novel is the better time complexity compared to the classical version, the stopping criterion in Alg. 5, and the analysis in Sec. E.3, F, and G, which leads to the optimal computational time complexities.
>
> Once again, of course, the concept of Local SGD is not new, but our contributions, to the best of our knowledge, are new.
>
> ---
>
> **We hope that we have addressed all weaknesses and questions, and that the reviewer will reconsider the score. Notice that we have clarified the comment in the weaknesses section (see the example in Section 2 and Figure 4 in the updated PDF). If you have more questions, please let us know.**

---

> > ### Author Response · Authors · 2025-11-27
> >
> > Dear Reviewer sATR,
> >
> > Thank you for taking the time to review our response. We believe that we have addressed all the comments and updated the PDF accordingly. The presentation of Local SGD is improved in Section 2 and Figure 4. We would be glad to clarify any further questions. Thank you!
> >
> > Best regards,
> > Authors

---

### Author Response · Authors · 2025-12-01
**Public Summary Comment by Authors**

Dear AC and Reviewers,

Thank you for your time and work. We believe that we have addressed all the weaknesses and comments. Please note that all the required changes are already included in the updated PDF (see blue text). We added the missing references, discussion, and experiments according to the reviewers’ comments. Overall, Reviewers ZST5 and s9uQ are positive about our work and recommend acceptance.

---

Reviewer sATR gave us "Rating: 4" with "Confidence: 2" and outlined one weakness:

> The writing is not super clear. From algorithm 1, it is hard to tell why local SGD can be treated as an instance of birch SGD. I did not see any part of the algorithm to handle the communication (ie averaging gradients or parameters).

Reviewer s9uQ also asked a similar question:

> In local SGD and its variants, gradients are aggregated using AllReduce to update all models. Where does this step appear in the computation graph? At first glance, each edge seems to represent a single stochastic gradient (per-sample or mini-batch) rather than an average of gradients from multiple workers. However, upon closer reading and examining Figure 4, it appears that edges can represent averages of multiple gradients computed at different model versions. Is the gradient variance (now reduced due to taking the average) captured by ? How does the aggregation step of local SGD occur in the computation graph?

As a response to the weakness and the question, we have added a detailed example in Section 2 (blue text) and improved Figure 4 to illustrate each step of the process. Please take a look at the updated PDF, where we clarified how the computational graph of Local SGD is constructed. We believe that this important modification addresses and resolves the weakness, and everything should now be clear.

---

Thank you once again!

Authors

---

### Meta-Review · Area_Chair_kUSz · 2026-01-06

**Summary:**

The reviewers generally acknowledge the paper's core contribution: a novel unified framework for representing distributed SGD methods as computation trees. Reviewer s9uQ (score: 6) captures the consensus well, noting "The elegance of the unified framework and analysis is a clear strength of this paper. Even if the bounds are not improved or new algorithms are not proposed, a unified representation is valuable for researchers working in this field to put algorithms in perspective." The main concerns raised were presentation clarity (both sATR and s9uQ asked how gradient aggregation appears in the computation graph), practical guidance for method selection, overstated claims about scope, and limitation to homogeneous data distributions. Reviewer ZST5 (score: 8) was positive overall, finding the framework "quite powerful" with "helpful visualizations." Reviewer sATR (score: 4) struggled with presentation but had low confidence in their assessment.

**Reviewer Concerns:**

The authors adequately addressed the primary technical concerns. The presentation issues raised by sATR and s9uQ regarding how Local SGD maps to the framework were resolved through detailed additions to Section 2 and improvements to Figure 4, showing how aggregated gradients appear as edge weights in the computation tree. As s9uQ noted, "upon closer reading and examining Figure 4, it appears that edges can represent averages of multiple gradients computed at different model versions," and the authors' clarifications make this explicit. The concern about iteration complexity K not reflecting all gradient computations was addressed by explaining that Theorem 2.4 is an intermediate result with full time complexities derived in subsequent theorems. ZST5's questions about tightness were answered by showing the bounds lead to optimal time complexities, and requested experiments for Cycle SGD and sensitivity analysis were added. The authors also broadened the literature survey as requested by s9uQ.

The remaining concerns are acknowledged limitations rather than flaws. The restriction to homogeneous i.i.d. settings is clearly stated, and as s9uQ notes, "I don't consider this a major weakness since the paper provides a valuable unified framework and analysis for the homogeneous setting." The authors added discussion in Section 4 about potential extensions to heterogeneous settings. Regarding the eight methods and practical guidance, Table 1 provides systematic comparison across metrics (computational complexity, communication complexity, AllReduce support, peak bandwidth), and as the authors explain, practitioners can "select your requirements, find the methods in Table 1 that satisfy them (those with green ticks), and choose any of these methods." While s9uQ requested more specific regime-based guidance, Table 1 enables practitioners to make informed choices. The claims about scope were appropriately toned down in the revision.

**Reviewer Scores:**

Reviewer sATR would likely remain at 4 or increase to 6. While the presentation improvements address their concerns, their low confidence (2) suggests they may not be well-positioned to evaluate this theoretical contribution. Their brief review provides limited technical engagement.

Reviewer ZST5 would maintain their score of 8. All their specific requests were addressed: tightness is justified through optimal time complexities, experiments for Cycle SGD were added (Section C.5), and sensitivity to parameter B was demonstrated (Section C.6). This reviewer clearly values the theoretical contribution and finds the framework "quite powerful."

Reviewer s9uQ would either maintain or increase their score from 6 to 8. Their technical questions were comprehensively answered, missing references were added, and the clarifications about gradient aggregation and worker dependencies in convergence bounds were satisfactory. As they stated in their follow-up, "I appreciate the authors' effort to address my comments." While concerns about method proliferation and scope remain, this reviewer recognizes that "a unified representation is valuable for researchers working in this field". The homogeneous setting focus, while limiting, does not invalidate the theoretical contributions.

Average hypothetical score: 6-7.3. Recommendation: accept. The paper makes genuine theoretical contributions: the tree representation provides new geometric intuition for distributed SGD methods, Theorem 2.4 offers a unified convergence analysis that is "solid and clean" (sATR), and the framework enables systematic design of methods with optimal time complexity. As s9uQ notes, the unified framework has value even independent of new algorithms or improved bounds. The limitation to homogeneous settings restricts practical applicability but does not diminish the theoretical insights.

---

### Decision · Program_Chairs · 2026-01-26

Accept (Poster)